



# Long-term INP measurements from four stations across the globe

Jann Schrod[1], Erik S. Thomson[2], Daniel Weber[1,3], Jens Kossmann[1], Christopher Pöhlker[4],
Jorge Saturno[4,5], Florian Ditas[4,6], Paulo Artaxo[7], Valérie Clouard[8,9], Jean-Marie Saurel[8], Martin Ebert[10],
Joachim Curtius[1], and Heinz G. Bingemer[1]

[1]Institute for Atmospheric and Environmental Sciences, Goethe University Frankfurt, Frankfurt am Main, Germany
[2]Department of Chemistry and Molecular Biology, Atmospheric Science, University of Gothenburg, Gothenburg, Sweden
[3]now at: Federal Waterways Engineering and Research Institute, Karlsruhe, Germany
[4]Max Planck Institute for Chemistry, Biogeochemistry and Multiphase Chemistry Departments, Mainz, Germany
[5]now at: Physikalisch-Technische Bundesanstalt, Braunschweig, Germany
[6]now at: Hessisches Landesamt für Naturschutz, Umwelt und Geologie, Wiesbaden, Germany
[7]Physics Institute, University of São Paulo, São Paulo, Brazil
[8]Université de Paris, Institut de physique du globe de Paris, Paris, France
[9]now at: Géosciences Environnement Toulouse, Toulouse, France
[10]Institute for Applied Geosciences, Technical University of Darmstadt, Darmstadt, Germany

**Correspondence:** Jann Schrod (schrod@iau.uni-frankfurt.de)

**Abstract.**

Ice particle activation and evolution have important atmospheric implications for cloud formation, initiation of precipitation and radiative interactions. The initial formation of atmospheric ice by heterogeneous ice nucleation requires the presence of a nucleating seed, an ice nucleating particle (INP), to facilitate its first emergence. Unfortunately, few long-term measurements of INPs exist and as a result, knowledge about geographic and seasonal variations of INP concentrations is sparse. Here we present data from nearly two years of INP measurements from four stations in different regions of the world: the Amazon (Brazil), the Caribbean (Martinique), Central Europe (Germany) and the Arctic (Svalbard). The sites feature diverse geographical climates and ecosystems that are associated with dissimilar transport patterns, aerosol characteristics and levels of anthropogenic impact (ranging from near pristine to mostly rural). Interestingly, observed INP concentrations, which represent measurements in the deposition and condensation freezing modes, do not differ greatly from site to site, but usually fall well within the same order of magnitude. Moreover, short-term variability overwhelms all long-term trends and/or seasonality in the INP concentration at all locations. An analysis of the frequency distributions of INP concentrations suggests that INPs tend to be well-mixed and reflective of large-scale air mass movements. No universal physical or chemical parameter could be identified to be a causal link driving INP climatology, highlighting the complex nature of the ice nucleation process. Amazonian INP concentrations were mostly unaffected by the biomass burning season, even though aerosol concentrations increase by a factor of 10 from the wet to dry season. Caribbean INPs were positively correlated to parameters related to transported mineral dust, which is known to increase during the northern hemispheric summer. A wind sector analysis revealed the absence of an anthropogenic impact on average INP concentrations at the Central European site. Likewise, no Arctic Haze influence was observed on INPs at the Arctic site, where low concentrations were generally measured. We consider the collected data to be a unique resource for the





community that illustrates some of the challenges and knowledge gaps of the field in general, while specifically highlighting the need for more long-term observations of INPs worldwide.

# 1 Introduction

Ice nucleating particles (INPs) are a crucial element in cloud formation and precipitation processes (DeMott et al., 2010; Lohmann, 2015). INPs are a rare subclass of aerosol particles with special physicochemical properties that enable the first emergence of ice crystals by reducing the critical energy barrier for spontaneous nucleation (Vali et al., 2015). Depending on ambient temperature and supersaturation, INPs serve as a kind of cold temperature equivalent to cloud condensation nuclei (CCN) in the atmosphere. Known species of INPs include mineral dust, soil dust, primary biological aerosol particles and marine (biological) aerosol. In a supercooled and supersaturated cloud regime INPs may activate to ice crystals, which will then grow and possibly form secondary ice by splintering or other multiplication processes. Once grown to critical size, crystals may initiate precipitation. This is especially important for mixed-phase clouds that consist of both supercooled water droplets and ice crystals. In the presence of ice crystals water droplets will evaporate, feeding the crystals with more water vapor, a phenomenon known as the Wegener-Bergeron-Findeisen process. It is well established that the majority of global precipitation is formed through this pathway, especially over continental regions and the mid-latitude oceans (e.g. Mülmenstädt et al., 2015). INPs also influence local and global radiation budgets and related aerosol cloud interactions by affecting the phase of clouds (Lohmann, 2015).

Ice nucleation research first received some scientific attention in the 1950s and '60s, and since then interest has intensified, especially during the last one or two decades (DeMott et al., 2011). However, due to several difficulties in quantifying and characterizing INPs in the atmosphere, there are still large knowledge gaps concerning geographic and vertical distributions, seasonal and/or interannual variations, chemical composition and sources of INPs. Although there have been significant advances in identifying globally relevant INP species (e.g. Atkinson et al., 2013; O'Sullivan et al., 2015; Wilson et al., 2015; Hiranuma et al., 2019), understanding microscopic freezing processes (e.g. Marcolli, 2014; Kiselev et al., 2017; David et al., 2019), parameterizing INP concentrations (e.g. DeMott et al., 2010; Niemand et al., 2012; DeMott et al., 2015), intercomparing INP instrumentation in standardized procedures (Hiranuma et al., 2015; DeMott et al., 2018; Hiranuma et al., 2019) and designing and characterizing new measurement techniques (e.g. Garimella et al., 2016; Schrod et al., 2016; Lacher et al., 2019), we are still far from having a complete picture. A major shortcoming is the lack of global coverage and continuous longer-term observations of INPs. Large regions of the Earth (including whole continents and oceans) are underrepresented or even completely missed by INP measurements. For those regions where observations exist, measurements primarily cover periods of a few days or weeks. Very few published INP measurements qualify as long-term observations that cover multiple seasons or year-to-year variations. A further obstacle to INP monitoring is that many instruments are currently not suited for sustained, long-term monitoring tasks due to their complex and labor intensive operating principles.

It is noteworthy, however, that several projects in the early years of ice nucleation research actually succeeded in acquiring longer-term records of INP abundance. Although changes in instrumentation and sampling techniques sometimes make it



difficult to utilize these decades old results in an absolute sense, the trends and relative results are quite informative. For example, Soulage (1966) coordinated a regional network of nine European stations that synchronously collected samples during the summer of 1964. In that study INP samples were analyzed using mixing chambers at $-21\,°C$. In the resulting publication the author speculated that some positive anomalies in the record of INPs were associated with episodes of advected Saharan

dust and/or might have been affected by industrial particles. From 1959 to 1962 Kline (1963) measured INPs for the U.S. National Weather Service at 15 sites using an expansion-type chamber that operated from $-20\,°C$ to $-24\,°C$. The measurements present evidence that terrestrial aerosol particles dominate the INP budget of the lower atmosphere. The geographic site-to-site variability was about an order of magnitude, while day-to-day fluctuations of up to several orders of magnitude were recorded at single sites. Bigg collected INPs on membrane filters across Eastern Australia and New Zealand during several months of

1962 – 1964 (Bigg and Miles, 1964). Later he continued sampling in the marine boundary layer over large parts of the remote Southern Ocean around Australia from 1969 to 1972 (Bigg, 1973). Samples were analyzed in a thermal vapor diffusion chamber. Bigg's most striking results were (i) the similarity of INP abundances in the continental atmosphere and in the marine boundary layer, and (ii) the occurrence of high concentrations of INPs in remote areas far west and east of the Australian continent. In another multi-year study from 1964 to 1968 the Austrian Meteorological Service (Zentralanstalt für Meteorologie

und Geodynamik) measured the abundance of INPs at $-21\,°C$ using the method of Soulage (1965) at three sites, three times per day (Müller, 1969). That data also displays a relatively high day-to-day variability of INP abundance. The sparseness of these few historic measurement efforts highlights the need for more long-term INP observations.

Measurements of cloud-active aerosols in remote and/or near pristine environments are particularly rare and therefore inherently valuable. Such regions may be studied to gain insight into aerosol conditions in environments that are only marginally

perturbed by humans. This information is needed to estimate the reference baseline of pre-industrial aerosol (Carslaw et al., 2017). Such a baseline is vital to the accurate evaluation of anthropogenic climate effects, as it is integral to the assessment of the anthropogenic contribution to present day radiative forcing. All estimates of anthropogenic aerosol effects are highly sensitive to the assumed pre-industrial baseline, including the degree to which cooling aerosol effects have compensated for the radiative forcing by greenhouse gases in the past and present (Andreae et al., 2005; Andreae, 2007; Carslaw et al., 2013;

Gordon et al., 2016, 2017). Moreover, the largest uncertainty with respect to global radiative forcing emerges from knowledge gaps related to interactions between aerosols and clouds, as highlighted within IPCC assessment reports (IPCC, 2014). Given the poor understanding of INP climatology and life cycles, it is not surprising that the magnitude and effects of a potential anthropogenic INP perturbation cannot yet be assessed (Boucher et al., 2013, IPCC AR5, chapter 7). Similarly, in a review of the state of knowledge on pre-industrial aerosols Carslaw et al. (2017) were unable to comprehensively discuss the matter due to a

lack of thorough understanding regarding which aerosol components dominate the INP spectra. They argue that the concentrations of INPs, which tend to be large particles, likely have not changed as much as those of smaller particles, which have been found to be significantly altered since the pre-industrial era (Hamilton, 2015; Gordon et al., 2017). Nonetheless, Carslaw et al. (2017) acknowledge that potential anthropogenic modifications to INP concentrations or compositions and related impacts on cloud formation, radiation interactions and precipitation processes since the industrialization remain unquantified.





Here we present long-term measurement data of INPs from a small but unique network of stations spanning over 80° in latitude. Observational sites were located within vastly contrasting climates and ecosystems, featuring continental tropical, marine subtropical, continental mid-latitude and Arctic mountaintop locations. The sites are exposed to varying and seasonally different degrees of anthropogenic influence; yet all can be classified as rural, remote and/or pristine environments for at least

parts of the year. However, truly pristine regions, which still resemble their pre-industrial state in all facets, may be hard to find on an increasingly polluted planet (Hamilton et al., 2014). For this study, INPs were sampled at the Amazon Tall Tower Observatory (Brazil), the Volcanological and Seismological Observatory of Martinique (Caribbean Sea), the Taunus Observatory (Germany) and the Zeppelin Observatory (Norwegian Arctic).

Between May 2015 and January 2017 a total of 1212 aerosol samples (7704 data points) were collected and analyzed for INPs

in the deposition and condensation freezing modes (Fig. 1). Aerosol samples were collected using electrostatic precipitation onto silicon substrates in a semi-automated routine. Samples were shipped to our laboratory in Frankfurt and were subsequently analyzed for ice nucleation activity using the FRIDGE isothermal vacuum diffusion chamber (Schrod et al., 2016, 2017). Each sample was analyzed at multiple combinations of temperature ($-20\,°C$, $-25\,°C$ and $-30\,°C$) and relative humidity (95%, 97%, 99%, 101% w.r.t. water).

The main objectives of this study were to (1) observe the long-term concentrations and variability of INPs, (2) investigate potential trends and/or seasonalities, (3) compare the INP concentrations of diverse geographic locations, (4) estimate the anthropogenic impact on INPs at semi-pristine sites and (5) try to identify what factors control ice nucleation in the atmosphere.

## 2  Methods

### 2.1  Aerosol sampling

In all locations aerosol samples were collected using the Programmable Electrostatic Aerosol Collector (PEAC7, Schrod et al., 2016). PEAC7 precipitates aerosol particles, which have been charged by collision with corona-discharge electrons, electrostatically onto a semi-conducting grounded sample substrate made from commercially available silicon wafers. The 45 mm diameter substrates have three laser-engraved crosses used to generate a coordinate system that allows ice crystals and thus particles to be located in the INP counter FRIDGE (Sec. 2.2). Electrostatic precipitation is advantageous compared to simple

impaction, as particles are distributed more homogeneously across the surface. Thereby, FRIDGE is able to activate and count up to about 1000 separate ice crystals simultaneously on one sample substrate. The PEAC7 collection efficiency has been found to be about 60 %, independent of particle size (Schrod et al., 2016). No inlet size-cutoffs were used for the results presented here, and thus we expect to sample the complete particle spectrum, except for the usual particle losses that may occur for large particle sizes. This is of some advantage compared to in-situ INP counters with optical detection that need to distinguish

unactivated large aerosol particles from activated INP (i.e. ice crystals) by eliminating large particle intake ($> 1.5$ or $2.5\,\mu m$). Since ice nucleation tends to increase with particle size/surface area (DeMott et al., 2010), this may bias results.

PEAC7 utilizes a step motor powered rotary disc with seven sample substrate slots for programmed sampling. When connected to a PC with an internet connection, PEAC7 can be programmed either directly or remotely to start and stop sampling





at prescribed times. This configuration enables daily sampling for one week with minimal service and maintenance. Combined with the ease of operation, this makes PEAC7 a well-suited instrument for use within an INP monitoring network.

A PEAC7 unit was first installed in 2012 at the Mt. Kleiner Feldberg Observatory of the Goethe University of Frankfurt. PEAC7 units were deployed at the other three sites during the summer and fall of 2014. The local staff of each observatory conducted the regular measurements and maintenance after being trained in the handling of the instrument. Concurrent sampling began in May 2015 and continued (with some interruptions) until January 2017 for the three overseas stations. The measurements at Taunus Observatory began earlier and continued longer, but here we focus on the concurrent sampling effort conducted within the framework of the EU FP7 BACCHUS project (Fig. 1). Due to the failure of the complementary aerosol instrumentation at the Caribbean site, sampling was interrupted between December 2015 and May 2016. Furthermore, the exchange of sampling substrates with the Amazonian site was logistically challenging. Thus, the Amazon data set is represented by several shorter periods of continuous measurements. Difficulties in obtaining sufficiently clean new substrates also prevented measurements below $-25\,°\mathrm{C}$ between October 2015 and February 2016 in some cases.

Typically, aerosol samples were collected with PEAC7 daily or once every two days at local noon. Sampling time was prescribed to 50 minutes with a $2\,\mathrm{L\,min^{-1}}$ flow rate. The sampled aerosol particles resulting from this $100\,\mathrm{L}$ of air were found to generate well-resolved ice crystal numbers for a broad spectrum of temperatures using the FRIDGE analysis system. Samples were stored in PetriSlide containers after collection until they were shipped in packages of $25-50$ to our laboratory in Frankfurt (transport time was usually less than a week). As a result, several weeks often passed between sample collection and analysis, which may introduce an aging effect. However, in a previous study no effect of storage time on ice nucleation activity was observed within the investigated temperature regime (Schrod et al., 2016).

## 2.2 Analysis of ice nucleation samples

FRIDGE is an isothermal static diffusion chamber for offline analysis of ice nucleation. In its standard operation mode FRIDGE analyzes deposition and condensation mode INPs on substrates that had been laden with atmospheric aerosol particles by electrostatic precipitation. FRIDGE was originally introduced by Bundke et al. (2008) and Klein et al. (2010), but was fundamentally reevaluated and updated by Schrod et al. (2016). Since this effort FRIDGE has participated in meaningful laboratory intercomparisons (Hiranuma et al., 2015; DeMott et al., 2018; Hiranuma et al., 2019) and field campaigns (Schrod et al., 2017; Thomson et al., 2018; Gute et al., 2019; Marinou et al., 2019). In this context FRIDGE has been validated as a reliable method for INP measurements and can be regarded as a valuable addition to the widely used online continuous flow diffusion chambers and offline droplet freezing assays. To avoid confusion, we like to point out that the FRIDGE instrument can in fact be modified to serve as a cold stage for droplet freezing assay measurements as well, which was, however, not done for the results presented here.

During measurements the sample substrate is placed on the cold table inside a sealed measurement cell. The temperature of the substrate is controlled by a Peltier element and monitored by a PT-1000 sensor at the wafer surface. The measurement cell is connected by a valve to a water vapor source. The vapor source vessel is evacuated, except for water vapor from a thin ice film coating the inner walls, which are temperature-controlled by a cryostat (Huber Petit Fleur). Thus, the temperature of ice





film defines the water vapor pressure (Clausius-Clapeyron equation), which is measured by an Edwards Barocel capacitance manometer. The measurement cell is kept at near vacuum conditions, until a controlled amount of water vapor is introduced as a measurement begins by opening the valve to the vapor source. Ice crystals activate rapidly on the surface of INPs and grow to macroscopic sizes within some tens of seconds. The ice crystals are counted automatically by a CCD camera viewing the

measurement from above. After a measurement is completed, the valve to the water vapor source is closed and the valve to the vacuum pump is opened. Subsequently, ice crystals evaporate and a new combination of temperature and ice supersaturation can be selected (Tab. 2). Typical measurement uncertainties are summarized in the caption of Tab. 3. A complete description of the method can be found in Schrod et al. (2016).

One of the main strengths of FRIDGE is the direct visual observation of the ice crystal formation on the surface of the

sample substrate. No complicated data analysis is required to establish the number of INPs. Furthermore, knowing the exact location of a specific ice crystal, allows for a subsequent electron microscopy analysis to determine the chemical composition and morphology of individual INPs. We refer to our previous study for methodological details about coupling FRIDGE to an scanning electron microscope (Schrod et al., 2017).

## 2.3  Measurement Sites

### 2.3.1  Amazon Tall Tower Observatory – AZ

The Amazon Tall Tower Observatory (ATTO, 2.144° S, 59.000° W, 130 m AMSL) was established to investigate atmosphere-biosphere interactions and for observing long-term changes in the Amazonian environment. The ATTO site is located about 150 km northeast of Manaus, Brazil. Atmospheric measurements have been conducted here since 2012, when two 80 m towers were constructed. The main ATTO tower (325 m) was finished in 2015.

The Amazon basin contains the largest rainforest in the world and thus has great importance for global and regional carbon- and water cycles and biodiversity. Although the "green ocean" is unparalleled in size, distinct changes in the water- and energy budgets of the Amazon basin are becoming apparent due to anthropogenic impacts such as agricultural expansion, deforestation and climate change (Davidson et al., 2012). According to Davidson et al. (2012) the Amazon basin is already in transition and it is therefore important to monitor changes and their related effects on the biosphere and the atmosphere. Andreae et al.

(2015) present a detailed description of the site characteristics and provide an overview of the vast array of ecological, meteorological, trace gas and aerosol measurements at ATTO. Pöhlker et al. (2019) expanded upon the general site characterization by presenting a comprehensive analysis of the backward trajectory footprint region for the ATTO site and included an in-depth discussion about land cover transformations. Back trajectories differ significantly between the wet season (February to May: northeast) and the dry season (August to November: southeast). There is also a distinct seasonality in pollution markers mea-

sured at the site (Saturno et al., 2018; Holanda et al., 2020). During the dry season biomass burning heavily influences the site, resulting in an order of magnitude increase in aerosol number concentration. Pöhlker et al. (2016) conducted size segregated CCN measurements that provide near continuous coverage of a complete seasonal cycle. They found a pronounced CCN seasonality that covaries with both aerosol number concentration and pollution markers. However, during the wet season





aerosol conditions remain largely unaffected by pollution and are considered to be comparatively clean. According to Pöhlker et al. (2018) there are typically 10 to 40 days from March to May, which can be considered as pristine. During both dry and wet season (and especially in February and March) plumes of long-range transported aerosol are relatively frequent. These aerosols include Saharan dust (mainly wet season), particles from biomass burning in Africa and sea salt from the Atlantic

Ocean. During such episodes of long-range transport, coarse mode particle concentrations may rise above $100\,\mu g\,m^{-3}$, altering the aerosol size spectrum and composition substantially (Moran-Zuloaga et al., 2018). Except for these singular events, coarse mode particle concentrations remain fairly constant throughout the year, showing only a weak seasonality. In the absence of long-range transport, primary biological aerosol particles (PBAP) dominate the coarse mode population. It is important to note here, however, that PBAPs peak during the night time in Amazonia. At local noon, when the samples were collected,

concentrations of PBAPs are typically a factor of $2-5$ lower (Huffman et al., 2012).

The INP sampling device PEAC7 was installed inside a container at the base of one of the smaller $80\,m$ towers. Ambient air was introduced to PEAC7 through a $25\,mm$ stainless steel line, connected to a total suspended particle inlet at $60\,m$ AGL (i.e. $30\,m$ above canopy height). Moran-Zuloaga et al. (2018) show that losses of particles $< 2\,\mu m$ at realistic particle densities are usually well below $10\%$ for this setup and the transmission efficiency only drops below $50\%$ for particles larger than $6\,\mu m$ and

a high particle density of $2\,g\,cm^{-3}$.

INP measurements from this site are labeled with the abbreviation AZ.

### 2.3.2   Volcanological and Seismological Observatory of Martinique – MQ

The Volcanological and Seismological Observatory of Martinique (OVSM, 14.735° N, 61.147° W, $487\,m$ AMSL) is located on the Morne des Cadets mountaintop in northwestern Martinique, which is an island in the Lesser Antilles in the Caribbean

Sea. The observatory, in its current form, was built in 1937 after scientific interest increased following a devastating volcanic eruption of the close-by Mt. Pelée in 1902 and a second period of activity from 1929 to 1932. The observatory is operated by the Institut de Physique du Globe de Paris and closely monitors the local volcanic and regional seismic activity.

About two thirds of Martinique is protected by regional natural parks to preserve the island's environment. A regional park completely surrounds the observatory and encompasses the majority of northwestern Martinique, which contains large areas

that are labeled as natural zones of major interest (PNRM). The observatory is about $15\,km$ north of the capital Fort-de-France and about $20\,km$ northwest of the island's airport. Eastern trade winds are dominant and air masses reaching the site are primarily of maritime origin. Thus, we find it unlikely that our measurements of INPs are significantly influenced by local pollution.

Similar to the Amazonian site, the Caribbean site is also subject to a seasonality in precipitation and atmospheric transport

patterns due to the migration of the Intertropical Convergence Zone (ITCZ). The dry season begins in December and ends in May, the wet season lasts from June to November.

Stevens et al. (2016) analyzed two years of daily 10-day back trajectories arriving at $3\,km$ over Barbados. Qualitatively, this analysis should also be representative of the large scale transport pattern to Martinique, which is only $200\,km$ northwest of Barbados. It was found that the majority of air masses originated north of 10° N and east of 55° W. Approximately half





of the air masses traveled from this direction during the dry season and about two-thirds during the wet season, respectively. During the dry season 8% of these air masses are influenced by the European or African continent(s), and 55% during the wet season. Accordingly, seasonal wind shifts regulate the amount of long-range transported aerosol arriving in the Caribbean. The maximum contribution of Saharan mineral dust over the Lesser Antilles is found in the Northern Hemisphere (NH) summer.

Yet, the interactions between dust and precipitation introduce considerable variability in aerosol optical depth (AOD) during the wet season (Stevens et al., 2016). Some "clean periods" with AOD below 0.01 were observed, despite the generally heavy dust load during the wet season. In the dry season the amount of dust reaching the Caribbean is reduced considerably. The long-term trends, variation and seasonality of mineral dust transported to the Caribbean have been monitored almost continuously since 1965 (Prospero and Lamb, 2003). The Lesser Antilles' location at the end of the transatlantic trade wind flow and the

well characterized dust fraction make it an excellent place to investigate the influence of mineral dust on cloud formation.

Sea salt makes up most of the remaining mass fraction of Caribbean aerosol. The mass concentration of sea salt is typically of the same order of magnitude as mineral dust, yet its seasonality is different (Stevens et al., 2016). Due to higher wet scavenging and slower wind speeds from June to November, the sea salt contribution is at a minimum during the wet season and a maximum during the dry season.

To date few investigations of atmospheric aerosol and cloud formation have been conducted in the Lesser Antilles and none at OVSM. In 2011 the extensive field campaign DOMEX-2011, which focused on the formation of orographic clouds and related precipitation events, was based from Dominica, the island just to the north of Martinique (Smith et al., 2012). The DOMEX campaign, which included several research flights, found that the clouds and precipitation were strongly sensitive to trade wind speeds and therefore local dynamics and convection. In that campaign INPs were not considered as an important

variable or driver of clouds and precipitation.

Prior to the initiation of the PEAC7 sampling in September 2014, OVSM was not equipped with aerosol instrumentation. Therefore, a TSI OPS 3330 (optical particle diameter: $0.3 - 10\,\mu\text{m}$) was installed to compliment the PEAC7 measurements. Both instruments were connected to a $2\,\text{m}$ stainless steel line, mounted on the north side of the building. The inlet was open to freely circulating air coming from west, north and the east (main wind direction) and was protected from precipitation and

spray water by a custom made lid.

INP measurements from this site are labeled with the abbreviation MQ.

### 2.3.3   Taunus Observatory – TO

The Taunus Observatory (TO, 50.221° N, 8.446° E) is located on top of the Mt. Kleiner Feldberg ($825\,\text{m}$ AMSL) within the Taunus highlands of central Germany. There are several mountain peaks of similar height in the immediate vicinity (e.g. Mt.

Großer Feldberg at $878\,\text{m}$ AMSL and Mt. Altkönig at $798\,\text{m}$ AMSL). The Taunus mountains are nearly completely forested, predominantly with coniferous trees. Sobanski et al. (2016) described the land cover of the area surrounding the Kleiner Feldberg, and found that about 80% of the area within $5\,\text{km}$ is covered by forest. Within $50\,\text{km}$ about one third of the area is forested, while agriculture makes up another 40% and urban areas about 10%. The Taunus mountain range extends about $70\,\text{km}$ from the Rhine river to the northeast and serves as a natural barrier to the Rhine-Main metropolitan region, with its





center located to the south of the range. The Rhine-Main metropolitan region is heavily industrialized and densely populated, with about 2.2 million people living in and around the city of Frankfurt. The city lies about 20 km southeast of the Taunus Observatory. At the southwestern end of Frankfurt is the industrial area Höchst, which is one of the largest chemical and pharmaceutical industrial sites in Europe. The Frankfurt Airport is also roughly 20 km to the south of the observatory. The cities
of Wiesbaden and Mainz are also located 20 – 30 km to the southwest. In contrast, the northern sector is sparsely populated and predominantly devoid of industrial influence for 50 to 100 km.

Pollutant data, measured routinely by the Hessian Agency for Nature Conservation, Environment and Geology (HLNUG) using a Horiba APNA 370 (NO and $NO_2$), a Horiba APOA 370 ($O_3$) and Digitel DHA-80 ($PM_{10}$), has been analyzed for TO for the years 2015 to 2017, in order to quantify the predominant wind direction at TO and the anthropogenic influence.
Thirty-minute mean concentrations of pollutants have been divided into wind sectors and are presented in Table 1. As expected, pollutant concentrations are significantly higher when originating from the metropolitan sector, compared to air coming from other directions. However, as can be seen in Table 1 the site is rarely downwind from the highest pollution sources. In fact, the main wind direction is west. Thus generally, the site may be categorized as primarily rural with anthropogenic impacts.

INP measurements from this site are labeled with the abbreviation TO.

### 15  2.3.4   Zeppelin Observatory – SB

The Zeppelin Observatory, operated by the Norwegian Polar Institute, is located on Zeppelin Mountain close to Ny-Ålesund in Svalbard (78.908° N, 11.881° E, 474 m AMSL). Svalbard, and Ny-Ålesund in particular, is a well-established site for Arctic and atmospheric research. The scientific focus of the observatory is to characterize the Arctic atmosphere and identify relevant atmospheric processes in a changing Arctic climate. The mountain top Zeppelin Observatory was chosen for its elevated
position, which limits the effects of local pollution and direct influence of sea salt aerosol. However, the observatory largely remains within the planetary boundary layer (Tunved et al., 2013). The station is representative of the remote Arctic, making it a unique location to study atmospheric aerosol. A variety of trace gases, greenhouse gases, aerosol particles, heavy metals and other compounds are monitored continuously at Zeppelin.

Tunved et al. (2013) calculated a monthly climatology of air masses arriving at Mt. Zeppelin between 2000 and 2010. They
observed two primary transport patterns: For most of the year trajectories predominantly originated from Siberia and Eurasia. These air masses are mainly transported over the Arctic Ocean before arriving at Svalbard. During the summer months Atlantic air masses arriving from the southwest are most frequent. Although the Arctic is generally associated with clean atmospheric conditions, there are times of the year when contaminants are transported to the Arctic, leading to a significant decrease of air quality. This so-called Arctic Haze phenomenon has been long known and is well-studied. The Arctic Haze occurs during
late winter and spring when air is transported from industrialized source regions in Eurasia and North America. Tunved et al. (2013) observed the aerosol mass concentration over a period of 10 years at Zeppelin Observatory and confirmed an annually repeating Arctic Haze signal with a maximum in spring. Long-term black carbon measurements at Zeppelin show virtually the same seasonal pattern (Eleftheriadis et al., 2009). Cruise ships have been identified as an additional important local source at Zeppelin (Eckhardt et al., 2013). In a generally clean environment this might be of importance to INPs, as ship emissions have





previously been observed to amplify INPs (Thomson et al., 2018). Weinbruch et al. (2012) analyzed over 50,000 individual particles in 27 aerosol samples collected between summer 2007 and winter 2008 at Mt. Zeppelin by electron microscopy. Potential INP-related particles, i.e. particles with a diameter larger than $0.5\,\mu\text{m}$, were mainly categorized as sea salt, aged sea salt, silicates or mixed particles (i.e. mixtures of sea salt, silicates and calcium sulphates). Mineral dust particles were found to
follow a seasonal pattern with a summer minimum.

INP measurements from this site are labeled with the abbreviation SB.

## 3 Results and Discussion

### 3.1 Concentrations, variations, trends and seasonality of INPs

Except when noted otherwise, the following discussion will focus on the highest ice supersaturation(s) $RH_{ice}$ at each of the
three examined activation temperatures (Tab. 2). These are highlighted, because pure deposition nucleation is considered to be relatively unimportant to ice nucleation at the observed temperatures of most mixed phase clouds (e.g. Murray et al., 2012). At these highest saturation conditions, at or slightly above water saturation, we expect the nucleation mechanism to be a mixture of deposition nucleation and (incomplete) condensation freezing. At lower supersaturations we qualitatively observe trends and variability in INPs that are similar to what we present here, but at lower absolute concentration levels.

In Figs. 2, 3 and 4 the INP concentrations from May 2015 to January 2017 at the four stations at $-20\,^{\circ}\text{C}$, $-25\,^{\circ}\text{C}$ and $-30\,^{\circ}\text{C}$ are presented. Key statistical parameters of the data set are summarized in Tab. 3. The most striking result from the time series is that the INP concentrations do not fundamentally differ from station to station. We find that average INP concentrations at the examined temperatures are of the same order of magnitude for all sites. This observation is somewhat surprising, since the sites represent drastically different environments. It seems that the climate and ecosystem defining characteristics, like the
geography of maritime versus continental locations, Arctic versus temperate versus tropical systems, and the altitude within the planetary boundary layer, are not overly critical to INP abundance. Instead, it appears as though these differences are mostly lost in the large variability of the INP concentrations. Figure 5 shows the day-to-day variability of the time series. The magnitude of the short-term variability is often almost as high as the total variability of the complete data set, which is represented as the standard deviation in the figure. Overall, short-term variability far outweighed any long-term trend or seasonality at any location
or temperature. In fact, mean INP concentrations remained remarkably constant throughout the investigated time period, which is apparent from the 10 point moving averages in Figs. 2, 3 and 4. Moreover, the lack of well-defined peak INP concentrations is evident (on the logarithmic scale). These findings are remarkable considering that the climatic and geographical features of the sites are accompanied by vastly dissimilar air mass transport patterns, aerosol source locations and levels of anthropogenic impact. However, it should be noted here that the collected data represents single 50 minute sampling intervals at local noon
with a frequency of 0.5 to 1 sample per day. The level to which the sampling strategy may implicitly result in the observed high short-term variability is uncertain and should be carefully explored. Comparisons with other recently published data sets suggest that long-term trends may be better captured using different sampling strategies (Höhler et al., 2019). Thus it remains an open challenge for the INP community to establish robust measurement protocols for monitoring efforts. Moreover, we



cannot entirely exclude the possibility that storage effects may have dampened the trends of the INP concentrations to some degree.

On average, INP concentrations were lowest at SB, which is what can be expected for an Arctic environment. Yet, mean INP concentrations at the other stations were only higher by a factor of $2-4$ at $-30\,^{\circ}\mathrm{C}$, with those at MQ being the greatest. Especially during the summer of 2015 MQ concentrations were relatively high. However, at the warmest temperature ($-20\,^{\circ}\mathrm{C}$), the highest INP concentrations are measured at TO. In addition, there are fewer samples below the detection limit or the significance level at $-20\,^{\circ}\mathrm{C}$ at TO. As the site is surrounded by forests, this might point to a local source of biological INPs, which are known to activate at warmer temperatures. Decreasing the nucleation temperature by $5\,^{\circ}\mathrm{C}$ enhances average INP concentrations by a factor of 2.4 to 5.6. However, as seen in Figs. 6 and S1, average INP concentrations depend predominantly on ice supersaturation. Decreasing temperature alone does not significantly enhance INP concentrations in the addressed nucleation mode. Rather, decreasing the temperature by $5\,^{\circ}\mathrm{C}$ implicitly leads to an increase in $\mathrm{RH_{ice}}$ by approximately $6\,\%$ for our ascribed conditions (Tab. 2). Thus, if there were any temperature dependence of note in our data, it would appear as a discontinuity in these plots. Figure 6a shows the median INP concentrations for all measured conditions at each site as a function of supersaturation. The figure implies strong exponential correlations between the INP concentrations and the ice supersaturation ($R^2$ between 0.95 and 0.98). Once more, we see that at TO more active INP are found at lower supersaturation, i.e. warmest temperature ($-20\,^{\circ}\mathrm{C}$). At intermediate ice supersaturations INP concentrations at TO, MQ and AZ are all similar. At the highest $\mathrm{RH_{ice}}$ ($-30\,^{\circ}\mathrm{C}$) AZ and MQ INP concentrations are the greatest. The concentrations at SB are lowest throughout the full RH spectrum, the reason likely being that the site is farthest away from substantial INP sources. Interestingly, the slopes fitted to the measurements from the European TO and SB stations are nearly identical, as are the slopes of data from the tropical MQ and AZ stations (Figs. 6a and 6b). The differences in the activation spectra may result from different contributions of certain species of INPs at the respective sites. MQ and AZ samples possibly entail a larger fraction of mineral dust compared to TO and SB samples, which may have led to the steeper increase of concentrations. Moreover, median INP concentrations at TO are a factor of approximately 2 higher than SB throughout the spectrum. An increase of $2\,\%$ in $\mathrm{RH_{ice}}$ results in 1.5 (SB and TO) to 1.7 (MQ and AZ) times higher INP concentrations. Increasing $\mathrm{RH_{ice}}$ by $10\,\%$ yields 7.4-fold (SB), 7.6-fold (TO), 12.8-fold (MQ) or 14.7-fold (AZ) changes in INP concentrations. Supplementary Fig. S1 expands upon Fig. 6 by adding more statistical information, such as the arithmetic mean, the interquartile range and the 5–95 % range. The findings complement those presented in Fig. 6. Overall, the variability of relative abundance with temperature suggests that the dominant species of INPs do change temporally and between locations. The extent to which changes can be attributed to local versus more remote INP sources is an interesting question that should be a focus in future studies.

Figure 7 shows the relative frequency distributions of INP concentrations at $-20\,^{\circ}\mathrm{C}$ (a), $-25\,^{\circ}\mathrm{C}$ (b) and $-30\,^{\circ}\mathrm{C}$ (c). For the purposes of Fig. 7, samples below the detection limit (or with zero active INPs) or, in a few cases, overloaded samples are excluded. As a result, the distribution tails may be somewhat truncated, because the highest and lowest values are not adequately represented. Such an effect is likely more important at $-20\,^{\circ}\mathrm{C}$, because relatively more samples are below the background detection limit at this temperature. Scaled Gaussian fits in log space are added to emphasize the log-normal nature of the binned frequency distributions that emerge. Welti et al. (2018) analyzed INP data from the subtropical maritime boundary





layer and various other marine environments (Welti et al., 2020) in a similar fashion and argued that the observed log-normal nature of the distributions can be explained in an analogous manner to the distributions of pollutant species suggested by Ott (1990). The assertion is that for any species of interest (i.e. INPs), many consecutive random dilutions of an air mass containing that species will result in a log-normal distribution of species concentration. Such dilutions will naturally occur during transportation through the atmosphere from the sources to the measurement sites. Variations in source strength are associated with systematic shifts of the whole concentration distribution. For example, when a measurement site is close to a local source, a more left-skewed distribution is to be expected, as a higher proportion of air masses with fewer dilutions will occur. That said, the picture that we construct from the INP measurements made at a single point of arrival are convoluted, because there is not necessarily one singular source of INPs.

Considering the vastly different geographical locations and environments of the four measurement sites, as well as the inherent variance of atmospheric transportation patterns over time, we do not expect to find simple answers by inspection of the frequency distributions. A few interesting features are, however, apparent and the log-normal fitting agrees very well with the shape of the INP frequency distributions, which means that the dilution effect may be of importance here. The log-normal shape of the $-25\,°C$ and $-30\,°C$ distributions is especially evident ($R^2$ ranges from 0.92 to 0.97). Here we observe unimodal and regular bell shapes at all four sites. At $-20\,°C$ the fits are not as good ($R^2$ ranges from 0.74 to 0.91) and some distributions appear to be potentially bimodal. However, the strength of the fit may also be related to the fact that at $-20\,°C$ few ice crystals activate on each sample substrate, introducing a relatively high uncertainty in the INP concentration. Consequently, the incrementation is not ideal for $-20\,°C$, because measured concentrations are often near the limit of detection and have a poor resolution. This explanation is self-consistent with the observed minimum $R^2$ found for SB, where the distribution is heavily skewed to the right. In addition to reflecting the generally low INP concentration of the Arctic environment, this may point to reduced biological activity over much of the year. Interestingly, the shape of the distribution at TO seems to indicate a slight shift towards higher concentrations, pointing to a potential local source of INPs. However, at lower temperatures we do not find this feature. This could mean that, in addition to whatever long-range transported aerosols contribute to INPs at TO, there might be a biological source from the surrounding forest. However, there is no strong evidence for such a signal in our data overall. Remarkably, such a feature seems to be absent from the Amazonian rainforest site, where one would more readily expect to find a local source of primary biological particles that may be potential INPs. On the other hand, surface temperatures never drop below $0\,°C$ in the Amazon. Therefore, local species of plants or bacteria may be less likely to have evolved traits that induce freezing in order to gain an evolutionary benefit as is believed to be the case for certain bacteria such as *Pseudomonas syringae* (Morris et al., 2014).

At $-25\,°C$ we find relatively minor differences between the four sites. SB concentrations are slightly shifted to lower concentrations and the spectrum at MQ concentrations is slightly broader. Differences are more apparent at $-30\,°C$. Here we find distinctly dissimilar shapes of INP concentration frequency distributions. SB and AZ exhibit narrow peaks relative to the more broad shapes of TO and MQ. The curves are also more distinctly separated in concentration space, with the maximum of the distribution at a minimum concentration for SB, followed by TO, MQ and AZ.





Supplementary Fig. S2 visualizes the information presented in Fig. 7 as function of the relative humidity. The occurrence frequency is color-coded, with cool colors indicating a low and warm colors a high likelihood of this INP concentration at a given saturation condition. Thus Fig. S2 can be understood as follows: a single column (e.g. the rightmost column) gives the full frequency distribution of a single measurement condition (e.g. 135 % RH$_{\text{ice}}$, corresponds to Fig. 7c). Fewer warm colors

appear in a column, when the distribution of INP concentrations is broad at that condition. Conversely, fewer cool tones indicate a narrow distribution. The respective median INP concentration will be close to the maximum of the relative frequency at each condition. Consequently, following the maxima yields information about the steepness of the INP spectra, similar to what is depicted in Fig. 6.

Overall, Figs. 7 and S2 suggest that the INP concentrations measured in the investigated temperature regime at these stations

are largely defined by background air masses, and that local sources are only of secondary importance.

### 3.2   Site specific INP characteristics

At each measurement station a diverse array of supplementary meteorological, aerosol and gas data from the stations were collected in parallel to the INP sampling. Unfortunately, the parameters, instrumentation and time coverage vary considerably between the four sites. Observations include typical meteorological parameters such as temperature, relative humidity, pre-

cipitation, etc., as well as the total aerosol particle number and mass concentrations, aerosol size distributions, black carbon concentrations, aerosol optical thickness, gaseous pollutant markers and greenhouse gases. However, despite a rigorous effort including correlation analysis, factor analysis and trajectory sector analysis, we were ultimately unable to identify a single parameter or a set of parameters that account for the total observed variation of INPs. This highlights the complex nature of the ice nucleation process and the particles involved. Whereas similar but somewhat larger-scale long-term measurements of CCN

are able to largely explain the corresponding variability and provide closure studies (Schmale et al., 2018), unfortunately, the same cannot yet be said for INPs.

Although a common, definitive driver of INP climatology was not identified in our study, we will point out a few key findings specific to the respective measurement sites.

#### 3.2.1   AZ

The Amazonian site is characterized by a distinct seasonality of pollutants that follow the biomass burning season. During the dry season the aerosol concentration and other pollution markers rise by about one order of magnitude compared to the cleaner wet season – a change which is largely attributable to human activities. Notably, an effect of the strong anthropogenic biomass burning is absent in the INP signal. In fact, the number of INPs normalized by the total number of aerosol particles (TSI OPS 3330) in a volume of air (i.e. the activated fraction AF) is anti-correlated to parameters related to biomass burning

(Fig. 8). The AF can be understood as a simple metric that indicates the ice nucleating efficiency of particles within a specific aerosol sample. The observed anti-correlation suggests that aerosol particles from fire sources are relatively poor ice nuclei, an observation that agrees with previously published findings (Kanji et al., 2017, 2020). The significance of low AFs resulting from biomass burning in this study is difficult to assess, as the seasonality of the AF is largely dominated by the seasonal changes in




aerosol concentration for the AZ site. Vegetation fires therefore seem to emit disproportionally more (non ice-active) aerosol particles than INPs. Another way to interpret the anti-correlation of AF and markers of biomass burning is by coupling the metric to precipitation rates. There are several intricate interactions of note here. On the one hand more precipitation leads to a higher removal of aerosol particles (and INPs) by wet deposition. Moreover, the enhanced precipitation during the wet season

largely prevents wild fires and the accompanied particle emissions in the first place. On the other hand, it has been postulated previously that precipitation may be another driver of INP abundance, and large tropical rainforests like the Amazon have been highlighted in that regard (Morris et al., 2014). Huffman et al. (2013) suggested a connection between precipitation in a semi-arid pine forest and an increase in primary biological particle production, which subsequently might act as INPs. The processes responsible for the release of the biological particles have not yet been deciphered in detail. Huffman et al. (2013) hypothesized

that a) mechanical agitation by rain causes fungal spores and bacteria to be released into the air and/or b) mechanisms that stimulate bio-particle emissions (fungal spores, pollen fragments) are activated by a longer phase of high humidity and leaf moisture. Although the AZ measurements are somewhat more sparse than those of other stations, our observations do not support significant differences in absolute INP concentrations between dry and wet seasons.

Overall, the INP concentrations of our study compare reasonably well to the measurements of Prenni et al. (2009), who

observed average INP concentrations of about $1\,L^{-1}$ at $-20\,°C$, $4\,L^{-1}$ at $-25\,°C$ and $10\,L^{-1}$ at $-30\,°C$ using a continuous flow diffusion chamber (CFDC) to study condensation and immersion mode ice nucleation during a field campaign in February/March 2008 in a region close to the present location of the ATTO site. However, our observed concentrations are clustered at the low end of Prenni et al. (2009), which is likely due to the different nucleation modes addressed. During that short campaign Prenni et al. (2009) identified mineral dust and carbonaceous aerosol (mostly biological particles) to be the main contributors

to atmospheric INPs in the Amazon using transmission electron microscopy and energy-dispersive X-ray spectroscopy.

Within our sampling period, Moran-Zuloaga et al. (2018) identified several long-range transport (LRT) events at the site with markedly increased concentrations of mineral dust during the wet season of 2015/2016 (Dec./Jan.). INP concentrations of these LRT samples were positively correlated with the aerosol number concentration measured with an optical particle counter (TSI OPS 3330, $R = 0.80$, $N = 9$, $p < 0.01$). However, mineral dust may be a relevant INP in this region even in the absence of distinct

LRT events. An analysis of the average composition of INPs for six samples (4 in April 2016, 2 in December 2016) using scanning electron microscopy (Figure 9), identified that nearly half of the particles that activated to ice crystals in FRIDGE were mineral dust. Most of the characterized INPs in this study were measured to be between 1 and a couple of micrometers in diameter (Figure 9b). The second half of identified INPs had a strong carbonaceous fraction and consisted of biological particles and biomass burning products. Furthermore, it is possible that some PBAP activity was missed due to the chosen

sampling strategy, given local noon is a daily minimum for PBAPs. Qualitatively, these findings agree very well to those of Prenni et al. (2009).

### 3.2.2   MQ

Of the results presented here, the average INP concentration of the Caribbean site was the highest, but only by a small margin. There is some evidence that summertime INP concentrations are higher on average than those during winter, although there is





no clear seasonality. However, the possible seasonal ice nucleation effects are difficult to assess due to the large interruption of measurements between December 2015 to May 2016. Although we consider it rather speculative, a trend of higher concentrations during summer does stand to reason, as it would reflect the annual cycle of the mineral dust transport, which is driven by the movement of the ITCZ. The seasonality of mineral dust is well reflected by the $PM_{10}$ concentration, which is

monitored routinely in Martinique by the local agency for air quality (MadininAir). The seasonality of dust motivates a deeper investigation with respect to INPs. In general, we observe a significant correlation between the INP concentration at OVSM and the $PM_{10}$ concentration at an air quality station close to the observatory (Schoelcher, 14 km distance), as well as between INPs and the OPS aerosol number concentration at the observatory. The correlations improve for colder temperatures and higher ice supersaturations. At $-30\,°C$ and $135\,\%$ $RH_{ice}$ the Pearson correlation coefficients between INP and aerosols are $R = 0.46$

($N = 124$, $p \ll 0.01$) for $PM_{10}$ (Fig. S3) and $R = 0.50$ ($N = 69$, $p \ll 0.01$) for the OPS concentration, respectively. We conclude that the MQ INP concentration at the investigated temperatures is likely dominated by natural processes such as the long-range transport of Saharan mineral dust. However, there is still a large variability in the INP signal, which cannot be fully explained only considering the seasonal dust transport.

     We observe significantly lower INP concentrations for all conditions after the large interruption in measurements. For exam-

ple, the average INP concentration at $-30\,°C$ and $135\,\%$ $RH_{ice}$ in 2015 was $7.47 \pm 6.42\,L^{-1}$ ($N = 58$) and only $1.37 \pm 1.39\,L^{-1}$ ($N = 72$) in 2016. This observation does, however, correspond to measured $PM_{10}$ concentrations, which also show a significantly lower average in 2016 ($25\,\mu g\,m^{-3}$, $N = 8225$) than 2015 ($35\,\mu g\,m^{-3}$, $N = 8636$). Although, the observed 2015 to 2016 factor of 5 decrease in INP concentrations is large compared to the $\approx 30\%$ difference in $PM_{10}$, the cubic scaling implicit in the number to mass translation needs to be considered.

DeMott et al. (2016) presented results from offline immersion freezing experiments and characterized INP concentrations from research flights from St. Croix in the US Virgin Islands, and ground sampling from Puerto Rico, which were collected during the ICE-T campaign in July 2011. The focus was on marine INPs and determining representative marine background concentrations and only samples collected within the marine boundary layer were presented. They measured INP concentrations of $0.06\,L^{-1}$ at $-20\,°C$ and $0.3\,L^{-1}$ at $-24\,°C$, which are only a factor of 3 to 4 lower than our average concentrations at

$-20\,°C$ and $-25\,°C$, respectively. Considering that marine INPs are typically thought to represent a minor fraction of the total INP population (except for certain regions like the Southern Ocean), the comparison seems to be reasonable.

### 3.2.3   TO

During the time frame of the global sampling effort (about 640 days) 400 PEAC7 samples were collected and analyzed from TO (i.e. 1 sample every 1.6 days). The sampling frequency of valid INP concentrations (i.e. above the detection limit) remains

as good as 1 sample every 2 days for measurements at $-20\,°C$ and $-25\,°C$. This is by far the best data coverage of the four stations.

     We found a moderate but significant correlation between the $PM_{10}$ concentrations and INPs throughout the spectrum of $T$ and $RH$ conditions. The Pearson correlation coefficient is as high as $R = 0.28$ ($N = 293$) at $-25\,°C$, where we have the best data coverage. Although the particulate matter was significantly enhanced, when wind was coming from the heavily populated





and industrialized Rhine-Main metropolitan region (Tab. 1), the average INP concentration was not found to differ significantly from other times, when air masses were arriving from other directions (Fig. 10). Therefore, a strong anthropogenic impact on INPs at TO is unlikely.

### 3.2.4 SB

Given its remoteness and relatively clean atmosphere the Arctic may be particularly sensitive to small changes in aerosol particulate. Furthermore, within the Arctic climate system there are well known feedbacks that can amplify small changes in significant ways (Serreze and Francis, 2006; Boy et al., 2019). Historically, this has motivated quite a few research studies targeting ice nucleation in the Arctic environment. For example, clay was identified in the center of Greenlandic snow crystals by Kumai and Francis (1962) as early as 1960. Past studies generally agree that INP concentrations in the Arctic tend to be

on the lower side of the spectrum. Yet, individual findings and conclusions vary considerably (e.g. see Tab. 2 in Thomson et al., 2018). New ice core records may illuminate long-term trends of Arctic INPs by estimating historic (pre-industrial) concentrations from droplet freezing experiments of ice core melt water (Hartmann et al., 2019; Schrod et al., 2020).

In two recent studies immersion mode ice nucleation in the Arctic was investigated by Tobo et al. (2019) and Wex et al. (2019). Tobo et al. (2019) focused on two field campaigns held in Ny-Ålesund (Zeppelin) in July 2016 (6 samples) and March

2017 (7 samples). Wex et al. (2019) report INP concentrations from four pan-Arctic locations (Canada, Alaska, Ny-Ålesund and Greenland) that cover observations ranging from 10 weeks to a full year of mostly weekly sampling. Both studies observed enhanced INP concentrations during summer months. Tobo et al. (2019) report INP concentrations at $-20\,°C$ of about $0.01\,L^{-1}$ in March 2017 and about $0.1\,L^{-1}$ in July 2016. At $-25\,°C$ INP concentrations were on the order of $0.1\,L^{-1}$ and $0.5\,L^{-1}$ for the March and July field campaigns, respectively. Wex et al. (2019) distinguished between samples that were collected in

Ny-Ålesund from March to May 2012 (5 samples) and those from June to September 2012 (7 samples). During spring, INP concentrations at $-20\,°C$ were consistently found to be about $0.01\,L^{-1}$. Most summertime samples were completely frozen before reaching $-20\,°C$, and thus seem to to suggest that concentrations were up to one order of magnitude higher in summer. Further, Wex et al. (2019) report correlation coefficients with complementary measurements that are mostly insignificant including $PM_{10}$. Exceptions include significant correlations between INPs and sulphate (R = $-0.6$) and potassium (R = $-0.57$),

pointing to complex factors determining the Arctic INP population. Moreover, Tobo et al. (2019) present evidence that mineral dust (possibly with organic inclusions) from Arctic glacial outwash plains influence the INP activity in Ny-Ålesund. They conclude that these glacial sediments may be a large-scale source of mineral dust in the Arctic.

We present a significantly larger data set with respect to temporal coverage and our INP concentrations agree well with these previous studies from Ny-Ålesund. At $-20\,°C$ we find concentrations of about $0.1\,L^{-1}$. At $-25\,°C$ the average INP

concentration increases to about $0.3\,L^{-1}$. However, the consistently reported finding of summertime INP enhancement, does not emerge from our analysis. Furthermore, we did not observe any seasonal changes in the INP signal with regards to the anthropogenic Arctic Haze phenomenon. Moreover, we did not observe significant correlations between INPs and available aerosol parameters. The lack of meaningful correlations and/or seasonal trends may be in part related to a relatively poor signal-to-noise ratio for the SB measurements. INP concentrations were often at or close to the limit of detection or the significance





level, respectively. In retrospect, we now would increase the sampling volume for SB measurements to be able to resolve lower concentrations more accurately.

# 4    Conclusions

The data from our small but unique measurement network can be considered particularly valuable, and we hope lessons
can be taken from this effort that will help to guide future INP monitoring efforts. Significant infrastructural and logistical investments are represented by the INP measurements that cover an observational period of 21 months in total. Well above 1000 samples were collected, retrieved and analyzed in this project at a large array of temperature and supersaturation conditions, characterizing the INP concentrations in the deposition and condensation freezing modes. The investigated sites represent diverse climatic regions and ecosystems that experience varying degrees of anthropogenic influence.

In spite of the great differences in basically all characteristics that are expected to define the aerosol concentration, composition and source apportionment, we observed fairly similar INP concentrations for all four stations. Recently, Welti et al. (2020) reported a qualitatively similar finding: ship-based measurements of marine INPs in the Arctic, Atlantic, Pacific and Southern Ocean showed surprisingly little differences in the INP concentration despite their distant geographic locations. In our study, average concentrations differed between sites by less than a factor of 5. Short-term variability dominated most of the total
variability at all locations. Trends, annual cycles and well-defined peak concentrations were prominently absent from the time series. Still, the range of observed INP concentrations do compare reasonably well with previously published literature, where available. Importantly, the relative frequencies of observed INP concentrations are generally well-represented by log-normal distributions, a finding that suggests distributed INP sources that result from INPs being well-mixed within sampled air masses. These findings emphasize the important contribution of INPs from background air masses. Moreover, no physical or chemical
parameter was identified to continuously co-vary with INPs, and therefore a comprehensive causal link to INP concentrations remains lacking.

Overall, we did not detect much evidence for a strong anthropogenic impact on the concentrations of ice nucleating particles. At AZ the INP concentrations appear unrelated to human induced biomass burning, which otherwise leads to a tenfold increase in aerosol particle number concentrations during the dry season. The INP concentrations at MQ were well correlated with
aerosol characteristics that are driven by natural processes, like long-range transport of Saharan mineral dust and marine aerosol production. Average TO INP concentrations showed no significant difference between wind sectors that can be separated into anthropogenically dominated areas and rural environments. Likewise, no significant changes in the INP concentration were observed at SB during the Arctic Haze period.

Considering these findings, the approach of estimating order-of-magnitude pre-industrial INP concentrations from present-
day measurements in near-pristine locations does seem to both be viable and yields reasonable results, which merit further investigation. In this sense, we consider the lower concentration end of our measurements likely to be the most realistic assessment of pre-industrial atmospheric INP concentrations. However, we strongly advise cautious use of the presented data,




as there are substantial limitations and uncertainties in conceptional aspects of the approach, as well as to the measurements themselves.

It should be noted here that we cannot predict with certainty how these measurements would translate to the immersion freezing mode in the atmosphere. Measurements by the FRIDGE diffusion chamber and FRIDGE droplet freezing assay of previous campaigns were usually well-correlated, but the INP concentration in the standard mode is often lower by a factor in the order of a few up to one magnitude.

## 5 Outlook

This study clearly highlights that there is a strong need for increased continuous observations of INPs worldwide. Several important open questions need to be addressed by the community, when considering how to best implement a systematic long-term monitoring strategy:

1) What are the best time resolution(s) and sampling frequencies? Obviously, the answer to this depends on the scientific question that a group is trying to address and the requirements and capabilities of the specific INP counter. Furthermore, bearing in mind that INP measurements are primarily of interest for illuminating cloud and precipitation processes it may be important to recognize that cloud processes are geographically different, and thus the answers to these questions may also differ. For example, for areas where tropical and sub-tropical deep convection dominates cloud formation, sampling priorities may differ from the mid-latitudes where synoptic scale weather systems are predominant. Judging from the results we have presented, it could be argued that longer sampling times (several hours to days) are advantageous for elucidating longer-term trends. Longer sampling effectively acts as a low-pass filter and would reduce the considerable short-term variability in INPs that is observed everywhere. Naturally, ideal monitoring could be done with short but densely spaced sampling times. This would enable averaging to be done at the data analysis level and thus both short- and long-term variation could be reasonably captured. However, such an instrument and/or technique is currently not available and will likely present both technological and human resource challenges. In the meantime, we suggest that long-term measurements are initiated with some weeks of intensive measurements to establish baseline information with regard to INP concentration and variability.

2) What supporting instrumentation/measurements should accompany the INP monitoring? Naturally, it is good to have as much information about aerosol concentration, size distribution and chemical composition as possible. However, a thorough aerosol characterization does not guarantee robust correlations with INPs or even a fully explained total INP variability. To-date the community has had limited success in using co-located measurements to establish causal links with INPs. Initiating further long-term measurements at highly equipped research stations may be a pathway towards learning what additional tools best compliment INP studies.

3) What measurement conditions and nucleation modes should be addressed? Again, this depends on the both the researchers scientific focus and the capabilities of the respective INP instrument. However, the literature consensus firmly suggests that immersion freezing is the most atmospherically relevant nucleation mechanism.



From the experience built in this study, we recommend future studies, especially those that include remote sampling locations, plan conservatively for the required logistics and workload, with respect to sampling frequency, analysis conditions, sampling consistency, etc. We emphasize the importance of a well-conceived sampling strategy and well laid out logistics. Instrument malfunctions, maintenance and various other difficulties and interruptions are to be expected. All of these may be

easily addressed separately, but amount to a significant challenge, when a long-distance network is to be kept running across a hemisphere. Nonetheless, we encourage other groups and collaborations to more strongly emphasize long-time observations in addition to the common standard of campaign-based measurements. Although we are currently far from the best case scenario of an automated global network of INP measurements, there are promising new instruments (e.g. PINE, Lacher et al., 2019) that may provide a vital intermediate step towards long-term semi-automated measurements of immersion mode INPs in the

near future.

*Data availability.* The INP data will be made available using the Data Publisher for Earth & Environmental Science PANGAEA (https://www.pangaea.de/).

*Author contributions.* JS, JK and DW performed the INP measurements. JS compiled and analyzed the INP data with support of JK and HB. JS created the figures. Particle measurements at ATTO were performed by CP, JoS and FD. ME performed the SEM measurements. All

authors took part in the discussion of the results. JS wrote the manuscript together with HB and ET, receiving valuable input from the other co-authors.

*Competing interests.* The authors declare no competing interests.

*Acknowledgements.* The work was initiated and primarily carried out within the context of the EU FP7-ENV-2013 BACCHUS project under Grant Agreement 603445. The experimental methods and instruments were established through financial support of Deutsche Forschungs-

gemeinschaft under Research Cooperations SFB 641 and FOR 1525 (INUIT). We thank the technical and scientific personnel of the stations ATTO, Zeppelin Observatory (Norwegian Polar Institute), and of the Volcanological Observatory of Martinique (Frédéric Jadélus and David Mélézan) for their valuable help on site during the campaigns. We thank AWI and AWIPEV staff, including Verena Mohaupt, for the logistics provided in sample transport to and from Ny-Ålesund. We thank Hans-Christen Hansson for his valuable scientific input and the supporting aerosol data from Ny-Ålesund. We thank HLNUG and MadininAir for providing air quality data for TO and MQ, respectively. For the

operation of the ATTO site, we acknowledge the support by the German Federal Ministry of Education and Research (BMBF contract nos. 01LB1001A and 01LK1602B) and the Brazilian Ministério da Ciência, Tecnologia e Inovação (MCTI/FINEP contract 01.11.01248.00) as well as the Amazon State University (UEA), FAPEAM, LBA/INPA, and SDS/CEUC/RDS-Uatumã. This paper contains results of research conducted under the Technical/Scientific Cooperation Agreement between the National Institute for Amazonian Research, the State Uni-



versity of Amazonas, and the Max-Planck-Gesellschaft e.V.; the opinions expressed are the entire responsibility of the authors and not of the participating institutions. We further thank Susan Trumbore, Carlos Alberto Quesada, Reiner Ditz, Stefan Wolff, Jürgen Kesselmeier, Andrew Crozier, Antonio Huxley Melo Nascimento, Wallace Rabelo Costa, Daniel Moran-Zuloaga, Isabella Hrabe de Angelis, Maria Praß, Björn Nillius, Meinrat O. Andreae, and Ulrich Pöschl for their support and inspiring discussions. PA acknowledges funding from FAPESP
5   – Fundação de Amparo a Pesquisa do Estado de São Paulo through grant 2017-17047-0. EST's contribution has also been supported by the Swedish Research Councils, VR (2013-05153) and FORMAS (2017-00564) and the Swedish Strategic Research Area MERGE.





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





**Table 1.** Air quality data at Taunus Observatory as function of local wind direction: average values of major pollutants measured between 2015 and 2017 (HLNUG).

| Region | Wind sector [°] | Relative frequency | NO [µg m$^{-3}$] | NO$_2$ [µg m$^{-3}$] | O$_3$ [µg m$^{-3}$] | PM$_{10}$ [µg m$^{-3}$] |
|---|---|---|---|---|---|---|
| Frankfurt | 110 – 140 | 6% | 0.73 | 8.17 | 78.30 | 12.36 |
| Airport / Autobahn interchange | 150 – 170 | 3% | 1.08 | 11.67 | 78.64 | 11.23 |
| Wiesbaden / Mainz | 200 – 230 | 12% | 0.68 | 8.06 | 67.44 | 8.02 |
| Total Rhine-Main area | 110 – 230 | 29% | 0.78 | 9.26 | 72.79 | 9.87 |
| Not Rhine-Main (rural) | 0 – 110; 230 – 359 | 71% | 0.61 | 6.42 | 67.35 | 8.93 |
| All sectors | 0 – 359 | 100% | 0.66 | 7.25 | 68.96 | 9.21 |

**Table 2.** Thermodynamic conditions for INP analysis in FRIDGE. Conditions were selected in order to steadily progress from lower to higher $RH_{\text{ice}}$ with the respective highest and lowest supersaturations overlapping at each temperature increment.

| $T$ [°C] | $RH_{\text{water}}$ [%] | $RH_{\text{ice}}$ [%] |
|---|---|---|
| -20 | 95 | 115.6 |
| | 97 | 118.0 |
| | 99 | 120.4 |
| | 101 | 122.9 |
| -25 | 95 | 121.3 |
| | 97 | 123.9 |
| | 99 | 126.4 |
| | 101 | 129.0 |
| -30 | 95 | 127.4 |
| | 97 | 130.1 |
| | 99 | 132.7 |
| | 101 | 135.4 |





**Table 3.** Statistical parameters extracted from the INP concentrations measured at the four sites. $N_{sig}$ is the number of samples that had concentrations above the significance level, which was set to twice the background INP concentration. $N_{valid}$ is the number of valid measurements at this condition, i.e. measurements were non-zero and above the detection limit. Such measurements typically have an uncertainty of RH $\pm 2\%$. Repeated measurements revealed typical uncertainties in the INP concentration on the order of $\pm 30\%$.

| | $-20\,°C$, $101\%$ | | | | $-25\,°C$, $101\%$ | | | | $-30\,°C$, $101\%$ | | | |
| --- | --- | --- | --- | --- | --- | --- | --- | --- | --- | --- | --- | --- |
| | SB | TO | MQ | AZ | SB | TO | MQ | AZ | SB | TO | MQ | AZ |
| Median [L$^{-1}$] | 0.06 | 0.17 | 0.11 | 0.10 | 0.19 | 0.39 | 0.44 | 0.31 | 0.69 | 1.09 | 1.80 | 2.06 |
| Arith. Mean [L$^{-1}$] | 0.12 | 0.22 | 0.20 | 0.17 | 0.29 | 0.55 | 0.86 | 0.55 | 1.15 | 2.01 | 4.09 | 3.07 |
| Geo. Mean [L$^{-1}$] | 0.07 | 0.14 | 0.11 | 0.10 | 0.17 | 0.32 | 0.38 | 0.30 | 0.64 | 0.99 | 1.79 | 2.03 |
| Std. Dev. [L$^{-1}$] | 0.20 | 0.20 | 0.23 | 0.19 | 0.34 | 0.52 | 1.22 | 0.78 | 1.27 | 2.39 | 5.35 | 3.01 |
| $N_{valid}$ | 186 | 342 | 184 | 110 | 182 | 307 | 164 | 120 | 115 | 194 | 130 | 107 |
| $N_{sig}$ / $N_{valid}$ [%] | 60 | 86 | 77 | 75 | 73 | 83 | 82 | 84 | 78 | 84 | 88 | 97 |
| $N_{valid}$ / $N_{total}$ [%] | 54 | 86 | 65 | 60 | 53 | 77 | 58 | 66 | 33 | 49 | 46 | 59 |



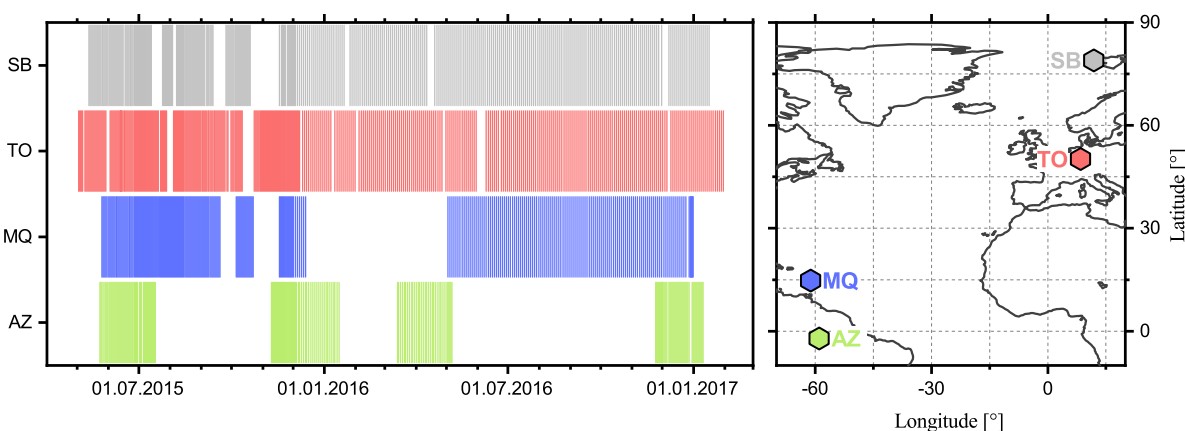

**Figure 1.** Sampling days at the four stations (left) and their corresponding geographic location (right). Line thickness indicates the sampling frequency (thick connected lines: daily; thin non-connected lines: 1 sample every two days).

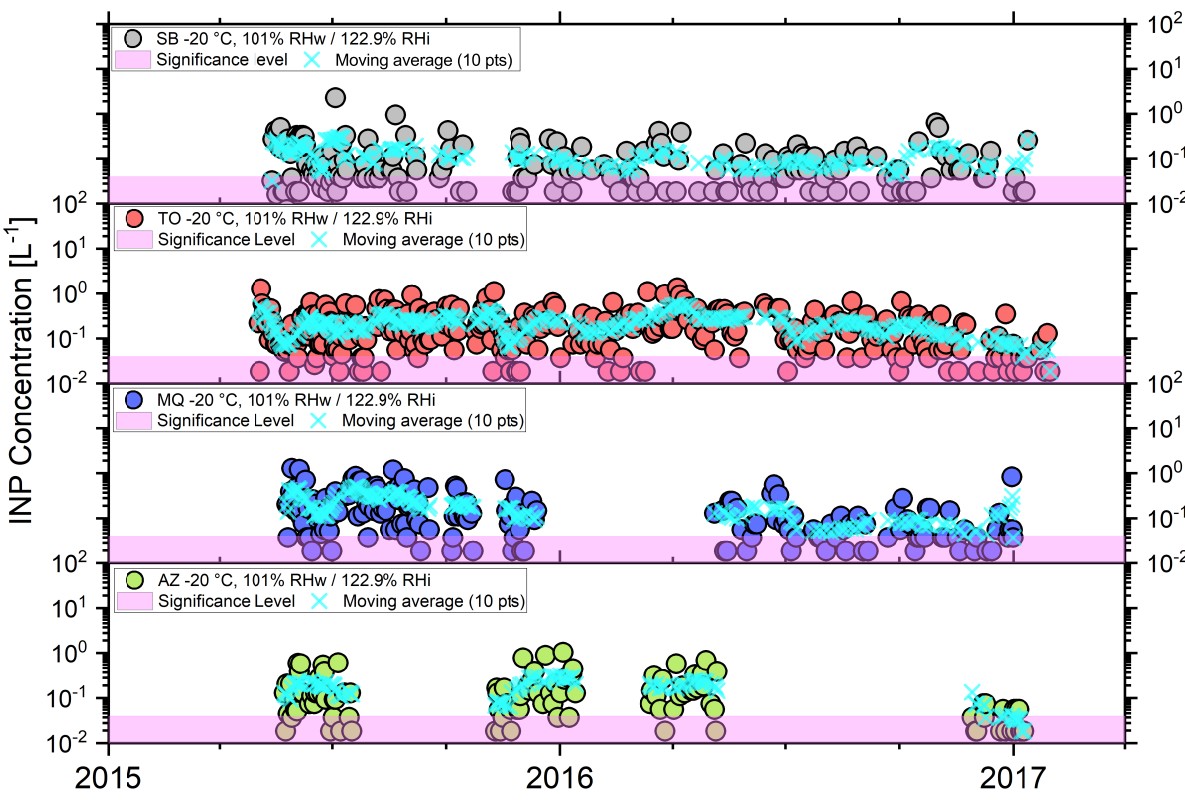

**Figure 2.** INP concentration at $T = -20\,°C$ and $RH_{water} = 101\,\%$, which is $RH_{ice} = 122.9\,\%$. The significance level is indicated in pink shading. Cyan crosses are the result of a 10 point moving average for INP concentrations. X-axis ticks are shown for January 1st, April 1st, July 1st and October 1st.

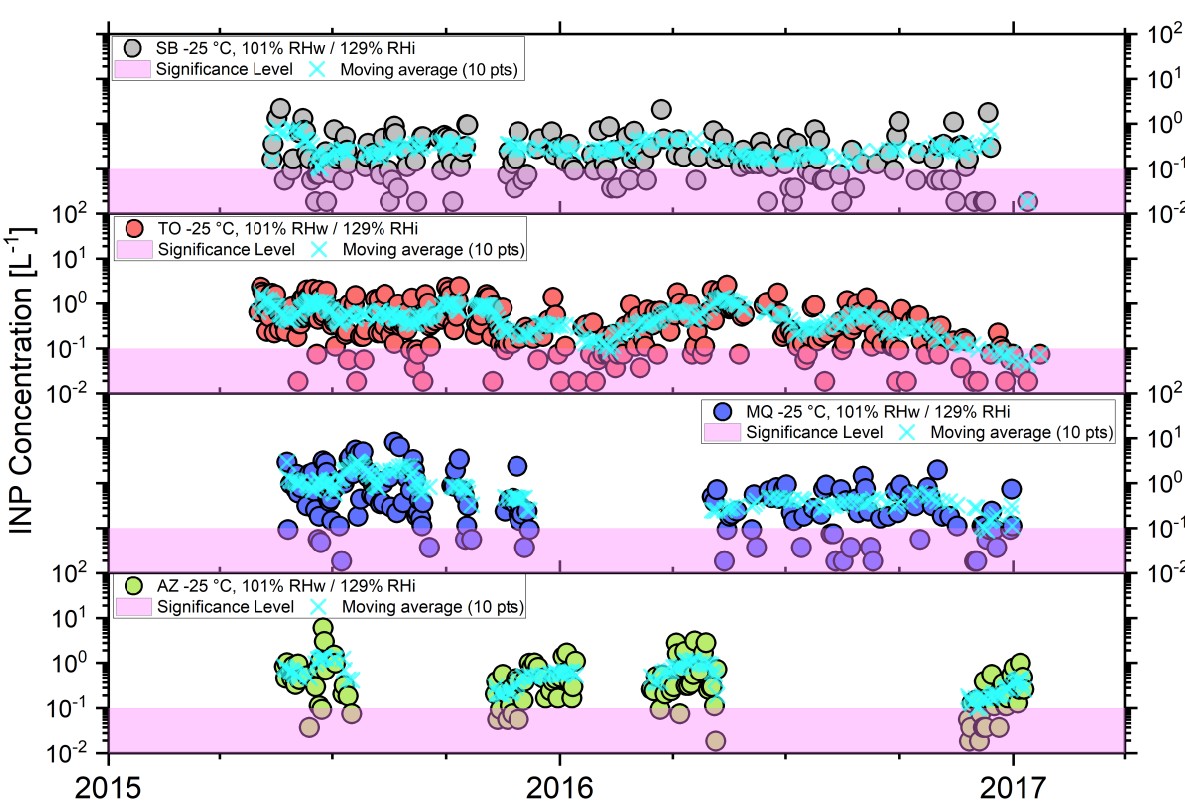

**Figure 3.** INP concentration at $T = -25\,^{\circ}\mathrm{C}$ and $RH_{water} = 101\,\%$, which is $RH_{ice} = 129\,\%$. The significance level is indicated in pink shading. Cyan crosses are the result of a 10 point moving average for INP concentrations. X-axis ticks are shown for January 1st, April 1st, July 1st and October 1st.





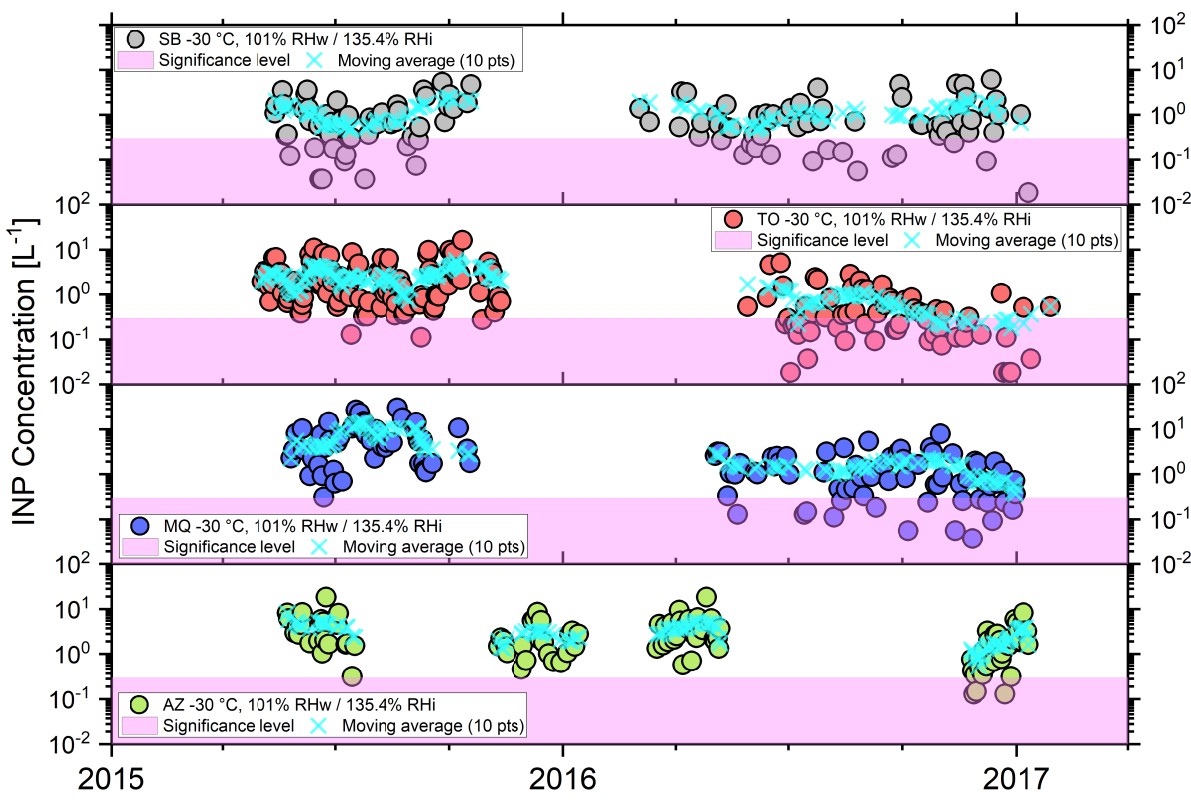

**Figure 4.** INP concentration at $T = -30\,^\circ\text{C}$ and $RH_{water} = 101\,\%$, which is $RH_{ice} = 135.4\,\%$. The significance level is indicated in pink shading. Cyan crosses are the result of a 10 point moving average for INP concentrations. X-axis ticks are shown for January 1st, April 1st, July 1st and October 1st.

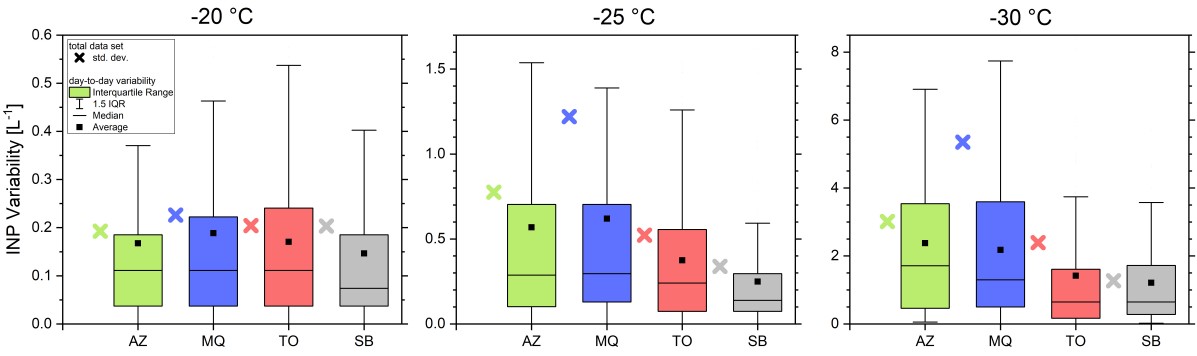

**Figure 5.** INP variability for $T = -20\,^\circ\text{C}$ (left), $T = -25\,^\circ\text{C}$ (middle) and $T = -30\,^\circ\text{C}$ (right) at $RH_{water} = 101\,\%$. Crosses show the standard deviation of the total data set for each site. Box-plots show the distribution of sample-to-sample differences in the INP concentration of consecutive samples (i.e. day-to-day or every other day).




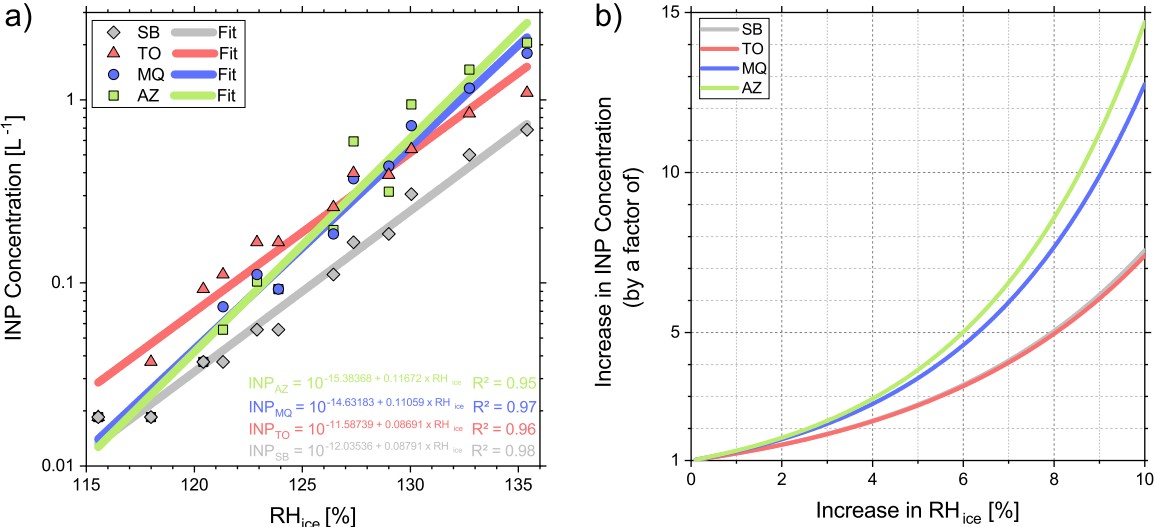

**Figure 6.** a) Median INP concentrations at the four sites measured at T= $-20\,°C$, $-25\,°C$ and $-30\,°C$ plotted as a function of $RH_{ice}$ according to Table 2. b) The sensitivity of INP concentrations to increasing ice supersaturation based on the fits shown in a). Please note that the gray SB curve is largely superimposed by the red curve.

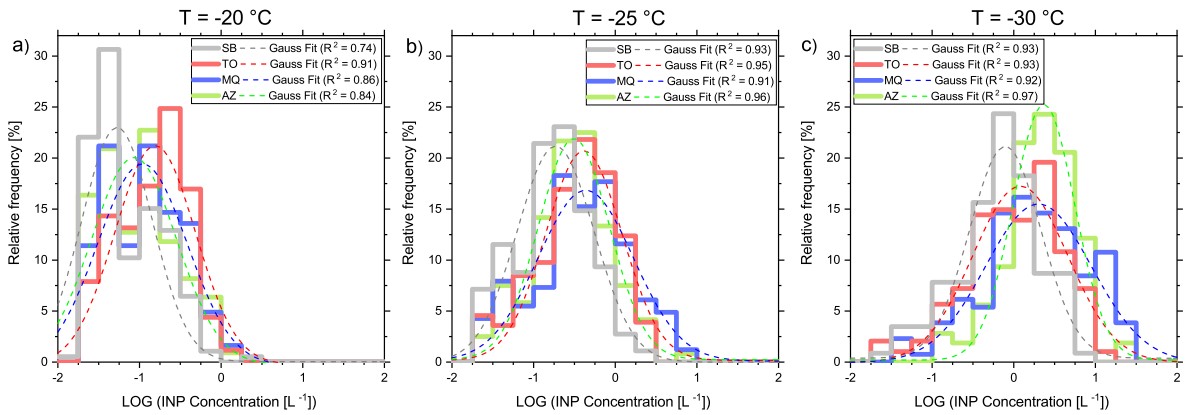

**Figure 7.** Probability density distribution plots of the INP concentrations at $RH_{water} = 101\,\%$ and $-20\,°C$ (a), $-25\,°C$ (b) and $-30\,°C$ (c).

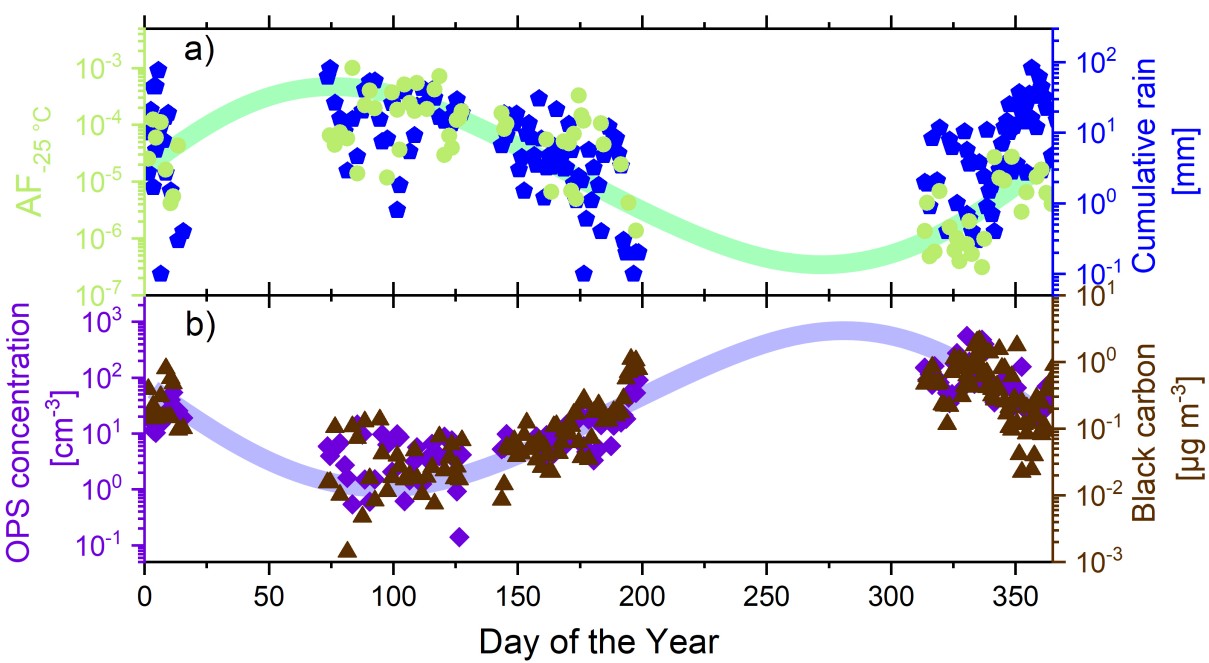

**Figure 8.** Seasonal variation of the activated fraction at $T = -25\,^{\circ}\mathrm{C}$ and $RH_{water} = 101\,\%$ at AZ and the cumulative precipitation along the trajectory reaching the site at the time of sampling (both a). Aerosol number concentration retrieved from a co-located OPC and the black carbon mass concentration (both b) at AZ. Lines are added to guide the eye.

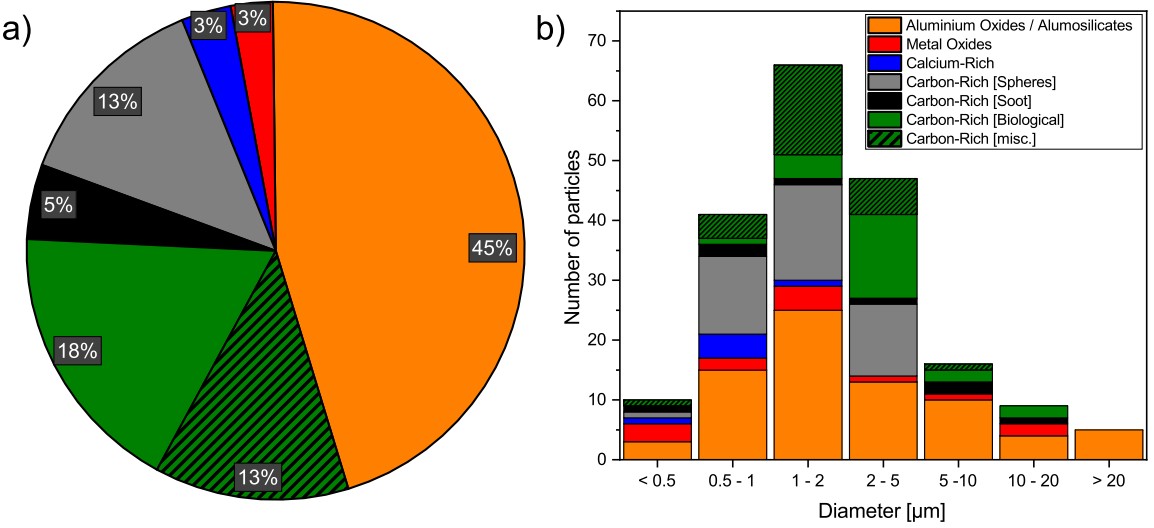

**Figure 9.** a) Average INP composition of six equally weighted AZ samples measured by electron microscopy (N = 196). b) Size distribution of identified INPs. Particles labeled as Carbon-Rich [Spheres] show distinct features of tar balls and are likely products from biomass burning.

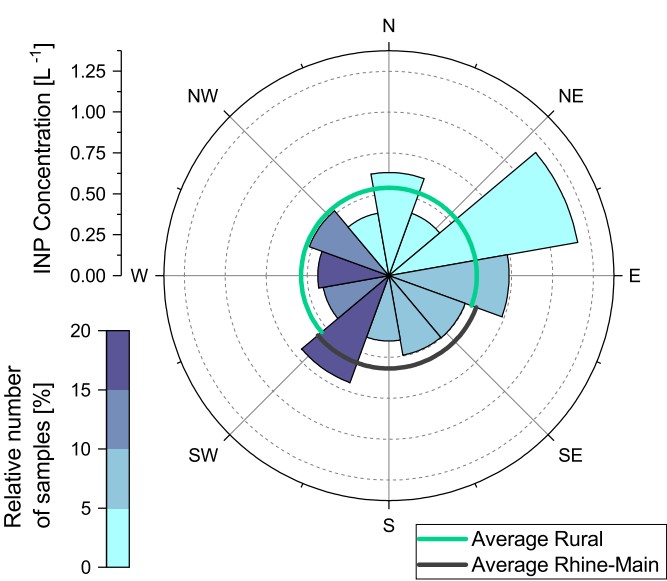

**Figure 10.** Average INP concentration at $T = -25\,^\circ\mathrm{C}$ and $RH_{water} = 101\,\%$ at TO depending on local wind direction. Note that the wind rose is divided into sectors to match the distinction between the sectors of the metropolitan (black line average) and rural (green line average) area according to Tab. 1. The relative frequency of local wind directions during sampling is indicated by the color-coding.