# Peer review of "Long-term INP measurements from four stations across the globe"

_Atmospheric Chemistry and Physics, 2020_

## Short Comment (SC1) · 29 Aug 2020

This is an important global view of INPs over space and over time that will be the basis on which generalities about INP abundance and distribution will be defended in future works. In this light, I think that it is critical that the temperatures at which INP abundances were evaluated be mentioned in the abstract. While reading the abstract I asked myself "is this true for INPs at warm temperatures (warmer than -10°C) that are likely to be of biological origin?" As I can see from the contents of the paper, it does not address this question. I suppose that the numerous sources of "dust" and the massive amounts that are in the atmosphere contribute to the ability for them to be mixed up throughout the atmosphere all over the planet leading to a sort of homogenization. I would have been surprised (and disappointed) if this were the case for

warmer-temperature INPs (but so be it, if the authors had indeed made that observation).

---

## Author Comment (AC1) · 29 Aug 2020

Dear Cindy Morris,

thank you for the short comment.

Unfortunately, we cannot say anything about INPs at temperatures as warm as -10 °C or warmer, as we measured only in the temperature range between -20 °C to -30 °C. We see now that we missed to clarify this in the abstract. We will mention the temperature range in the abstract of the manusrcipt's next iteration.

---

## Referee Comment (RC1) · Anonymous Referee #1 · 30 Aug 2020

Review of "Long-term INP measurements from four stations across the globe" by Schrod et al.

General comments: The authors made enormous amount of efforts tackling the current challenges of the INP research community – wide spatiotemporal coverage of ambient INP measurements. This reviewer is impressed with a comprehensiveness of this work (for 1212 samples) as well as persistence and articulation of the authors, and supports publication of this manuscript in ACP. The results and discussions provided in this manuscript tightly fit in the scope of ACP. The reviewer has only technical (some are minor) suggestions to make (see below). But, the reviewer noticed different writing styles/tones involved over different sections (before/after Sect. 2.3.). Consistency in writing will improve the readability as well as importance of this paper even more.

[Figure]

Specific and technical comments:

P1L4: –> Unfortunately, only a few . . .

P1L14-15: This statement introduces a multitude of perspectives – one may consider physicochemical properties have negligible impact on INP abundance/propensity, thereby ambient INP estimation could be rather simple than 'complex'. This may be true and somehow supported by what the authors found (i.e., P1L9-11 & P1L18-19; great statements, by the way). Perhaps, incorporating this counter-thought (on top of what already exists) in an abstract and other parts in the main text would increase the readability/flexibility to both authors and readers.

P2L6-8: Depending on. . . - the reviewer Is not sure if this statement is adding any meaningful aspects in this paper. The authors may consider removing this statement. The CCN is not discussed in tandem with INP much in this manuscript.

P2L9: Non-biological organics are deemed to be overlooked here. The authors may review Knopf et al. & Kanji et al.?

P2L19: Vertical distribution – very good point. This is somehow one of the things INP community has been missing for a long time in the reviewer's opinion. This should be pointed out in the outlook section?

P2L21: . . .in identifying globally relevant INP. . . –> . . .in identifying some or potentially atmospheric-related INP. . .

P3L1: The reviewer totally agrees with this statement. This statement is a nice complement to previous studies. Nice writing. P3L2-17: Perhaps summarizing the examined temperature and the n_INP ranges from these previous studies in a tabular format with minimum explanation instead of prolonged texts would increase the readability of this section.

P3L20-21: Yes. This is a very good motivation statement. Good job.

P4L6-8: Seeing long/lat coordinates for these locations in their first appearances would be nice. The reviewer is aware these coordinates appear later on. This is just a suggestion from the reader's perspective. The authors can decide what to do.

P4L11: Please clarify what "semi-automated" really means. Please also clarify how the samples were stored while transporting here. Frozen at a certain temperature all the way? The reviewer is aware that the authors mention an insignificance of storage method on their INP characterization in P5L18-19. Perhaps, transportation and storage discussion can be combined here or P5?

P4L17: factors could include local dynamics, thermodynamics, large scale meteorology, and/or a combination of any?

P4L26-27: So is this correction incorporated/applied in relevant INP # in this study? Please state it if so.

P4L27-: It would be meaningful to have a discussion of all inlets configuration and properties (e.g., length, flow rate – if any, cut-size – if an impactor was used in part, transmission efficiency, transmitted aerosol particle size range etc.) from individual sites here (rather than in Sect. 2.3). Maybe, the authors can use a table summarizing the inlet config., characterization (if done/any). Also, listing previous INP research done at the sites would be meaningful info for the readers.

P5L2: "for use within an INP monitoring network" seems misleading – sounds like a strong promotion. The reviewer suggests altering this to –> to collect aerosol particles at multiple field sites for subsequent offline INP analysis. This way, the tone would be reduced, and the point can be made for the concurrent work.

P5L12-13: Please elaborate the difficulties a bit further.

P5L13: Representativeness of local noon & short sampling time is questionable (the reviewer is aware that the discussion is given later on). On the other hand, the reviewer supports the best practice of pursuing consistency with this strategy employed by the

authors for this study. Perhaps, such should be mentioned here to justify the strategy. The readers will understand.

P6 Sect. 2.3.: Very informative and detailed. But, this section seemingly better fits as SI in the reviewer's opinion. Especially, P6L20-P7L10 & P7L23-P8L20 seem not relevant to the main focus of this study. Putting a subset in SI at the least would even increase the readability – the reviewer's suggestion is based on the readers' perspective.

P10L15: Delete (incomplete).

P10L10-11, 16-12, and 27-29: The reviewer is impressed with these statements. Congratulations on finding these.

P11L1: Besides storage effects, inconsistency in inlet configurations and IN mechanisms can also play a role in the reviewer's opinion. If a proper inlet is not used for aerosol particles sampling, sampling efficiency of the sampler could be affected by local turbulence and other dynamic/thermodynamic conditions (e.g., sampler port get frozen/clogged). These points should be incorporated, otherwise the readers might be misled.

P11L7-8: Add reference(s) for bio-INPs that the authors are mentioning here or elaborate it.

P11L9-16: So what is the implication of such a strong IS dependence? Are the authors trying to point out the condensation/droplet freezing is more predominant as compared to deposition?

P11L19-22: This part is speculative. The reviewer sees lots of "may" words. But, it does justify that the sentence can remain speculative. Please introduce some references/citations to support the authors' idea at the least.

P11L28-29: Yes. The reviewer agrees.

P12L8: That said, –> However (too informal for a scientific journal).

P12L8-9: The source of INPs is important, but how aerosol particles are sampled at the sampling location through what sort of inlets is also an important source of potential data variation. See the reviewer's comment regarding an inlet above.

P12L16: which one is bimodal? Please clarify this in the text.

P12L19-19: distribution analysis with higher sensitivity at high Ts would be a good future work (may be incorporated in depth in an outlook section?).

P12L27-29: This sentence is running too long, diluting an important message. The reviewer suggest breaking it down and carefully reformulate this sentence.

P13L9-10: background air masses mean local ambient T and RH etc.? The authors may want to add "More discussion of insignificant role of local sources is provided in the next section" or something similar to smoothly guide the readers to e.g., P13L31.

P14L1-13: Though the reviewer finds this part (bio aerosol - INP - precipitation interactions) very interesting, some parts sound speculative simply due to the lack of sufficient data – e.g., rain intensity, wind/gust condition, rain duration etc. etc. What is discussed in this sub-section seems supplementary, not the main point of this study. The reviewer suggests either elaborate it rigorously or eliminate it completely.

P14L18: likely –> presumably

P14L24: Then, the local source seems important. . . This seems contradicting to the point made in P13L9-10. Please clarify.

P14L29-31: Very good statement.

P16L5: Given –> Due to

P16L5: . . .atmosphere, the Arctic. . . (comma)

P16L13-27: The authors may consider mentioning about a more recent study by Rinaldi et al. (2020 - https://acp.copernicus.org/preprints/acp-2020-605/). The reviewer

believes that findings of Rinaldi et al. (Ny-Alesund, Gruvebadet station through a semi-laminar flow TSP inlet during 2018) are consistent with what is presented in this study (2015-2017). Another place to potentiall add Rinaldi et al. is on P17L11 in addition to Welti et al. (2020).

P16L32: –> . . . anthropogenic Arctic Haze phenomenon during our study period. The reviewer supports the authors' view, but the authors may want to reduce the tone. Otherwise, it may sound like a personal attack even without an intention. Just a suggestion to be fair on everyone in our community.

P17L4-9 & P17L22: Very good summary – the reviewer's additional hope is a consistency in an inlet sampling system.

P18L17-18: The reviewer disagrees. The finer time resolution of INP measurements for prolonged period of time with a reasonable detection - perhaps by semi-autonomous technique as mentioned towards the end of this section by the authors - is an ultimate goal/outlook for ambient INP measurements in the reviewer's opinion. With a long(er) sampling time, researchers would overlook subtle change in INP episodes or local dynamic condition that has certain roles on INP propensity.

There are quite more important things to be listed as more specific future study ideas out of this study (e.g., inlet consistency, P2L19, P12L19-19 etc.). These could be addressed in this section.

Other general outlook can be made, but the authors may look through Murray et al. (2020 -https://acp.copernicus.org/preprints/acp-2020-852/), and adapt the authors' ideas on top? Just a suggestion.

P19L8: Möhler et al. may become publicly available soon. The authors may keep an eye on it, or touch base with Dr. Möhler.

The reviewer enjoyed reading this paper. Hope some of suggestions/comments made here help the authors (and future readers).

---

## Referee Comment (RC2) · Paul DeMott (Referee) · 1 Sep 2020

**General Comments**

This paper is excellent as a large compilation of INP data that has been processed in a consistent manner. The effort is to be commended for that reason alone. It is also a very well written manuscript, and with most of the details one would wish for, and the abstract highlights several key points: well mixed populations that do not vary greatly overall between northern and southern continental and marine sites, short-term variability dominating at all sites, certain site specific aerosol drivers of INPs, but no universal driving aerosol property driver, and no indication of anthropogenic influences. Nevertheless, as I read the paper as it is currently organized, I struggled in

knowing how to relate the method and results from the standard FRIDGE method to drop freezing assays (or the immersion mode method sometimes applied using the FRIDGE device), which are possibly the most widely used present method. It seems to me that two things are required to assist readers in understanding the nature of the results, and potentially how to consider them in relation to immersion freezing data. First, the title should explicitly describe the basis for INP measurements. In other words, "Long-term deposition/condensation freezing INP measurements..." or something to that effect. When one sees the INP versus ice supersaturation data in this manuscript, there is no discontinuity that occurs at water saturation (as the authors readily note), and so it seems apparent that immersion mode freezing is indeed not represented at all. The authors provide a discussion of the dominant mechanisms at play in the data and the likely underestimate in comparison to immersion freezing mode operation of the FRIDGE only very late in the paper. This is critically important in understanding if the findings can be ascribed only to deposition and condensation-freezing mode INPs, or if the same is expected for immersion freezing populations. I suggest in the specific comments that the methods used may indeed limit assessment of strong local/regional impacts, at least for biomass burning. Of course, it will not be possible to make a conclusion about what was not measured, but it should be highlighted as a question for future inspection. This should all be made crystal clear. Hence, the second recommended change is to bring a discussion forward of what types of INPs the data describe, and what types the generalized results may not describe. It will not detract from the great effort the authors have made to collect large quantities of ice nucleation data from multiple sites and discern answers to some of the key and enduring questions related to INP sources. However, I believe that it will better frame future needs.

**Specific Comments**

1) Introduction

Page 2, Lines 8-9: Is there a reason to separate primary biological aerosol and marine biological aerosols? They are both primary biological aerosols, no? If referring to

secondary marine aerosols, you might require evidence that those play any role as INPs.

Page 2, Lines 30-32: I find this statement quickly becoming untrue, with many laboratories now involved in long-term measurements of immersion freezing (e.g., Schneider et al., 2020), some with agency support, and multiple online instruments are in development (or are already there) for automated or semi-automated operation.

Page 3, line 30 to end of paragraph: With regard to anthropogenic influences, I do think that there is some literature on this topic. Some is recent, e.g., Levin et al. (2019) found no apparent influence of urban pollution on INPs in studies in CA, USA. Chen et al. (2019) and Bi et al. (2019) discuss urban pollution impacts in Beijing.

2) Methods

Page 4, lines 28-29: Have larger particle losses been quantified? This is important, as it is a weakness compared to an open-faced filter for example, and it is not clear as an advantage over the in situ instruments mentioned in the last sentence of the paragraph. For example, Schrod et al. (2016) report collection efficiencies only to 3 microns, which is not measurably much different that impactors used on some in situ devices. And larger particles might be imagined as the most efficient deposition nuclei. While collection of and a role for larger INPs is evident ultimately in Fig. 9 for the AZ site, one wonders if the drop off of INPs at sizes above 2 microns reflects the true contributions in these size classes or is influenced at all by collection efficiencies.

Page 5, line 13 paragraph: This description of the aerosol samples had me already wondering about sampler inlets and placement. You might state that this will be covered for each specific site. I do question the statement that 100 L samples provide for "well-resolved ice crystal numbers for a broad spectrum of temperatures..." INP concentrations can span several orders of magnitude from -5 to $-35°C$ . This paper covers a 10C range for data presentation. Finally, is the statement on storage effects necessarily assured for biological INPs that might be exposed to dessicated and higher

temperature conditions? This was qualified in Schrod et al. (2016).

Section 2.2: It is worth carefully explaining the valid activation modes for this work (should be deposition and condensation "freezing" mode on line 22), perhaps by reiterating a few points from Schrod et al. (2016). This first paragraph appears to be the clear place to expound on what is known about the potential underestimations compared to immersion freezing mode INP data as well. Instead, there is only a sentence, "In this context...", which is awkward and defensive considering that the FRIDGE instrument pre-dated many of the droplet freezing assays. The instrument is clearly a tool within the wider array of ice nucleation instrument types, and to my knowledge one of the few well-characterized and documented ones that allows for exploring the full temperature and ice relative humidity space (in the mixed-phase cloud regime) for single samples, in the same manner that droplet freezing assays allow for full temperature spectra. All of the advantages of the technique compared to more labor intensive diffusion chambers and drop freezing assays are well acknowledged. What is missing for this assessment of long-term records at multiple sites is a clear indication of the relation of the modes assessed to immersion freezing. What is known and what remains for future exploration, if the method could be meshed with additional immersion freezing measures?

Page 5, line 24: The word meaningful seems unnecessary.

Page 7, line 15: An additional question here is if there are any considered additional particle losses in the inlet entry to the sampling system. That is, is sampling from the main inlet isokinetic (or sub- or super-isokinetic) and are any additional large particle losses characterized for that last step in collection? Similarly on page 8, line 23, it says that the sampler and the OPS instruments were connected to a 2 m stainless steel line at OVSM. Were particle transmission efficiencies characterized/expected to be the same at this site? Given the outsized role of larger particles as INPs at some surface sites (e.g., Mason et al., 2016), it seems important to know if the relative collection efficiencies were the same, and what the upper limit might be. I also note no mention

of sampling inlet protocol for either TO or SB sites.

Page 7, line 29: A minor note here that it would be interesting to know the vegetative differences in these sites. Images of the sampling sites could also be interesting, for supplemental information.

Page 9, line 20: What is meant by "direct influence of sea salt aerosol"? Is the Zeppelin site not within the boundary layer? This is important to know with regard to what influences are being measured there. Sea spray particles would seem as one key source

3) Results and Discussion

Page 10, lines 10-11: Considering the discussion above about INP mechanisms, this statement about deposition being considered relatively unimportant for mixed phase clouds is confusing. Is this not what is measured by the FRIDGE instrument? If the traces of INP versus ice supersaturation are continuous, how to know the difference between deposition and condensation freezing? Is not the highest RH value of processing used here so that the highest INP concentrations assessable are accessed? This is the only way to understand the following statement that "incomplete" condensation freezing is assessed. Again, this may be material to consolidate in the Methods section, where it can be pointed out that an emphasis will be placed on the highest RH values for inter-comparison of site data.

Page 10, lines 31-32: It is great that the authors qualify the results regarding timing of the sampling, storage impacts, etc. However, I am not sure what this statement means about long-term trends being better captured by different sampling strategies. Can you expound? Does it mean spreading the sampling periods out across daily periods? Larger volume samples collected over longer time periods? Additional use of immersion freezing methods, as in that study, to investigate if that mode of ice nucleation also shows a lack of long-term trends at sites. Also, please note that the full publication on the noted results is now in press and under review in ACP (Schneider et al., 2020).

That study does show trends linked to a regional source. One can imagine that regions close to mineral dust sources also show impacts of a strong regional source, where much higher INP concentrations are noted (e.g., Price et al., 2018). Likewise, higher latitude and polar regions, especially from ship campaigns in the Southern Hemisphere (McCluskey et al., 2018; Welti et al., 2020), appear to represent extraordinarily pristine INP environments. It is simply the case that for the sites selected for this paper and the methods applied, strong cycles are not noted and short-term variability dominated. The extent to which this can be generalized for tropical and mid-latitude regions remains to be seen.

Page 11, lines 15-16 and elsewhere: I have a suggestion to consider for demonstrating the spectral differences between sites, and where they are distinguished for given sites. Currently, a temperature spectral plot is not included in the paper, with too much emphasis on ice supersaturation in my opinion. Figure 5 could be made differently or augmented with an additional panel. While sometimes a linear scale is preferable, in this case if you alternately (or additionally) put these data on the same log scale, one could see the temperature differences more clearly. For example, if the y-axis scaled from 0.01 to 10 on a log scale, the temperature spectra becomes evident for conditions near water saturation, which are arguably the most important for clouds.

Page 12, lines 18-19: This comment harps back a little bit to the statement in Methods regarding the large dynamic range of measurements. While 100 L samples are more useful than the smaller sample volumes used in online instruments, the lack of resolution in the $-20°$C and warmer regime means that there is little or no access to the temperature range where one might expect most sites to be distinguished, considering for example the results shown in Petters and Wright (2015). This is also an important point to remember in the discussion here regarding whether any of the sites are distinguished by apparent biological particle influences. The measurements are just touching the regime of interest.

Page 13, discussion of Fig. 7: Figure 7 is a remarkable figure, and I find it astonishing

that local sources do not come into play for either TO or AZ. I wonder if the authors might comment on whether INP removal is also a factor to consider, not only dilution/mixing out from strong sources, as is inferred in the comment about "background" air masses?

Page 13, section 3.2.1: First, can you please clarify the timing of the "dry season" at AZ? It becomes obvious in Fig. 8, but it would be nice to see it stated in the discussion. And then one has to go back to figures to note the lack of an apparent influence of smoke. The reduction is AF is not really unexpected, right, in consideration of previous results regarding biomass burning INPs? Considering laboratory studies of surrogate and real combustion particles (Petters et al., 2009; Levin et al., 2016; Kanji et al., 2020) and field studies (Prenni et al, 2012; McCluskey et al., 2014; Schill et al., 2020)? Hence, the discussion could be clarified here, including the most recent references. One might even support that for realistic combustion particles, and not only black carbon isolated (Kanji et al., 2020) or contained in real biomass burning particles (Schill et al., 2020), water supersaturations and immersion freezing are required to see the influence of biomass burning on INP concentrations (e.g., Petters et al., 2009; Schill et al., 2020). That is, there are clear impacts of biomass burning on regional INP concentrations already demonstrated in the literature for other regions. I think this discussion needs more specifics than referencing a review paper and a single laboratory study on black carbon surrogates. Activity within the deposition and condensation freezing regime up to water saturation may be quite limited, so this may represent a case where the methods applied in this paper cannot resolve real influences on INPs, or it may indicate that fires are not sources at AZ. I think it is unresolved still.

Page 14, lines 17-18: Following in the same line of comment, in fact the INP concentration results herein seem to be a factor of several lower compared to Prenni et al. (2009). It would be good to quantify what is stated presently as "on the low end".

Page 15, lines 25-26: It is unclear if the conclusion here is that marine contributions to the INPs at MQ are represented in the lower range of values observed?

Page 15, lines 32-33: Is this correlation with PM10 at TO shown anywhere? Can you at least state the $r^2$ and p values?

Page 16, SB section: As I read this section, I wondered about the issues brought forward at the end of the section with regard to signal to noise ratio, and how this influenced the lack of a seasonal cycle. For example, Hartmann et al. (2020) should also be referenced here. They also report winter values consistent with Tobo et al. (2019) and Wex et al. (2019). Hence, one wonders why no seasonal cycle is present in the data here. Is it just noise, or is the baseline potentially somehow even higher than you have estimated from blank data?

4) Conclusions

Page 17, lines 11-13: I find alluding to the Welti et al. paper results to not be a great comparison. In fact differences in the most remote locations were striking compared to mid-latitude and tropical locations in that paper and in other recent ship campaigns (McCluskey et al., 2018).

Page 17, line 20: I think you should add "at all sites" when referring to the inability of single parameters to describe results. This is important, as influences were noted at some sites.

Page 18, lines 3-6: This is the discussion point that needs to be introduced earlier in the paper, as I mentioned previously. One even wonders if the processing conditions emphasize certain INP types that are more well mixed in the atmosphere and contain few hygroscopic materials that would limit ice nucleation until strong condensation occurs at most of the temperatures investigated.

5) Outlook

Page 18, lines 18-19: One wonders about varying sampling times over daily schedules to represent diurnal cycles. However, here, I wonder if it is necessarily true that longer sampling times would reduce short term variability? Would several hour samples reflect

less differences than the short sample times used in this study? How do you know?

Page 18, lines 21-22: Again I find myself disagreeing with this conclusion that automated and higher frequency sampling methods are too much of a technological challenge. It simply needs impetus and being made a priority, and I would judge that the time has already arrived.

Page 18, lines 30-32: A reason that immersion freezing is considered so important is because clouds, and how they form, in many cases determine this result. Could immersion freezing measurements become an integral part of sampling and processing protocol for a device like the FRIDGE? Then all mechanisms except contact freezing would be assessed.

Page 19, lines 9-10: I find the calling out of a single device to be inappropriate here, from a conference paper no less. Fortunately for this reference, the prime publication on the PINE came out the same day as this review (Möhler et al., 2020). However, automated CFDC instruments are already being built for surface sites (Bi et al., 2019) and under development for aircraft use. I do not understand the statement about a "vital intermediate step".

**References**

Bi, K., G. R. McMeeking, D. Ding, E. J. T. Levin, P. J. DeMott, D. Zhao, F. Wang, Q. Liu, P. Tian, X. Ma, Y. Chen, M. Huang, H. Zhang, T. Gordon, and P. Chen, 2019: Measurements of ice nucleating particles in Beijing, China. Journal of Geophysical Research: Atmospheres, 124, 8065–8075, https://doi.org/10.1029/2019JD030609.

Chen, J., Wu, Z., Augustin-Bauditz, S., Grawe, S., Hartmann, M., Pei, X., Liu, Z., Ji, D., and Wex, H.: Ice-nucleating particle concentrations unaffected by urban air pollution in Beijing, China, Atmos. Chem. Phys., 18, 3523–3539, https://doi.org/10.5194/acp-18-3523-2018, 2018.

Hartmann, M., Blunier, T., Brügger, S. O., Schmale, J., Schwikowski, M., Vogel, A., et

al. (2019). Variation of ice nucleating particles in the European Arctic over the last centuries. Geophys. Res. Lett., 46, 4007– 4016. https://doi.org/10.1029/2019GL082311

Levin, E. J. T., P. J DeMott, K. J Suski, Y. Boose, T. C. J. Hill, C. S. McCluskey, G. P Schill, K. Rocci, H. Al-Mashat, L. J. Kristensen, G. C. Cornwell, K. A. Prather, J. M. Tomlinson, F. Mei, J. Hubbe, M. S. Pekour, R. J. Sullivan, L. R. Leung and S. M. Kreidenweis, 2019: Characteristics of ice nucleating particles in and around California winter storms, Journal of Geophysical Research: Atmospheres, 124, 11,530-11,551, https://doi.org/10.1029/2019JD030831.

Levin, E. J. T., G. R. McMeeking, P. J. DeMott, C. S. McCluskey, C. M. Carrico, S. Nakao, C. E. Stockwell, R. J. Yokelson, and S. M. Kreidenweis, 2016: Ice-nucleating particle emissions from biomass combustion and the potential importance of soot aerosol, J. Geophys. Res. Atmos., 121 (10), 5888-5903, doi:10.1002/2016JD024879.

Mason, R. H., Si, M., Chou, C., Irish, V. E., Dickie, R., Elizondo, P., Wong, R., Brintnell, M., Elsasser, M., Lassar, W. M., Pierce, K. M., Leaitch, W. R., MacDonald, A. M., Platt, A., Toom-Sauntry, D., Sarda-Estève, R., Schiller, C. L., Suski, K. J., Hill, T. C. J., Abbatt, J. P. D., Huffman, J. A., DeMott, P. J., and Bertram, A. K.: Size-resolved measurements of ice-nucleating particles at six locations in North America and one in Europe, Atmos. Chem. Phys., 16, 1637–1651, https://doi.org/10.5194/acp-16-1637-2016, 2016.

McCluskey, C. S., T. C. J. Hill, R. S. Humphries, A. M. Rauker, A. M., S. Moreau, S., P. G. Strutton, S. D. Chambers, A. G. Williams, I. McRobert , J. Ward, M. D. Keywood, J. Harnwell, W. Ponsonby, Z.M. Loh , P. B. Krummel, A. Protat , S.M. Kreidenweis, and P. J. DeMott, 2018: Observations of ice nucleating particles over Southern Ocean waters. Geophysical Research Letters, 45, 11,989–11,997. https://doi.org/10.1029/2018GL079981.

McCluskey, C. S., P. J. DeMott, A. J. Prenni, E. J. T. Levin, G. R. McMeeking, A. P. Sullivan, T. C. J. Hill, S. Nakao, C. M. Carrico, and S. M. Kreidenweis (2014), Characteristics of atmospheric ice nucleating particles associated with biomass burning in the

US: Prescribed burns and wildfires, J. Geophys. Res. Atmos., 119, 10,458–10,470, doi:10.1002/ 2014JD021980.

Möhler, O., Adams, M., Lacher, L., Vogel, F., Nadolny, J., Ullrich, R., Boffo, C., Pfeuffer, T., Hobl, A., Weiß, M., Vepuri, H. S. K., Hiranuma, N., and Murray, B. J.: The portable ice nucleation experiment PINE: a new online instrument for laboratory studies and automated long-term field observations of ice-nucleating particles, Atmos. Meas. Tech. Discuss., https://doi.org/10.5194/amt-2020-307, in review, 2020.

Petters, M. D., and T. P. Wright (2015), Revisiting ice nucleation from precipitation samples, Geophysical Research Letters, 42(20), 8758-8766, doi:10.1002/2015gl065733.

Petters, M. D., M. T. Parsons, A. J. Prenni, P. J. DeMott, S. M. Kreidenweis, C. M. Carrico, A. P. Sullivan, G. R. McMeeking, E. Levin, C. E. Wold, J. L. Collett, Jr., and H. Moosmüller, 2009: Ice nuclei emissions from biomass burning. J. Geophys. Res., 114, D07209, doi: 10.1029/2008JD011532.

Prenni, A. J., P. J. DeMott, , A. P. Sullivan, R. C. Sullivan, S. M. Kreidenweis, and D. C. Rogers, 2012: Biomass burning as a potential source for atmospheric ice nuclei: Western wildfires and prescribed burns. Geophys. Res. Lett., 39, L11805, doi:10.1029/2012GL051915.

Price, H. C., Baustian, K. J., McQuaid, J. B., Blyth, A., Bower, K. N., Choularton, T., Cotton, R. J., Cui, Z., Field, P. R., Gallagher, M., Hawker, R., Merrington, A. Miltenberger, A., Neely III, R. R., Parker, S. T., Rosenberg, P. D., Taylor, J. W., Trembath, J., Vergara‐Temprado, J., Whale, T. F., Wilson, T. W., Young, G., and Murray, B. J. (2018). Atmospheric ice‐nucleating particles in the dusty tropical Atlantic. J. Geophys Res. – Atmos, 123, 2175-2193 https://doi.org/10.1002/2017JD027560

Schill, G. P., P. J. DeMott, E. W. Emerson, A. M. C. Rauker, J. K. Kodros, K. J. Suski, T. C. J. Hill, E. J. T. Levin, J. R. Pierce, D. K. Farmer, and S. M. Kreidenweis. The contribution of black carbon to global ice nucleating particle concentrations relevant to mixedphase clouds. Proceedings of the National Academy of Sciences, 2020; 202001674 DOI: 10.1073/pnas.2001674117

Schneider, J., Höhler, K., Heikkilä, P., Keskinen, J., Bertozzi, B., Bogert, P., Schorr, T., Umo, N. S., Vogel, F., Brasseur, Z., Wu, Y., Hakala, S., Duplissy, J., Moisseev, D., Kulmala, M., Adams, M. P., Murray, B. J., Korhonen, K., Hao, L., Thomson, E. S., Castarède, D., Leisner, T., Petäjä, T., and Möhler, O.: The seasonal cycle of ice-nucleating particles linked to the abundance of biogenic aerosol in boreal forests, Atmos. Chem. Phys. Discuss., https://doi.org/10.5194/acp-2020-683, in review, 2020.

———————————————————————

---

## Author Response (AR1)

**Response to Anonymous Referee #1**

First of all, we thank the referee for submitting helpful and productive comments and annotations, which have led to improvements and clarifications within the revised manuscript we submit with this review response.

We have prepared a revised manuscript that addresses the questions and comments of all referees. Furthermore, below we explicitly respond to each of the items raised in the comments of anonymous referee #1. These comments are indicated in *italics,* whereas the author's response is presented in blue. Changes in the manuscript are given in green; changes to the supplement are given in purple. A response with "Okay." means we accept the reviewers' suggestion and have implemented it within the revised manuscript. The differences are also highlighted in separate PDFs using latexdiff. All line and page numbers refer to the ACPD manuscript (version 2), not the revised manuscript.
* * *
*Review of "Long-term INP measurements from four stations across the globe" by Schrod et al.*

*General comments:*

*The authors made enormous amount of efforts tackling the current challenges of the INP research community – wide spatiotemporal coverage of ambient INP measurements. This reviewer is impressed with a comprehensiveness of this work (for 1212 samples) as well as persistence and articulation of the authors, and supports publication of this manuscript in ACP. The results and discussions provided in this manuscript tightly fit in the scope of ACP. The reviewer has only technical (some are minor) suggestions to make (see below). But, the reviewer noticed different writing styles/tones involved over different sections (before/after Sect. 2.3.). Consistency in writing will improve the readability as well as importance of this paper even more.*

> We are grateful for the positive feedback of the reviewer. We hope to improve the readability (and substance) of the manuscript by implementing the suggestions of the reviewer.

*Specific and technical comments:*

- *P1L4: → Unfortunately, only a few ...*
  > Okay.

- *P1L14-15: This statement introduces a multitude of perspectives – one may consider physicochemical properties have negligible impact on INP abundance/propensity, thereby ambient INP estimation could be rather simple than 'complex'. This may be true and somehow supported by what the authors found (i.e., P1L9-11 & P1L18-19; great statements, by the way). Perhaps, incorporating this counter-thought (on top of*

*what already exists) in an abstract and other parts in the main text would increase the readability/flexibility to both authors and readers.*

> Admittedly, one could argue as the reviewer proposes here. We argue for a "complex" and unresolved interplay of factors determining the INP concentration as we did not find an individual parameter (i.e. particle number concentration $> 0.5\mu m$, $PM_{10}$, etc.), which managed to predict the number of INPs to a satisfyingly high degree at any one site let alone all sites. Hence, we think that the observed high short-term variability is a clear sign that we do not sufficiently know all processes involved, or at least that the supporting physical and chemical parameters at hand did not cover all relevant aspects of the ice nucleation process.

- *P2L6-8: Depending on ... - the reviewer is not sure if this statement is adding any meaningful aspects in this paper. The authors may consider removing this statement. The CCN is not discussed in tandem with INP much in this manuscript.*

> Okay.

- *P2L9: Non-biological organics are deemed to be overlooked here. The authors may review Knopf et al. & Kanji et al.?*

> We now include non-biological organics in the list. The sentence now reads:
> "Known species of INPs include mineral dust, soil dust, primary biological aerosol particles of terrestrial and marine origin, as well as organics and glassy aerosols (Kanji et al., 2017)."

- *P2L19: Vertical distribution – very good point. This is somehow one of the things INP community has been missing for a long time in the reviewer's opinion. This should be pointed out in the outlook section?*

> We agree with the referee that vertical distributions of INP need to be explored more by the community, as INP concentrations at heights where clouds form may differ significantly from those at ground level (e.g. Schrod et al., 2017). We added a paragraph to the outlook (see later).

- *P2L21: ...in identifying globally relevant INP... → ...in identifying some or potentially atmospheric-related INP...*

> We changed the phrasing to:
> "...in identifying some of the INP species of global relevance..."

- *P3L1: The reviewer totally agrees with this statement. This statement is a nice complement to previous studies. Nice writing.*

> Thank you.

- *P3L2-17: Perhaps summarizing the examined temperature and the n_INP ranges from these previous studies in a tabular format with minimum explanation instead of prolonged texts would increase the readability of this section.*

> We understand the argument for a better readability, yet we feel that the paragraph and the entailed efforts made in the early years of ice nucleation research deserve some space in a manuscript, highlighting the importance of long-term INP measurements.

- *P3L20-21: Yes. This is a very good motivation statement. Good job.*
    Thank you.

- *P4L6-8: Seeing long/lat coordinates for these locations in their first appearances would be nice. The reviewer is aware these coordinates appear later on. This is just a suggestion from the reader's perspective. The authors can decide what to do.*
    We have added the coordinates within the introduction as well.

- *P4L11: Please clarify what "semi-automated" really means. Please also clarify how the samples were stored while transporting here. Frozen at a certain temperature all the way? The reviewer is aware that the authors mention an insignificance of storage method on their INP characterization in P5L18-19. Perhaps, transportation and storage discussion can be combined here or P5?*
    We think there is no need to add a very detailed description of these sampling related specifics here in the introduction. Much of these questions are answered (i.e. "semi-automated" sampling) within the following section (see section 2.2 Aerosol sampling). We will, however, add to this section, addressing the items raised by the referee. P5L19:
    "Since a frozen storage and transport could not be logistically guaranteed for all sites and for all times, samples were stored and transported at ambient temperatures, which may have affected the warm end of (biological) INPs."

- *P4L17: factors could include local dynamics, thermodynamics, large scale meteorology, and/or a combination of any?*
    All these factors surely influence ice nucleation in the atmosphere, yet we largely did not consider these in the analysis as we feel they are outside the scope of this manuscript. When formulating this sentence we were mainly thinking about aerosol species and sources.

- *P4L26-27: So is this correction incorporated/applied in relevant INP # in this study? Please state it if so.*
    Yes, it is. The sentence now reads:
    "The PEAC7 collection efficiency has been found to be about 60%, independent of particle size (Schrod et al., 2016). Accordingly, a correction factor of 0.6 has been applied to the data."

- *P4L27-: It would be meaningful to have a discussion of all inlets configuration and properties (e.g., length, flow rate – if any, cut-size – if an impactor was used in part, transmission efficiency, transmitted aerosol particle size range etc.) from individual sites here (rather than in Sect. 2.3). Maybe, the authors can use a table summarizing the inlet config. characterization (if done/any). Also, listing previous INP research done at the sites would be meaningful info for the readers.*
    We recognize from both reviews that more care should have been taken when describing the inlet configurations. Unfortunately, only the particle losses at AZ have been quantitatively characterized (Moran-Zuloaga et al., 2018, see section 2.4.1). Regrettably, we don't think a thorough inlet characterization is feasible at this point as the sampling devices are no longer at the sampling sites. We will add a paragraph that mentions this shortcoming more clearly. P4L27:

"No inlet size-cutoffs were used for the results presented here, and thus we expect to sample the complete particle spectrum, except for the usual particle losses that may occur for large particle sizes. The exact aerosol inlet configuration differed substantially between sites and was mainly predetermined by the local observatory facilities. Unfortunately, these inconsistencies may lead to some aerosol sampling artifacts with respect to the absolute particle losses. The individual sampling configurations are described in section 2.4 and Tab. 1."

**Table 1.** Main characteristics of the geographic sampling location and inlet configuration at the sites.

| | AZ | MQ | TO | SB |
|---|---|---|---|---|
| Geograph. coordinates | 2.144° S, 59.000° W | 14.735° N, 61.147° W | 50.221° N, 8.446° E | 78.908° N, 11.881° E |
| Altitude [m AMSL] | 130 | 487 | 825 | 474 |
| Climate | tropical | (sub-)tropical | temperate | Arctic |
| Continental / marine | continental | marine | continental | marine |
| Mountain site | no | yes | yes | yes |
| Predominant vegetation | tropical rainforest | diverse (i.e. ranging from alpine to tropical rainforest) | coniferous forest | low-growing tundra (summer) / snow-covered (winter) |
| Anthropogenic impact | near pristine to polluted | remote to polluted | rural to polluted | near pristine to polluted |
| Inlet type | Total Suspended Particulate (Moran-Zuloaga et al., 2018) | 1/4" tube, rain shield (no characterized inlet) | HORIBA ASS-370 type (ÖNORM, 2007) | Whole-air (Karlsson et al., 2020) |
| Inlet height [m AGL] | 60 | 2 | 11 | 7.5 |
| Isokinetic flow splitter | yes | no | yes | yes |
| Length of tubings to PEAC7 [m] | 1.5 | 2 | 1 | 2 |

We added to P9L14 (TO):
"Samples were collected from the upper level of Atmospheric Physics Laboratory at the hilltop. The aerosol inlet was at 11 m above ground. A main flow of ambient air was pumped through a Horiba ASS-370 type inlet (ÖNORM, 2007) with a 40 mm I.D. x 7 m length stainless steel tube into the laboratory. The PEAC7 collected aerosol isokinetically at 2 l min$^{-1}$ from the main flow through a nozzle of 2.2 mm diameter."

We added to P9L23 (SB):
"A whole air inlet was used for aerosol particle sampling according to the ACTRIS guideline for stations that are often embedded in clouds. The flow through the inlet was kept constant to ensure near isokinetic sampling conditions. A short description about the inlet characteristics of the Zeppelin Observatory can be found in Karlsson et al. (2020)."

As for previous INP research at the sites: This part is discussed in the respective sections 3.2.1, 3.2.2, 3.2.3 and 3.2.4 when available.

- *P5L2: "for use within an INP monitoring network" seems misleading – sounds like a strong promotion. The reviewer suggests altering this to → to collect aerosol particles*

*at multiple field sites for subsequent offline INP analysis. This way, the tone would be reduced, and the point can be made for the concurrent work.*

Okay.

- *P5L12-13: Please elaborate the difficulties a bit further.*

  The sample substrates need to be thoroughly cleaned before use as contaminant particles may introduce significant background freezing. This is a problem observed especially at lower temperatures (i.e. ≤ -30 °C). During the stated time frame we struggled to meet the workload associated with the cleaning procedure and INP analysis. As a result we were only able to guarantee clean (low background) substrates for temperatures ≥ -25 °C. The manuscript now reads: "Between October 2015 and February 2016 some unexplained contamination in the process of wafer cleaning prevented to clean substrates to below the desired background level of INP at the lowest temperature. As a consequence no data below -25°C are available for this period."

- *P5L13: Representativeness of local noon & short sampling time is questionable (the reviewer is aware that the discussion is given later on). On the other hand, the reviewer supports the best practice of pursuing consistency with this strategy employed by the authors for this study. Perhaps, such should be mentioned here to justify the strategy. The readers will understand.*

  We added the following to the description of the sampling strategy:
  "However, the level of representativeness of the deployed sampling strategy is difficult to assess (see discussion). Yet, the pursued sampling protocol ensured a consistent data base."

- *P6 Sect. 2.3.: Very informative and detailed. But, this section seemingly better fits as SI in the reviewer's opinion. Especially, P6L20-P7L10 & P7L23-P8L20 seem not relevant to the main focus of this study. Putting a subset in SI at the least would even increase the readability – the reviewer's suggestion is based on the readers' perspective.*

  We understand that the site descriptions are unusually long in comparison to other studies, but we feel it is important to this manuscript to emphasize the contrasting features of the measurement stations by including a rather thorough characterization here. Especially, as one of the main findings emerging from this study is that the deposition INPs do not seem to differ all that much from site to site, despite these differences.

- *P10L15: Delete (incomplete).*

  Okay. Note that we try to address the matter of nucleation mechanisms addressed more clearly in the revised manuscript in response to the feedback of reviewer 2.

- *P10L10-11, 16-12, and 27-29: The reviewer is impressed with these statements. Congratulations on finding these.*

  Thank you.

- *P11L1: Besides storage effects, inconsistency in inlet configurations and IN mechanisms can also play a role in the reviewer's opinion. If a proper inlet is not used for aerosol particles sampling, sampling efficiency of the sampler could be affected by local*

*turbulence and other dynamic/thermodynamic conditions (e.g., sampler port get frozen/clogged). These points should be incorporated, otherwise the readers might be misled.*

> We added the following to P11L1:
> "Furthermore, differences between the inlet configurations of the individual sites may have influenced the particle sampling process (see section 2.2)."

- *P11L7-8: Add reference(s) for bio-INPs that the authors are mentioning here or elaborate it.*

> We have added to this text passage:
> "For example, O'Sullivan et al. (2018) found that immersion INP concentrations at -20 °C at a northwestern European site were reduced by more than a factor of 2 in 59 % of the cases when samples were heated to 100 °C. For warmer temperatures the reduction was found to be significantly higher."

- *P11L9-16: So what is the implication of such a strong IS dependence? Are the authors trying to point out the condensation/droplet freezing is more predominant as compared to deposition?*

> As reviewer 2 (Paul DeMott) points out, an INP activation spectrum that is not dependent on temperature implies that immersion mode INPs were not represented in the data. We will try to address this point more clearly in the revised manuscript (see the revised text in section 2.1 and 2.3.1, and the responses to reviewer 2).

- *P11L19-22: This part is speculative. The reviewer sees lots of "may" words. But, it does justify that the sentence can remain speculative. Please introduce some references/ citations to support the authors' idea at the least.*

> We agree that the part is speculative and we believe the chosen phrasing makes this clear to the reader. Section 2.4 lists some references that indicate that long-range transport of mineral dust is a regular feature of the AZ and MQ sites.
> We added a sentence (P11L22):
> "Our view of a generally higher abundance of mineral dust at the low latitude sites MQ and AZ as compared to the high latitudes of SB and TO is supported by dust observations from surface stations (Prospero et al., 1996), remote sensing (Kaufmann et al., 2005) and models (Zender et al., 2003; Lee et al., 2009)."

- *P11L28-29: Yes. The reviewer agrees.*

> Good.

- *P12L8: That said, –> However (too informal for a scientific journal).*

> Okay.

- *P12L8-9: The source of INPs is important, but how aerosol particles are sampled at the sampling location through what sort of inlets is also an important source of potential data variation. See the reviewer's comment regarding an inlet above.*

> We agree. See previous responses.

- *P12L16: which one is bimodal? Please clarify this in the text.*

  There is no clear indication of a bimodal frequency distribution, yet, as stated, some of the data hint at it (e.g. SB).

- *P12L19-19: distribution analysis with higher sensitivity at high Ts would be a good future work (may be incorporated in depth in an outlook section?).*

  We agree with the reviewer that future works could focus on the frequency distribution of the INP concentrations for warm temperatures. Furthermore, we would like to point the reviewer to a publication by Welti et al. (2018) that shows a similar figure for INP measurements in the subtropical marine boundary layer at temperatures up to -8 °C. We have added a paragraph to the outlook (see later).

- *P12L27-29: This sentence is running too long, diluting an important message. The reviewer suggest breaking it down and carefully reformulate this sentence.*

  We rephrased the sentence:

  "Therefore, local species of plants or bacteria may be less likely to have evolved traits that induce freezing. It has previously been posited that some microbiology (e.g. bacteria like *Pseudomonas syringae*) gain an evolutionary advantage by being able to induce freezing (Morris et al., 2014)."

- *P13L9-10: background air masses mean local ambient T and RH etc.? The authors may want to add "More discussion of insignificant role of local sources is provided in the next section" or something similar to smoothly guide the readers to e.g., P13L31.*

  We are not sure what is meant by reviewer's first comment. We believe that the data supports the idea that the measured deposition INP concentrations are largely determined by large-scale background air mass movements. The ambient conditions (T and RH) define if and how many INPs will be activated to ice crystals.

  We have added a short sentence to guide the reader to section 3.2:

  "More discussion of the site specific local sources and characteristic features is provided in the following section."

- *P14L1-13: Though the reviewer finds this part (bio aerosol - INP - precipitation interactions) very interesting, some parts sound speculative simply due to the lack of sufficient data – e.g., rain intensity, wind/gust condition, rain duration etc. etc. What is discussed in this sub-section seems supplementary, not the main point of this study. The reviewer suggests either elaborate it rigorously or eliminate it completely.*

  We believe that although we cannot present sufficient evidence for the importance of biology-precipitation interactions in our data, the discussion would lack a potentially substantial INP feedback for AZ, if we completely removed the discussion. We have shortened some of the text passages. The manuscript now reads:

  "[...] Another way to interpret the anti-correlation of AF and biomass burning markers is by coupling the metric to precipitation rates. There are several intricate interactions of note here. On one hand more precipitation leads to higher aerosol particle (and INP) removal by wet deposition. Moreover, enhanced precipitation during the wet season can largely prevent wild fires and the accompanied particle emissions in the first place. On the other hand, it has

been postulated previously that precipitation may be a driver of biological INPs (Huffman et al., 2013), and large tropical rainforests like the Amazon have been highlighted in that regard (Morris et al., 2014). However, the processes responsible for the release of the biological particles have not yet been deciphered in detail."

- *P14L18: likely → presumably*
  Okay.

- *P14L24: Then, the local source seems important… This seems contradicting to the point made in P13L9-10. Please clarify.*
  On one hand we observed that during a distinct LRT episode INP concentrations were significantly correlated to mineral dust particles. However, even when no clear dust transport was registered (e.g. by back-trajectory analysis and particle measurements), electron microscopy analysis of six samples indicates that mineral dust is responsible for about half of the INPs at AZ. Therefore, we argue that there seems to be a well-mixed and diluted background concentration of mineral dust INPs at all times present at AZ. We have rephrased the text to make our argument more clear:
  "However, mineral dust may be a relevant INP in this region even in the absence of distinct LRT events: An analysis of the average composition of INPs of six samples (4 in April 2016, 2 in December 2016) using scanning electron microscopy (SEM, Figure 9), identified that nearly half of the particles that activated to ice crystals in FRDIGE were mineral dust. This finding suggests that there seems to be a well-mixed and diluted background concentration of mineral dust INPs at all times present at AZ. The diameter of most of the INPs investigated by SEM in this study was between one and a couple of micrometers (Figure 9b)."

- *P14L29-31: Very good statement.*
  Thank you.

- *P16L5: Given → Due to*
  Okay.

- *P16L5: …atmosphere, the Arctic…(comma)*
  Okay.

- *P16L13-27: The authors may consider mentioning about a more recent study by Rinaldi et al. (2020 - https://acp.copernicus.org/preprints/acp-2020-605/). The reviewer believes that findings of Rinaldi et al. (Ny-Alesund, Gruvebadet station through a semilaminar flow TSP inlet during 2018) are consistent with what is presented in this study (2015-2017). Another place to potentially add Rinaldi et al. is on P17L11 in addition to Welti et al. (2020).*
  We thank the referee for the suggested reading. We have added a paragraph:

  P16L23: "However, a recent study by Rinaldi et al. (2020) did not observe a distinct seasonal signal in their INP measurements between -15 °C and -22 °C in the spring and summer of 2018 in Ny-Alesund. Rinaldi et al. (2020) present

INP concentrations from two separate methods, one of which is fairly similar to FRIDGE, addressing the condensation freezing (DFPC) and immersion freezing (WT-CRAFT) modes."

P16L27: "Rinaldi et al. (2020) present evidence that Arctic INP concentrations are influenced by sources of marine biological INPs by providing a spatio-temporal correlation analysis between Chlorophyll-a fields from satellite data and a trajectory model."

- *P16L32: → … anthropogenic Arctic Haze phenomenon during our study period. The reviewer supports the authors' view, but the authors may want to reduce the tone. Otherwise, it may sound like a personal attack even without an intention. Just a suggestion to be fair on everyone in our community.*

    We meant to achieve quite the opposite effect here. In fact, we are rather a little concerned with the quality of our data due to the lack of a seasonal feature, as is frequently reported by others. We had hoped to get this message across by the last sentences of the paragraph (see P16L33 and following).

    We have added the word "concerning" in P16L34:
    "The concerning lack of meaningful correlations and/or seasonal trends may be in part related to a relatively poor signal-to-noise ratio in our SB measurements."

- *P17L4-9 & P17L22: Very good summary – the reviewer's additional hope is a consistency in an inlet sampling system.*

    We now list inconsistencies in the inlet system as one cause of uncertainty in P17L32:
    "However, when using the presented data one should be aware of the substantial limitations of the conceptual aspect of the approach and the uncertainties that are inherent in the aerosol sampling and INP measurements themselves."

- *P18L17-18: The reviewer disagrees. The finer time resolution of INP measurements for prolonged period of time with a reasonable detection - perhaps by semi-autonomous technique as mentioned towards the end of this section by the authors - is an ultimate goal/outlook for ambient INP measurements in the reviewer's opinion. With a long(er) sampling time, researchers would overlook subtle change in INP episodes or local dynamic condition that has certain roles on INP propensity.*

    We agree with the reviewer and point him/her to the very next lines (P18L19 and following.

    *There are quite more important things to be listed as more specific future study ideas out of this study (e.g., inlet consistency, P2L19, P12L19-19 etc.). These could be addressed in this section.*
    *Other general outlook can be made, but the authors may look through Murray et al. (2020 -https://acp.copernicus.org/preprints/acp-2020-852/), and adapt the authors' ideas on top? Just a suggestion.*

    We thank the reviewer once again for sharing this excellent suggested reading. We feel it is beyond the scope of our manuscript even to attempt to fully and satisfyingly include all the listed needs of future INP research as done by Murray et al. (2020). However, we now refer to the paper to direct the interested reader

to the more extensive list. We have added a paragraph to the end of the manuscript:

"In addition to the goal of establishing more long-term global observations of continuous INP concentrations there are certainly other important areas for future research to address. For example, as most measurements are conducted at ground level, we believe there is a need to systematically study the vertical distribution of INPs – for example at heights where INPs are transported over long-ranges and/or where cloud formation occurs. Moreover, more extensive data sets from long-term INP monitoring might shed light on what mechanisms result in the observed log-normal INP frequency distributions (and departures from ideality etc.) as presented here and, for example, by Welti et al. (2018). Murray et al. (2020) has recently enumerated many crucial areas into which future INP research should delve. First and foremost, the authors emphasize the need to accurately implement ice nucleation related cloud-phase interactions in climate models in order to predict future climate scenarios correctly. We gladly refer the interested reader to Murray et al. (2020) for a more extensive list of future ice nucleation related research questions, as is presented in this study."

- *P19L8: Möhler et al. may become publicly available soon. The authors may keep an eye on it, or touch base with Dr. Möhler.*
  We have updated the reference from the Lacher et al. (2019) conference abstract to the newly available Möhler et al. (2020) paper.

- *The reviewer enjoyed reading this paper. Hope some of suggestions/comments made here help the authors (and future readers).*
  We are glad that the reviewer appreciated the manuscript. Again, we thank the reviewer for their valuable suggestions, which will most certainly improve the paper.

Literature

Huffman, J. A., Prenni, A. J., DeMott, P. J., Pöhlker, C., Mason, R. H., Robinson, N. H., Fröhlich-Nowoisky, J., Tobo, Y., Després, V. R., Garcia, E., Gochis, D. J., Harris, E., Müller-Germann, I., Ruzene, C., Schmer, B., Sinha, B., Day, D. A., Andreae, M. O., Jimenez, J. L., Gallagher, M., Kreidenweis, S. M., Bertram, A. K., and Pöschl, U.: High concentrations of biological aerosol particles and ice nuclei during and after rain, Atmos. Chem. Phys., 13, 6151–6164, https://doi.org/10.5194/acp-13-6151-2013, 2013.

Kanji, Z. A., Ladino, L. A., Wex, H., Boose, Y., Burkert-Kohn, M., Cziczo, D. J., and Krämer, M.: Overview of Ice Nucleating Particles, Meteorological Monographs, 58, 1.1–1.33, https://doi.org/10.1175/AMSMONOGRAPHS-D-16-0006.1, 2017.

Karlsson, L., Krejci, R., Koike, M., Ebell, K., and Zieger, P.: The role of nanoparticles in Arctic cloud formation, Atmos. Chem. Phys. Discuss., https://doi.org/10.5194/acp-2020-417, in review, 2020.

Kaufman, Y. J., Koren, I., Remer, L. A., Tanré, D., Ginoux, P., and Fan, S.: Dust transport and deposition observed from the Terra-Moderate Resolution Imaging Spectroradiometer (MODIS) spacecraft over the Atlantic Ocean, J. Geophhys. Res., 110, D10S12, https://doi.org/10.1029/2003JD004436, 2005.

Lacher, L., Vogel, F., Nadolny, J., Adams, M., Murray, B. J., Boffo, C., Pfeuffer, T., and Möhler, O.: The Portable Ice Nucleation Experiment (PINE): A New Instrument for Semiautonomous Measurements of Atmospheric Ice Nucleating Particles, in: 99th AMS Annual Meeting, Phoenix, USA, 6–10 January, 2019.

Lee, Y. H., Chen, K., and Adams, P. J.: Development of a global model of mineral dust aerosol microphysics, Atmos. Chem. Phys., 9, 2441–2458, https://doi.org/10.5194/acp-9-2441-2009, 2009.

Möhler, O., Adams, M., Lacher, L., Vogel, F., Nadolny, J., Ullrich, R., Boffo, C., Pfeuffer, T., Hobl, A.,Weiß, M., Vepuri, H. S. K., Hiranuma, N., and Murray, B. J.: The portable ice nucleation experiment PINE: a new online instrument for laboratory studies and automated long term field observations of ice-nucleating particles, Atmos. Meas. Tech. Discuss., https://doi.org/10.5194/amt-2020-307, in review, 2020.

Moran-Zuloaga, D., Ditas, F., Walter, D., Saturno, J., Brito, J., Carbone, S., Chi, X., Hrabˇe de Angelis, I., Baars, H., Godoi, R. H. M., Heese, B., Holanda, B. A., Lavriˇc, J. V., Martin, S. T., Ming, J., Pöhlker, M. L., Ruckteschler, N., Su, H., Wang, Y., Wang, Q., Wang, Z., Weber, B.,Wolff, S., Artaxo, P., Pöschl, U., Andreae, M. O., and Pöhlker, C.: Long-term study on coarse mode aerosols in the Amazon rain forest with the frequent intrusion of Saharan dust plumes, Atmos. Chem. Phys., 18, 10055–10088, https://doi.org/10.5194/acp-18-10055-2018, 2018.

Morris, C. E., Conen, F., Huffman, J. A., Phillips, V., Pöschl, U., and Sands, D. C.: Bioprecipitation: a feedback cycle linking Earth history, ecosystem dynamics and land use through biological ice nucleators in the atmosphere, Glob. Change Biol., 20, 341–351, https://doi.org/10.1111/gcb.12447, 2014.

Murray, B. J., Carslaw, K. S., and Field, P. R.: Opinion: Cloud-phase climate feedback and the importance of ice-nucleating particles, Atmos. Chem. Phys. Discuss., https://doi.org/10.5194/acp-2020-852, in review, 2020.

ÖNORM M 5852: Standard ÖNORM M 5852:2007, Austrian standards, air analysis – sampling for continuous immission monitoring, Committee 139, 2007.

O'Sullivan, D., Adams, M. P., Tarn, M. D., Harrison, A. D., Vergara-Temprado, J., Porter, G., Holden, M. A., Sanchez-Marroquin, A., Carotenuto, F., Whale, T. F., McQuaid, J. B., Walshaw, R., Hedges, D., Burke, I. T., Cui, Z., and Murray, B. J.: Contributions of biogenic material to the atmospheric ice-nucleating particle population in North Western Europe, Scientific reports, 8(1), 13821, https://doi.org/10.1038/s41598-018-31981-7, 2018.

Prospero, J. M.: Saharan Dust Transport Over the North Atlantic Ocean and Mediterranean: An Overview, In: Guerzoni S., Chester R. (eds) The Impact of Desert Dust Across the Mediterranean, Environmental Science and Technology Library, 11, Springer, Dordrecht, https://doi.org/10.1007/978-94-017-3354-0_130, 1996.

Rinaldi, M., Hiranuma, N., Santachiara, G., Mazzola, M., Mansour, K., Paglione, M., Rodriguez, C. A., Traversi, R., Becagli, S., Cappelletti, D.M., and Belosi, F.: Condensation and immersion freezing Ice Nucleating Particle measurements at Ny-Ålesund (Svalbard) during 2018: evidence of multiple source contribution, Atmos. Chem. Phys. Discuss., https://doi.org/10.5194/acp-2020-605, in review, 2020.

Schrod, J., Danielczok, A.,Weber, D., Ebert, M., Thomson, E. S., and Bingemer, H. G.: Re-evaluating the Frankfurt isothermal static diffusion chamber for ice nucleation, Atmos. Meas. Tech., 9, 1313–1324, https://doi.org/10.5194/amt-9-1313-2016, 2016.

Schrod, J., Weber, D., Drücke, J., Keleshis, C., Pikridas, M., Ebert, M., Cvetkovic, B., Nickovic, S., Marinou, E., Baars, H., Ansmann, A., Vrekoussis, M., Mihalopoulos, N., Sciare, J., Curtius, J., and Bingemer, H. G.: Ice nucleating particles over the Eastern Mediterranean measured by unmanned aircraft systems, Atmos. Chem. Phys., 17, 4817–4835, https://doi.org/10.5194/acp-17-4817-2017, 2017.

Welti, A., Müller, K., Fleming, Z. L., and Stratmann, F.: Concentration and variability of ice nuclei in the subtropical maritime boundary layer, Atmos. Chem. Phys., 18, 5307–5320, https://doi.org/10.5194/acp-18-5307-2018, 2018.

Zender, C. S., Bian, H., and Newman, D.: Mineral Dust Entrainment and Deposition (DEAD) model: Description and 1990s dust climatology, J. Gephys. Res., 108, D14, 4416, https://doi.org/10.1029/2002JD002775, 2003.

**Response to Referee #2 – Paul DeMott**

First of all, we thank Paul DeMott for submitting helpful and productive comments and annotations, which have led to improvements and clarifications within the revised manuscript we submit with this review response.

We have prepared a revised manuscript that addresses the questions and comments of all referees. Furthermore, below we explicitly respond to each of the items raised in the comments of Paul DeMott (reviewer 2). These comments are indicated in *italics,* whereas the author's response is presented in blue. Changes in the manuscript are given in green; changes to the supplement are given in purple. A response with "Okay." means we accept the reviewers' suggestion and have implemented it within the revised manuscript. The differences are also highlighted in separate PDFs using latexdiff. All line and page numbers refer to the ACPD manuscript (version 2), not the revised manuscript.
* * *
*This paper is excellent as a large compilation of INP data that has been processed in a consistent manner. The effort is to be commended for that reason alone. It is also a very well written manuscript, and with most of the details one would wish for, and the abstract highlights several key points: well mixed populations that do not vary greatly overall between northern and southern continental and marine sites, short-term variability dominating at all sites, certain site specific aerosol drivers of INPs, but no universal driving aerosol property driver, and no indication of anthropogenic influences. Nevertheless, as I read the paper as it is currently organized, I struggled in knowing how to relate the method and results from the standard FRIDGE method to drop freezing assays (or the immersion mode method sometimes applied using the FRIDGE device), which are possibly the most widely used present method. It seems to me that two things are required to assist readers in understanding the nature of the results, and potentially how to consider them in relation to immersion freezing data. First, the title should explicitly describe the basis for INP measurements. In other words, "Long-term deposition/condensation freezing INP measurements. . ." or something to that effect. When one sees the INP versus ice supersaturation data in this manuscript, there is no discontinuity that occurs at water saturation (as the authors readily note), and so it seems apparent that immersion mode freezing is indeed not represented at all. The authors provide a discussion of the dominant mechanisms at play in the data and the likely underestimate in comparison to immersion freezing mode operation of the FRIDGE only very late in the paper. This is critically important in understanding if the findings can be ascribed only to deposition and condensation-freezing mode INPs, or if the same is expected for immersion freezing populations. I suggest in the specific comments that the methods used may indeed limit assessment of strong local/regional impacts, at least for biomass burning. Of course, it will not be possible to make a conclusion about what was not measured, but it should be highlighted as a question for future inspection. This should all be made crystal clear. Hence, the second recommended change is to bring a discussion forward of what types of INPs the data describe, and what types the*

*generalized results may not describe. It will not detract from the great effort the authors have made to collect large quantities of ice nucleation data from multiple sites and discern answers to some of the key and enduring questions related to INP sources. However, I believe that it will better frame future needs.*

We thank the referee his helpful feedback and review. After re-reading the paper and the reviews, we recognize now that it may indeed be difficult to understand for the reader what is measured here and what is not. As a matter of fact, we do absolutely think that we need to explore the differences in our own measurement methods, i.e. FRIDGE standard and droplet freezing mode, to a greater extent. As for the suggested implementations, we agree to the proposed changes. Adding the addressed nucleation mode in the title will immediately help the reader orient themselves. Also, we come to the same conclusion as the referee, that the discussion about the nucleation mode appears too late in the paper and can be better introduced in the methods section as the reviewer proposes in the specific comments.

Accordingly, we have changed the title of the manuscript to:
"Long-term deposition/condensation INP measurements from four stations across the globe".

Furthermore, we now introduce both operational modes of the FRIDGE instrument shortly in a new section 2.1, indicating clearly at the beginning of the methods chapter that only the standard mode has been used in this study:
"2.1 FRIDGE operational modes

The FRIDGE instrument was originally introduced by Bundke et al. (2008) and Klein et al. (2010), but was fundamentally reevaluated and updated by Schrod et al. (2016). Since this effort FRIDGE has participated in laboratory intercomparisons (Hiranuma et al., 2015; DeMott et al., 2018; Hiranuma et al., 2019) and field campaigns (Schrod et al., 2017; Thomson et al., 2018; Gute et al., 2019; Marinou et al., 2019). In its original design FRIDGE serves as an isothermal static diffusion chamber for offline analysis of ice nucleation. In this standard operation mode FRIDGE analyzes deposition and condensation freezing INPs on substrates that had been laden with atmospheric aerosol particles by electrostatic precipitation. To avoid confusion, we point out that the FRIDGE instrument can in fact be modified to serve as a cold stage for droplet freezing assay measurements as well, which was, however, not done for the results presented here."

The former sections 2.1 (2.2 in the revised manuscript) and 2.2. (2.3. in the revised manuscript) then follow, describing the typical procedure during sampling and measurements.

Finally, we have added section 2.3.1, which goes into more detail regarding what kind of INPs were actually measured in this study, i.e. deposition/condensation vs. immersion mode and how these might relate to each other:
"2.3.1 Freezing modes

It should be noted that we cannot predict how our deposition/condensation freezing measurements would translate to the immersion freezing mode in a situation given in the atmosphere. Some conclusions may however be drawn from previous parallel

measurements (unpublished) with the FRIDGE diffusion chamber and the FRIDGE droplet freezing assay in different environments during the FIN-03 (Storm Peak Laboratory, SPL, USA, 2015), GLACE (Jungfraujoch, JFJ, Switzerland, 2017) and PICNIC (Puy de Dome, PDD, France, 2018) campaigns. Daily average INP concentrations (i.e. one day sample and one night sample) covered three orders of magnitude at -25°C. When transforming the INP concentrations to log-space, we find that the two operational modes are well-correlated (R = 0.81, N = 44), with the immersion freezing INPs being on average a factor of 10 higher than deposition/condensation INPs. In fact, the INP concentrations measured in the droplet freezing assay were always higher. One may speculate that both species simply covary for the reason of having the same sources and sinks, or that deposition INPs may represent just a subset of immersion INPs, when observed by FRIDGE, or both. We will present the results of this comparison in more detail in a forthcoming publication, in which we will further investigate how exactly the nucleation modes of both methods are connected to each other.

Except when noted otherwise, the discussion presented in section 3 will focus on the highest ice supersaturation(s) $RH_{ice}$ at each of the three examined activation temperatures (Tab. 2). At these highest saturation conditions, at or slightly above water saturation, we observe the highest INP concentrations. We expect the nucleation mechanism to be a mixture of deposition nucleation and condensation freezing. At lower supersaturations we qualitatively observe trends and variability in INPs that are similar, but at lower absolute concentration levels."

*Specific comments:*

1) *Introduction*

- *Page 2, Lines 8-9: Is there a reason to separate primary biological aerosol and marine biological aerosols? They are both primary biological aerosols, no? If referring to secondary marine aerosols, you might require evidence that those play any role as INPs.*

  This was rather unintentional. We have rephrased the sentence for more clarity (we included non-biological organics to the list as suggested by reviewer 1):
  "Known species of INPs include mineral dust, soil dust, primary biological aerosol particles of terrestrial and marine origin, as well as organics and glassy aerosols (Kanji et al., 2017)."

- *Page 2, Lines 30-32: I find this statement quickly becoming untrue, with many laboratories now involved in long-term measurements of immersion freezing (e.g., Schneider et al., 2020), some with agency support, and multiple online instruments are in development (or are already there) for automated or semi-automated operation.*

  We thank the referee for the interesting paper, which was not available at the time of submission. We certainly hope that the assessment of the referee proves to be correct, as we think that having more long-term INP measurements publicly available is a crucial step in understanding the spatio-temporal variation of INP concentrations worldwide. As for the sentences in question: We think that at least for the very recent past the phrasing is correct. We fully stand by the first sentence, stating that very few of the published INP

measurements cover multiple seasons or more. We edited the second sentence, adding the assessment of the reviewer:

"A further obstacle to INP monitoring was that many instruments were previously not suited for sustained, long-term monitoring tasks due to their complex and labor intensive operating principles. However, recent developments in INP instrumentation and a shift in sampling focus may lead to more long-term INP measurements becoming publicly available now and/or in the near future (e.g. Schneider et al., 2020)."

- *Page 3, line 30 to end of paragraph: With regard to anthropogenic influences, I do think that there is some literature on this topic. Some is recent, e.g., Levin et al. (2019) found no apparent influence of urban pollution on INPs in studies in CA, USA. Chen et al. (2019) and Bi et al. (2019) discuss urban pollution impacts in Beijing.*

    While we are aware that some literature exists on this topic, we wanted to emphasize in the paragraph that the anthropogenic influence on the INP abundance and efficiency is far from conclusive at this point. We added a sentence to the paragraph (P3L26):

    "Although some recent studies indicate that urban pollution aerosol do not make efficient INPs (e.g. Chen et al., 2018), the overall anthropogenic impact on the INP concentration is still rather inconclusive (see also Schrod et al., 2020)."

*2) Methods*

- *Page 4, lines 28-29: Have larger particle losses been quantified? This is important, as it is a weakness compared to an open-faced filter for example, and it is not clear as an advantage over the in situ instruments mentioned in the last sentence of the paragraph. For example, Schrod et al. (2016) report collection efficiencies only to 3 microns, which is not measurably much different that impactors used on some in situ devices. And larger particles might be imagined as the most efficient deposition nuclei. While collection of and a role for larger INPs is evident ultimately in Fig. 9 for the AZ site, one wonders if the drop off of INPs at sizes above 2 microns reflects the true contributions in these size classes or is influenced at all by collection efficiencies.*

    We recognize from both reviews that more care should have been taken when describing the inlet configuration. Unfortunately, only the particle losses at AZ have been quantitatively characterized (Moran-Zuloaga et al., 2018, see section 2.4.1). Regrettably, we don't think a thorough inlet characterization is feasible at this point as the sampling devices are no longer at the sampling sites. We will add a paragraph that mentions this shortcoming more clearly. P1L27:

    "No inlet size-cutoffs were used for the results presented here, and thus we expect to sample the complete particle spectrum, except for the usual particle losses that may occur for large particle sizes. The exact aerosol inlet configuration differed substantially between sites and was mainly predetermined by the local observatory facilities. Unfortunately, these inconsistencies may lead to some aerosol sampling artifacts with respect to the absolute particle losses. The individual sampling configurations are described in section 2.4 and Tab. 1."

**Table 1.** Main characteristics of the geographic sampling location and inlet configuration at the sites.

| | AZ | MQ | TO | SB |
|---|---|---|---|---|
| Geograph. coordinates | 2.144° S, 59.000° W | 14.735° N, 61.147° W | 50.221° N, 8.446° E | 78.908° N, 11.881° E |
| Altitude [m AMSL] | 130 | 487 | 825 | 474 |
| Climate | tropical | (sub-)tropical | temperate | Arctic |
| Continental / marine | continental | marine | continental | marine |
| Mountain site | no | yes | yes | yes |
| Predominant vegetation | tropical rainforest | diverse (i.e. ranging from alpine to tropical rainforest) | coniferous forest | low-growing tundra (summer) / snow-covered (winter) |
| Anthropogenic impact | near pristine to polluted | remote to polluted | rural to polluted | near pristine to polluted |
| Inlet type | Total Suspended Particulate (Moran-Zuloaga et al., 2018) | 1/4" tube, rain shield (no characterized inlet) | HORIBA ASS-370 type (ÖNORM, 2007) | Whole-air (Karlsson et al., 2020) |
| Inlet height [m AGL] | 60 | 2 | 11 | 7.5 |
| Isokinetic flow splitter | yes | no | yes | yes |
| Length of tubings to PEAC7 [m] | 1.5 | 2 | 1 | 2 |

As we cannot retrospectively quantify the particle losses reliably for the inlet configurations at the stations we deleted the sentence about a possible advantage of not using size-cutoffs (P4L29-31).

Concerning Figure 9, as already mentioned the inlet configuration has been characterized by Moran-Zuloaga et al., 2018, as presented in the supplementary Fig. S1 of that manuscript. Here it can be seen that transmission efficiency from the inlet was calculated to be between 90 and 100 % at 2 µm. Depending on the particle density the transmission drops for larger particle sizes. For example, at 5 µm the transmission efficiency is still between 80 and 90 % for particle densities around 1 g cm$^{-3}$, but may be as low as about 60% for particle densities of 2 g cm$^{-3}$, e.g. mineral dust or sea salt. Taking these calculations into account, we believe that the presented INP composition vs. size likely represents the true contributions quite well, at least for the bins up to 5 µm. However, the last bins may in fact be influenced by particle losses to an unknown, but non-neglectable, degree. We added a sentence to P14L28:
"Note however, that the contribution to the larger size bins might be potentially underrepresented due to particle losses from the inlet configuration."

- *Page 5, line 13 paragraph: This description of the aerosol samples had me already wondering about sampler inlets and placement. You might state that this will be covered for each specific site. I do question the statement that 100 L samples provide for "well-resolved ice crystal numbers for a broad spectrum of temperatures..." INP concentrations can span several orders of magnitude from -5 to −35 C . This paper covers a 10C range for data presentation. Finally, is the statement on storage effects necessarily assured for biological INPs that might be exposed to dessicated and higher temperature conditions? This was qualified in Schrod et al. (2016).*

First part: see previous response.

Second, yes, although we think a span of 10 °C is still quite good, we agree to rephrase the statement:

"The sampled aerosol particles resulting from this 100 L of air were found to usually generate well-resolved ice crystal numbers in the investigated temperature regime using the FRIDGE analysis system."

Further, we cannot guarantee that biological INPs remained active during storage and transport. We expanded upon the paragraph:
"As a result, several weeks often passed between sample collection and analysis, which may introduce an aging effect. However, in a previous study no effect of storage time on ice nucleation activity was observed within the investigated temperature regime (Schrod et al., 2016). Since a frozen storage and transport could not be logistically guaranteed for all sites and for all times, samples were stored and transported at ambient temperatures, which may have affected the warm end of (biological) INPs."

- *Section 2.2: It is worth carefully explaining the valid activation modes for this work (should be deposition and condensation "freezing" mode on line 22), perhaps by reiterating a few points from Schrod et al. (2016). This first paragraph appears to be the clear place to expound on what is known about the potential underestimations compared to immersion freezing mode INP data as well. Instead, there is only a sentence, "In this context...", which is awkward and defensive considering that the FRIDGE instrument pre-dated many of the droplet freezing assays. The instrument is clearly a tool within the wider array of ice nucleation instrument types, and to my knowledge one of the few well-characterized and documented ones that allows for exploring the full temperature and ice relative humidity space (in the mixed-phase cloud regime) for single samples, in the same manner that droplet freezing assays allow for full temperature spectra. All of the advantages of the technique compared to more labor intensive diffusion chambers and drop freezing assays are well acknowledged. What is missing for this assessment of long-term records at multiple sites is a clear indication of the relation of the modes assessed to immersion freezing. What is known and what remains for future exploration, if the method could be meshed with additional immersion freezing measures?*
    Yes, we agree with the referee. See the response to the general comments section. Furthermore, we have removed the sentence starting with "In this context".

- *Page 5, line 24: The word meaningful seems unnecessary.*
    Okay.

- *Page 7, line 15: An additional question here is if there are any considered additional particle losses in the inlet entry to the sampling system. That is, is sampling from the main inlet isokinetic (or sub- or super-isokinetic) and are any additional large particle losses characterized for that last step in collection? Similarly on page 8, line 23, it says that the sampler and the OPS instruments were connected to a 2 m stainless steel line at OVSM. Were particle transmission efficiencies characterized/expected to be the same at this site? Given the outsized role of larger particles as INPs at some surface sites (e.g., Mason et al., 2016), it seems important to know if the relative collection efficiencies were the same, and what the upper limit might be. I also note no mention of sampling inlet protocol for either TO or SB sites.*
    The inlet sampling configuration of AZ is well-characterized in the supplementary information of Moran-Zuloaga et al., 2018 to which we refer. The

main sampling line is in fact connected to an isokinetic flow splitter that was connected to the PEAC7. The main particle loss mechanism considered is sedimentation of particles >0.5 µm. P7L15:
"An extensive inlet characterization can be found in the supplementary information of Moran-Zuloaga et al., 2018."

We do assume more or less similar particle loss scenarios at the other sites to what is stated in this section, although we, unfortunately, did not characterize the transmission efficiencies at the other sites. Also see previous response.
We now add information about the inlet configurations at TO and SB:

P9L14 (TO):
"Samples were collected from the upper level of Atmospheric Physics Laboratory at the hilltop. The aerosol inlet was at 11 m above ground. A main flow of ambient air was pumped through a Horiba ASS-370 type inlet (ÖNORM, 2007) with a 40 mm I.D. x 7 m length stainless steel tube into the laboratory. The PEAC7 collected aerosol isokinetically at 2 l min$^{-1}$ from the main flow through a nozzle of 2.2 mm diameter."

P9L23 (SB):
"A whole air inlet was used for aerosol particle sampling according to the ACTRIS guideline for stations that are often embedded in clouds. The flow through the inlet was kept constant to ensure near isokinetic sampling conditions. A short description about the inlet characteristics of the Zeppelin Observatory can be found in Karlsson et al. (2020)."

- *Page 7, line 29: A minor note here that it would be interesting to know the vegetative differences in these sites. Images of the sampling sites could also be interesting, for supplemental information.*

  We have added one picture for each site to the supplement (Figs. S1 to S4). Further, we have added Tab. 1, which describes the most relevant site characteristics, including the predominant vegetation.

- *Page 9, line 20: What is meant by "direct influence of sea salt aerosol"? Is the Zeppelin site not within the boundary layer? This is important to know with regard to what influences are being measured there. Sea spray particles would seem as one key source.*

  Yes, sea spray is still expected to be a key aerosol source. Due to the elevated position of the observatory we expect lower absolute sea salt concentrations than what would have be measured at sea level (i.e. the other research stations in Ny-Alesund). The sentence now reads:
  "The mountain top Zeppelin Observatory was chosen for its elevated position, which likely limited the effects of locally produced pollution and of sea spray from the surf zone."

3) *Results and Discussion*

- *Page 10, lines 10-11: Considering the discussion above about INP mechanisms, this statement about deposition being considered relatively unimportant for mixed phase*

*clouds is confusing. Is this not what is measured by the FRIDGE instrument? If the traces of INP versus ice supersaturation are continuous, how to know the difference between deposition and condensation freezing? Is not the highest RH value of processing used here so that the highest INP concentrations assessable are accessed? This is the only way to understand the following statement that "incomplete" condensation freezing is assessed. Again, this may be material to consolidate in the Methods section, where it can be pointed out that an emphasis will be placed on the highest RH values for inter-comparison of site data.*

Yes, the results of the highest RH are shown to give the highest INP concentration. The intention here was to say that these concentrations are the closest we can come to immersion freezing in our instrument. We recognize that the phrasing adds more confusion than it actually helps. Therefore we have removed the sentence and moved the paragraph to the methods section (see response to the general comments).

- *Page 10, lines 31-32: It is great that the authors qualify the results regarding timing of the sampling, storage impacts, etc. However, I am not sure what this statement means about long-term trends being better captured by different sampling strategies. Can you expound? Does it mean spreading the sampling periods out across daily periods? Larger volume samples collected over longer time periods? Additional use of immersion freezing methods, as in that study, to investigate if that mode of ice nucleation also shows a lack of long-term trends at sites. Also, please note that the full publication on the noted results is now in press and under review in ACP (Schneider et al., 2020). That study does show trends linked to a regional source. One can imagine that regions close to mineral dust sources also show impacts of a strong regional source, where much higher INP concentrations are noted (e.g., Price et al., 2018). Likewise, higher latitude and polar regions, especially from ship campaigns in the Southern Hemisphere (McCluskey et al., 2018; Welti et al., 2020), appear to represent extraordinarily pristine INP environments. It is simply the case that for the sites selected for this paper and the methods applied, strong cycles are not noted and short-term variability dominated. The extent to which this can be generalized for tropical and mid-latitude regions remains to be seen.*

Yes, we primarily meant longer sampling periods/larger sampling volumes. For example, Schneider et al. (2020) have used a time resolution of 24 to 144 h at 11 L min$^{-1}$. However, it would be rather difficult to adapt the FRIDGE standard technique to such long sampling times, as the number of resolvable ice crystals on a substrate is limited. All of the suggestions to implement a different sampling strategy are good ideas, i.e. spreading the total sampling time out over short increments of time throughout a day as well as complementing the standard mode with the FRIDGE droplet freezing mode and longer sampling times. The manuscript has been modified to read:

"Comparisons with other recently published data sets suggest that long-term trends may be better captured using different sampling strategies (Schneider et al., 2020). The authors of that study observe a clear seasonal cycle of immersion INPs in a boreal forest using 24 –144 h filter sampling at 11 L min$^{-1}$, which is a much longer sampling period than has been used here. [...]"

We have not intended to overly generalize our results. To clarify we have added the freezing mode in more instances throughout the manuscript when discussing the "INP concentration". For example, P17L10 now reads:
"In spite of the great differences in basically all characteristics that are expected to define the aerosol concentration, composition and source apportionment, we observed fairly similar INP concentrations for all four stations for the methods and sampling strategy applied."

- *Page 11, lines 15-16 and elsewhere: I have a suggestion to consider for demonstrating the spectral differences between sites, and where they are distinguished for given sites. Currently, a temperature spectral plot is not included in the paper, with too much emphasis on ice supersaturation in my opinion. Figure 5 could be made differently or augmented with an additional panel. While sometimes a linear scale is preferable, in this case if you alternately (or additionally) put these data on the same log scale, one could see the temperature differences more clearly. For example, if the y-axis scaled from 0.01 to 10 on a log scale, the temperature spectra becomes evident for conditions near water saturation, which are arguably the most important for clouds.*

The suggested change to Fig. 5 is appreciated. We have changed it accordingly:

[Figure]

**Figure 5.** INP variability for $T = -20\,°C$ (left), $T = -25\,°C$ (middle) and $T = -30\,°C$ (right) at $RH_{water} = 101\,\%$. Crosses show the standard deviation of the total data set for each site. Box-plots show the distribution of sample-to-sample differences in the INP concentration of consecutive samples (i.e. day-to-day or every other day). Lower whiskers are not shown when the lower 1.5 IQR is at zero.

- *Page 12, lines 18-19: This comment harps back a little bit to the statement in Methods regarding the large dynamic range of measurements. While 100 L samples are more useful than the smaller sample volumes used in online instruments, the lack of resolution in the $-20$ C and warmer regime means that there is little or no access to the temperature range where one might expect most sites to be distinguished, considering for example the results shown in Petters and Wright (2015). This is also an important point to remember in the discussion here regarding whether any of the sites are distinguished by apparent biological particle influences. The measurements are just touching the regime of interest.*

Yes, we agree. We have added a comment on P12L24, when mentioning the potential for biological INPs:
"However, there is no strong evidence for such a signal in our data overall, possibly due to the comparably low sampling volume. As a result the temperature range of our measurements overlaps only very little with the regime where biological particles nucleate."

- *Page 13, discussion of Fig. 7: Figure 7 is a remarkable figure, and I find it astonishing that local sources do not come into play for either TO or AZ. I wonder if the authors might comment on whether INP removal is also a factor to consider, not only dilution/ mixing out from strong sources, as is inferred in the comment about "background" air masses?*

    It is possible that INPs are removed by either being activated to ice crystals or by deposition processes. However, it is difficult for us to assess how such effects may affect the distribution of INP concentrations found, and particularly if and how local sources are affected differently.

    Generally speaking, we also found the strong log-normal representations to be surprising. We believe that this is an area that deserves further attention within the community. In particular we would like to understand how skewness and departures from log-normality is affected by source function changes and atmospheric processing, but these questions go beyond what we can address in this manuscript.

- *Page 13, section 3.2.1: First, can you please clarify the timing of the "dry season" at AZ? It becomes obvious in Fig. 8, but it would be nice to see it stated in the discussion. And then one has to go back to figures to note the lack of an apparent influence of smoke. The reduction is AF is not really unexpected, right, in consideration of previous results regarding biomass burning INPs? Considering laboratory studies of surrogate and real combustion particles (Petters et al., 2009; Levin et al., 2016; Kanji et al., 2020) and field studies (Prenni et al, 2012; McCluskey et al., 2014; Schill et al., 2020)? Hence, the discussion could be clarified here, including the most recent references. One might even support that for realistic combustion particles, and not only black carbon isolated (Kanji et al., 2020) or contained in real biomass burning particles (Schill et al., 2020), water supersaturations and immersion freezing are required to see the influence of biomass burning on INP concentrations (e.g., Petters et al., 2009; Schill et al., 2020). That is, there are clear impacts of biomass burning on regional INP concentrations already demonstrated in the literature for other regions. I think this discussion needs more specifics than referencing a review paper and a single laboratory study on black carbon surrogates. Activity within the deposition and condensation freezing regime up to water saturation may be quite limited, so this may represent a case where the methods applied in this paper cannot resolve real influences on INPs, or it may indicate that fires are not sources at AZ. I think it is unresolved still.*

    The timing of the dry season is given in section 2.4.1: Dry season: August to November, wet season: February to May, transition periods in between. We will state this in 3.2.1 again now.

    We agree that the obtained results are not unexpected, as we wanted to indicate by including the Kanji et al. references. We will expand upon this point as the referee suggests, adding more references on this matter. We thank the referee for the suggested papers. P13L31 now reads:

    "The observed anti-correlation seems to suggest that aerosol particles from fires are relatively poor ice nuclei; an observation that agrees with previously published findings (Kanji et al., 2017, 2020). Considering the recent literature consensus regarding biomass burning INPs, these results are not unexpected. Biomass burning INPs have been studied in the laboratory investigating both surrogate and real combustion particles (Petters et al., 2009; Levin et al., 2016; Kanji et al., 2020), and in the field (Prenni et al., 2012; McCluskey et al., 2014;

Schill et al., 2020). Although at least a regional impact of biomass burning on INP abundance is reported, the nucleation temperatures are usually close to the homogeneous freezing limit. Some of these studies suggest that water supersaturation is a requirement for biomass burning aerosol to act as INPs (e.g. Petters et al., 2009; Schill et al., 2020). In this regard, our data may demonstrate the limits of what is explorable with FRIDGE. Either biomass burning aerosol is in fact a poor source for Amazonian INPs in the investigated temperature regime or the method simply cannot represent the freezing behavior of these particles accurately."

- *Page 14, lines 17-18: Following in the same line of comment, in fact the INP concentration results herein seem to be a factor of several lower compared to Prenni et al. (2009). It would be good to quantify what is stated presently as "on the low end".*

  Okay. The line now reads:

  "However, our observed concentrations are clustered at the low end of those presented by Prenni et al. (2009) (i.e. a factor of 5 lower at -30 °C), which is presumably due to the different nucleation modes addressed."

- *Page 15, lines 25-26: It is unclear if the conclusion here is that marine contributions to the INPs at MQ are represented in the lower range of values observed?*

  We removed the unclear sentence. The manuscript now reads:

  "[...] They measured INP concentrations of 0.06 L⁻¹ at -20 °C and 0.3 L⁻¹ at -24 °C, which agrees within a factor of two to our median INP concentrations at -20 °C and -25 °C."

- *Page 15, lines 32-33: Is this correlation with PM10 at TO shown anywhere? Can you at least state the r2 and p values?*

  In the next line we do present the Pearson correlation for -25°C (101%). The R-value is not impressively high, but is significant due to the large number of samples. We have added an indication of the p-value.

- *Page 16, SB section: As I read this section, I wondered about the issues brought forward at the end of the section with regard to signal to noise ratio, and how this influenced the lack of a seasonal cycle. For example, Hartmann et al. (2020) should also be referenced here. They also report winter values consistent with Tobo et al. (2019) and Wex et al. (2019). Hence, one wonders why no seasonal cycle is present in the data here. Is it just noise, or is the baseline potentially somehow even higher than you have estimated from blank data?*

  We have now added a sentence regarding the new measurements by Hartmann et al. (2020), and have added references to the measurements by Rinaldi et al. (2020) who do not find a seasonal shift in Ny-Alesund (suggested by referee 1). P16L23:

  "Very recent measurements from Greenland during March/April 2018 qualitatively agree very well to these concentration levels (Hartmann et al., 2020). However, a recent study by Rinaldi et al. (2020) did not observe a distinct seasonal signal in their INP measurements in the temperature range from -15 °C to -22 °C in the spring and summer of 2018 in Ny-Ålesund. Rinaldi et al. (2020) present INP concentrations from two separate methods, one of which is fairly

similar to FRIDGE, addressing the condensation freezing (DFPC) and immersion freezing (WT-CRAFT) modes."

Unfortunately, it is difficult to add more insight into why we did not observe a seasonal cycle in the SB data. Although some variation in the background concentration is present, we don't think that the baseline is higher overall than we have assumed. Rinaldi et al. (2020) conclude that the discrepancy of their time series with the clear spring-to-summer differences from other observations "likely indicates that the inter-annual variability of meteorological and biogeochemical conditions determining the INP atmospheric concentration over the Arctic is wider and more complex than previously assumed. For sure, the number of observations in the Arctic and their temporal coverage are still too limited to derive general conclusions on the INP concentration trends." In general, we agree with these statements, although it is still possible that we have missed a seasonal shift due to a poorly selected sampling strategy, as stated in the manuscript (see also above).

4) *Conclusions*

- *Page 17, lines 11-13: I find alluding to the Welti et al. paper results to not be a great comparison. In fact differences in the most remote locations were striking compared to mid-latitude and tropical locations in that paper and in other recent ship campaigns (McCluskey et al., 2018).*
  Okay. We have removed the reference.

- *Page 17, line 20: I think you should add "at all sites" when referring to the inability of single parameters to describe results. This is important, as influences were noted at some sites.*
  Okay.

- *Page 18, lines 3-6: This is the discussion point that needs to be introduced earlier in the paper, as I mentioned previously. One even wonders if the processing conditions emphasize certain INP types that are more well mixed in the atmosphere and contain few hygroscopic materials that would limit ice nucleation until strong condensation occurs at most of the temperatures investigated.*
  We agree and have moved the discussion to the methods section as described earlier.
  Second, unfortunately, we did not fully understand the arguments made regarding a potential bias towards more well-mixed INPs in our data, but we don't think that there is clear evidence for this hypothesis present.

5) *Outlook*

- *Page 18, lines 18-19: One wonders about varying sampling times over daily schedules to represent diurnal cycles. However, here, I wonder if it is necessarily true that longer sampling times would reduce short term variability? Would several hour samples reflect less differences than the short sample times used in this study? How do you know?*
  We do not know and this is a bit speculative. We have changed the wording:

"Longer sampling may effectively act as a low-pass filter and thereby reduce the considerable short-term variability in INPs that is observed everywhere."

- *Page 18, lines 21-22: Again I find myself disagreeing with this conclusion that automated and higher frequency sampling methods are too much of a technological challenge. It simply needs impetus and being made a priority, and I would judge that the time has already arrived.*

  We agree with the assessment of the referee and hope that soon such an effort will be made. We have changed the wording:

  "However, such an instrument and/or technique has not been available in the past and will likely present both technological and human resource challenges."

- *Page 18, lines 30-32: A reason that immersion freezing is considered so important is because clouds, and how they form, in many cases determine this result. Could immersion freezing measurements become an integral part of sampling and processing protocol for a device like the FRIDGE? Then all mechanisms except contact freezing would be assessed.*

  The sampling schedule of this manuscript began in 2014, when the immersion mode was not yet implemented. But in future FRIDGE measurements we will study the atmosphere with both the standard mode and the immersion mode setup whenever possible, just as the referee suggests.

- *Page 19, lines 9-10: I find the calling out of a single device to be inappropriate here, from a conference paper no less. Fortunately for this reference, the prime publication on the PINE came out the same day as this review (Möhler et al., 2020). However, automated CFDC instruments are already being built for surface sites (Bi et al., 2019) and under development for aircraft use. I do not understand the statement about a "vital intermediate step".*

  We thank the referee for bringing our attention to this paper (Bi et al., 2019), which we had missed. He makes a good point and we were not attempting to single out any particular device. In fact, since this time yet another instrument has also emerged (Brunner and Kanji, 2020), we reformulate to include both new references:

  "Although we are currently far from the best-case scenario of a (near) continuous automated global network of INP measurements, there are promising new developments (e.g. Bi et al., 2019; Brunner and Kanji, 2020; Möhler et al., 2020) that may provide a vital step towards long-term (semi-)automated measurements of immersion mode INPs in the near future."

Literature

Bi, K., McMeeking, G. R., Ding, D., Levin, E. J. T., DeMott, P. J., Zhao, D.,Wang, F., Liu, Q., Tian, P., Ma, X., Chen, Y., Huang, M., Zhang, H., Gordon, T., and Chen, P.: Measurements of ice nucleating particles in Beijing, China, Journal of Geophysical Research: Atmospheres, 124, 8065–8075, https://doi.org/10.1029/2019JD030609, 2019.

Brunner, C. and Kanji, Z. A.: Continuous online-monitoring of Ice Nucleating Particles: development of the automated Horizontal Ice Nucleation Chamber (HINC-Auto), Atmos. Meas. Tech. Discuss., https://doi.org/10.5194/amt-2020-306, in review, 2020.

Chen, J., Wu, Z., Augustin-Bauditz, S., Grawe, S., Hartmann, M., Pei, X., Liu, Z., Ji, D., and Wex, H.: Ice-nucleating particle concentrations unaffected by urban air pollution in Beijing, China, Atmos. Chem. Phys., 18, 3523–3539, https://doi.org/10.5194/acp-18-3523-2018, 2018.

Hartmann, M., Adachi, K., Eppers, O., Haas, C., Herber, A., Holzinger, R., Hünerbein, A., Jäkel, E., Jentzsch, C., van Pinxteren, M., Wex, H., Willmes, S., and Stratmann, F.: Wintertime airborne measurements of ice nucleating particles in the high Arctic: A hint to a marine, biogenic source for ice nucleating particles, Geophysical Research Letters, 47, e2020GL087770, https://doi.org/10.1029/2020GL087770, 2020.

Kanji, Z. A., Ladino, L. A., Wex, H., Boose, Y., Burkert-Kohn, M., Cziczo, D. J., and Krämer, M.: Overview of Ice Nucleating Particles, Meteorological Monographs, 58, 1.1–1.33, https://doi.org/10.1175/AMSMONOGRAPHS-D-16-0006.1, 2017.

Kanji, Z. A., Welti, A., Corbin, J. C., and Mensah, A. A.: Black Carbon Particles Do Not Matter for Immersion Mode Ice Nucleation, Geophysical Research Letters, 47, e2019GL086764, https://doi.org/10.1029/2019GL086764, 2020.

Karlsson, L., Krejci, R., Koike, M., Ebell, K., and Zieger, P.: The role of nanoparticles in Arctic cloud formation, Atmos. Chem. Phys. Discuss., https://doi.org/10.5194/acp-2020-417, in review, 2020.

Levin, E. J. T., McMeeking, G. R., DeMott, P. J., McCluskey, C. S., Carrico, C. M., Nakao, S., Stockwell, C. E., Yokelson, R. J., and Kreidenweis, S. M.: Ice-nucleating particle emissions from biomass combustion and the potential importance of soot aerosol, J. Geophys. Res. Atmos., 121 (10), 5888–5903, https://doi.org/10.1002/2016JD024879, 2016.

McCluskey, C. S., DeMott, P. J., Prenni, A. J., Levin, E. J. T., McMeeking, G. R., Sullivan, A. P., Hill, T. C. J., Nakao, S., Carrico, C. M., and Kreidenweis, S. M.: Characteristics of atmospheric ice nucleating particles associated with biomass burning in the US: Prescribed burns and wildfires, J. Geophys. Res. Atmos., 119, 10458–10470, https://doi.org/10.1002/2014JD021980, 2014.

Möhler, O., Adams, M., Lacher, L., Vogel, F., Nadolny, J., Ullrich, R., Boffo, C., Pfeuffer, T., Hobl, A.,Weiß, M., Vepuri, H. S. K., Hiranuma, N., and Murray, B. J.: The portable ice nucleation experiment PINE: a new online instrument for laboratory studies and automated

long term field observations of ice-nucleating particles, Atmos. Meas. Tech. Discuss., https://doi.org/10.5194/amt-2020-307, in review, 2020.

Moran-Zuloaga, D., Ditas, F., Walter, D., Saturno, J., Brito, J., Carbone, S., Chi, X., Hrabˇe de Angelis, I., Baars, H., Godoi, R. H. M., Heese, B., Holanda, B. A., Lavriˇc, J. V., Martin, S. T., Ming, J., Pöhlker, M. L., Ruckteschler, N., Su, H., Wang, Y., Wang, Q., Wang, Z., Weber, B.,Wolff, S., Artaxo, P., Pöschl, U., Andreae, M. O., and Pöhlker, C.: Long-term study on coarse mode aerosols in the Amazon rain forest with the frequent intrusion of Saharan dust plumes, Atmos. Chem. Phys., 18, 10055–10088, https://doi.org/10.5194/acp-18-10055-2018, 2018.

ÖNORM M 5852: Standard ÖNORM M 5852:2007, Austrian standards, air analysis – sampling for continuous immission monitoring, Committee 139, 2007.

Petters, M. D., Parsons, M. T. , Prenni, A. J., DeMott, P. J., Kreidenweis, S. M., Carrico, C. M., Sullivan, A. P., McMeeking, G. R., Levin, E., Wold, C. E., Collett, J. L. Jr., and Moosmüller, H.: Ice nuclei emissions from biomass burning, J. Geophys. Res., 114, D07209, https://doi.org/10.1029/2008JD011532, 2009.

Prenni, A. J., Petters, M. D., Kreidenweis, S. M., Heald, C. L., Martin, S. T., Artaxo, P., Garland, R. M., Wollny, A. G., and Pöschl, U.: Relative roles of biogenic emissions and Saharan dust as ice nuclei in the Amazon basin, Nature Geoscience, 2, 6, 402–405, https://doi.org/10.1038/ngeo517, 2009.

Prenni, A. J., DeMott, P. J., Sullivan, A. P., Sullivan, R. C., Kreidenweis, S. M., and Rogers, D. C.: Biomass burning as a potential source for atmospheric ice nuclei:Western wildfires and prescribed burns, Geophys. Res. Lett., 39, L11805, https://doi.org/10.1029/2012GL051915, 2012.

Rinaldi, M., Hiranuma, N., Santachiara, G., Mazzola, M., Mansour, K., Paglione, M., Rodriguez, C. A., Traversi, R., Becagli, S., Cappelletti, D.M., and Belosi, F.: Condensation and immersion freezing Ice Nucleating Particle measurements at Ny-Ålesund (Svalbard) during 2018: evidence of multiple source contribution, Atmos. Chem. Phys. Discuss., https://doi.org/10.5194/acp-2020-605, in review, 2020.

Schill, G. P., DeMott, P. J., Emerson, E. W., Rauker, A. M. C., Kodros, J. K., Suski, K. J., Hill, T. C. J., Levin, E. J. T., Pierce, J. R., Farmer, D. K., and Kreidenweis, S. M.: The contribution of black carbon to global ice nucleating particle concentrations relevant to mixed-phase clouds, Proceedings of the National Academy of Sciences, 2020; 202001674, https://doi.org/10.1073/pnas.2001674117, 2020.

Schneider, J., Höhler, K., Heikkilä, P., Keskinen, J., Bertozzi, B., Bogert, P., Schorr, T., Umo, N. S., Vogel, F., Brasseur, Z., Wu, Y., Hakala, S., Duplissy, J., Moisseev, D., Kulmala, M., Adams, M. P., Murray, B. J., Korhonen, K., Hao, L., Thomson, E. S., Castarède, D., Leisner, T., Petäjä, T., and Möhler, O.: The seasonal cycle of ice-nucleating particles linked to the

[revised manuscript text omitted]

**2.3.1 Freezing modes**

It should be noted that we cannot predict how our deposition/condensation freezing measurements would translate to the immersion freezing mode in a situation given in the atmosphere. Some conclusions may however be drawn from previous parallel measurements (unpublished) with the FRIDGE diffusion chamber and the FRIDGE droplet freezing assay in different environments during the FIN-03 (Storm Peak Laboratory, SPL, USA, 2015), GLACE (Jungfraujoch, JFJ, Switzerland, 2017) and PICNIC (Puy de Dome, PDD, France, 2018) campaigns. Daily average INP concentrations (i.e. one day sample and one night sample) covered three orders of magnitude at $-25\,^\circ$C. When transforming the INP concentrations to log-space, we find that the two operational modes are well-correlated (R = 0.81, N = 44), with the immersion freezing INPs being on average a factor of 10 higher than deposition/condensation INPs. In fact, the INP concentrations measured in the droplet freezing assay were always higher. One may speculate that both species simply covary for the reason of having the same sources and sinks, or that deposition INPs may represent just a subset of immersion INPs, when observed by FRIDGE, or both. We will present the results of this comparison in more detail in a forthcoming publication, in which we will further investigate how exactly the nucleation modes of both methods are connected to each other.

Except when noted otherwise, the discussion presented in section 3 will focus on the highest ice supersaturation(s) RH$_{ice}$ at each of the three examined activation temperatures (Tab. 2). At these highest saturation conditions, at or slightly above water saturation, we observe the highest INP concentrations. We expect the nucleation mechanism to be a mixture of deposition nucleation and condensation freezing. At lower supersaturations we qualitatively observe trends and variability in INPs that are similar, but at lower absolute concentration levels.

**2.4 Measurement Sites**

**2.4.1 Amazon Tall Tower Observatory – AZ**

[revised manuscript text omitted]

Samples were collected from the upper level of Atmospheric Physics Laboratory at the hilltop. The aerosol inlet was at 11 m above ground. A main flow of ambient air was pumped through a HORIBA ASS-370 type inlet (ÖNORM, 2007) with a 40 mm I.D. x 7 m length stainless steel tube into the laboratory. The PEAC7 collected aerosol isokinetically at 2 L min$^{-1}$ from the main flow through a nozzle of 2.2 mm diameter.

30  INP measurements from this site are labeled with the abbreviation TO.

**2.4.4 Zeppelin Observatory – SB**

The Zeppelin Observatory, operated by the Norwegian Polar Institute, is located on Zeppelin Mountain close to Ny-Ålesund in Svalbard (78.908° N, 11.881° E, 474 m AMSL). Svalbard, and Ny-Ålesund in particular, is a well-established site for Arctic

and atmospheric research. The scientific focus of the observatory is to characterize the Arctic atmosphere and identify relevant atmospheric processes in a changing Arctic climate. The mountain top Zeppelin Observatory was chosen for its elevated position, which  likely limited the effects of locally produced pollution and of sea spray from the surf zone. However, the observatory largely remains within the planetary boundary layer (Tunved

5   et al., 2013). The station is representative of the remote Arctic, making it a unique location to study atmospheric aerosol. A variety of trace gases, greenhouse gases, aerosol particles, heavy metals and other compounds are monitored continuously at Zeppelin. A whole air inlet was used for aerosol particle sampling according to the ACTRIS guideline for stations that are often embedded in clouds. The flow through the inlet was kept constant to ensure near isokinetic sampling conditions. A short description about the inlet characteristics of the Zeppelin Observatory can be found in Karlsson et al. (2020).

[revised manuscript text omitted]

5    Considering the vastly different geographical locations and environments of the four measurement sites, as well as the inherent variance of atmospheric transportation patterns over time, we do not expect to find simple answers by inspection of the frequency distributions. A few interesting features are, however, apparent and the log-normal fitting agrees very well with the shape of the INP frequency distributions, which means that the dilution effect may be of importance here. The log-normal shape of the $-25\,°C$ and $-30\,°C$ distributions is especially evident ($R^2$ ranges from 0.92 to 0.97). Here we observe unimodal

10    and regular bell shapes at all four sites. At $-20\,°C$ the fits are not as good ($R^2$ ranges from 0.74 to 0.91) and some distributions appear to be potentially bimodal  (e.g. SB). However, the strength of the fit may also be related to the fact that at $-20\,°C$ few ice crystals activate on each sample substrate, introducing a relatively high uncertainty in the INP concentration. Consequently, the incrementation is not ideal for $-20\,°C$, because measured concentrations are often near the limit of detection and have a poor resolution. This explanation is self-consistent with the observed minimum $R^2$ found for SB, where the distribution is

15    heavily skewed to the right. In addition to reflecting the generally low INP concentration of the Arctic environment, this may point to reduced biological activity over much of the year. Interestingly, the shape of the distribution at TO seems to indicate a slight shift towards higher concentrations, pointing to a potential local source of INPs. However, at lower temperatures we do not find this feature. This could mean that, in addition to whatever long-range transported aerosols contribute to INPs at TO, there might be a biological source from the surrounding forest. However, there is no strong evidence for such a signal in our

20    data overall, possibly due to the comparably low sampling volume. As a result the temperature range of our measurements overlaps only very little with the regime where biological particles nucleate. Remarkably, such a feature seems to be entirely absent from the Amazonian rainforest site, where one would more readily expect to find a local source of primary biological particles that may be potential INPs. On the other hand, surface temperatures never drop below $0\,°C$ in the Amazon. Therefore, local species of plants or bacteria may be less likely to have evolved traits that induce freezing. It has previously

25    been posited that some microbiology (e.g. bacteria like *Pseudomonas syringae*) gain an evolutionary  advantage by being able to induce freezing (Morris et al., 2014).

At $-25\,°C$ we find relatively minor differences between the four sites. SB concentrations are slightly shifted to lower concentrations and the spectrum at MQ concentrations is slightly broader. Differences are more apparent at $-30\,°C$. Here we find distinctly dissimilar shapes of INP concentration frequency distributions. SB and AZ exhibit narrow peaks relative to the

30    more broad shapes of TO and MQ. The curves are also more distinctly separated in concentration space, with the maximum of the distribution at a minimum concentration for SB, followed by TO, MQ and AZ.

Supplementary Fig.  S6 visualizes the information presented in Fig. 7 as function of the relative humidity. The occurrence frequency is color-coded, with cool colors indicating a low and warm colors a high likelihood of this INP concentration at a given saturation condition. Thus Fig.  S6 can be understood as follows: a single column (e.g. the rightmost column) gives

35    the full frequency distribution of a single measurement condition (e.g. 135 % $RH_{ice}$, corresponds to Fig. 7c). Fewer warm

colors appear in a column, when the distribution of INP concentrations is broad at that condition. Conversely, fewer cool tones indicate a narrow distribution. The respective median INP concentration will be close to the maximum of the relative frequency at each condition. Consequently, following the maxima yields information about the steepness of the INP spectra, similar to what is depicted in Fig. 6.

Overall, Figs. 7 and  S6 suggest that the INP concentrations measured in the investigated temperature regime at these stations are largely defined by background air masses, and that local sources are only of secondary importance. More discussion of the site specific local sources and characteristic features is provided in the following section.

**3.2 Site specific INP characteristics**

At each measurement station a diverse array of supplementary meteorological, aerosol and gas data from the stations were collected in parallel to the INP sampling. Unfortunately, the parameters, instrumentation and time coverage vary considerably between the four sites. Observations include typical meteorological parameters such as temperature, relative humidity, precipitation, etc., as well as the total aerosol particle number and mass concentrations, aerosol size distributions, black carbon concentrations, aerosol optical thickness, gaseous pollutant markers and greenhouse gases. However, despite a rigorous effort including correlation analysis, factor analysis and trajectory sector analysis, we were ultimately unable to identify a single parameter or a set of parameters that account for the total observed variation of INPs. This highlights the complex nature of the ice nucleation process and the particles involved. Whereas similar but somewhat larger-scale long-term measurements of CCN are able to largely explain the corresponding variability and provide closure studies (Schmale et al., 2018), unfortunately, the same cannot yet be said for INPs.

Although a common, definitive driver of INP climatology was not identified in our study, we will point out a few key findings specific to the respective measurement sites.

**3.2.1 AZ**

The Amazonian site is characterized by a distinct seasonality of pollutants that follow the biomass burning season. During the dry season (August to November) the aerosol concentration and other pollution markers rise by about one order of magnitude compared to the cleaner wet season (February to May) – a change which is largely attributable to human activities. Notably, an effect of the strong anthropogenic biomass burning is absent in the INP signal. In fact, the number of INPs normalized by the total number of aerosol particles (TSI OPS 3330) in a volume of air (i.e. the activated fraction AF) is anti-correlated to parameters related to  the abundance of biomass burning products (Fig. 8). The AF can be understood as a simple metric that indicates the ice nucleating efficiency of particles within a specific aerosol sample. The observed anti-correlation  seems to suggest that aerosol particles from  fires are relatively poor ice nuclei; an observation that agrees with previously published findings (Kanji et al., 2017, 2020).  Considering the recent literature consensus regarding biomass burning INPs, these results are not unexpected. Biomass burning INPs have been studied in the laboratory investigating both surrogate and real combustion particles (Petters et al., 2009; Levin et al., 2016; Kanji et al., 2020), and in the field (Prenni et al., 2012; McCluskey et al., 2014; Schill et al., 2020). Although at least a regional impact of biomass burning

on INP abundance is reported, the nucleation temperatures are usually close to the homogeneous freezing limit. Some of these studies suggest that water supersaturation is a requirement for biomass burning aerosol to act as INPs (e.g. Petters et al., 2009; Schill et al., 2 . In this regard, our data may demonstrate the limits of what is explorable with FRIDGE. Either biomass burning aerosol is in fact a poor source for Amazonian INPs in the investigated temperature regime or the method simply cannot represent the freezing behavior of these particles accurately.

Furthermore, the significance of low AFs resulting from biomass burning in this study is difficult to assess, as the seasonality of the AF is largely dominated by the seasonal changes in aerosol concentration for the AZ site. Vegetation fires therefore seem to emit disproportionally more (non ice-active) aerosol particles than INPs. Another way to interpret the anti-correlation of AF and  biomass burning markers is by coupling the metric to precipitation rates. There are several intricate interactions of note here. On  one hand more precipitation leads to  higher aerosol particle (and INP) removal by wet deposition. Moreover,  enhanced precipitation during the wet season  can largely prevent wild fires and the accompanied particle emissions in the first place. On the other hand, it has been postulated previously that precipitation may be  a driver of biological INPs (Huffman et al., 2013), and large tropical rainforests like the Amazon have been highlighted in that regard (Morris et al., 2014).  However, the processes responsible for the release of the biological particles have not yet been deciphered in detail.  Moreover, although the AZ measurements are somewhat more sparse than those of other stations, our observations do not support significant differences in absolute INP concentrations between dry and wet seasons.

Overall, the INP concentrations of our study compare reasonably well to the measurements of Prenni et al. (2009), who observed average INP concentrations of about $1\,L^{-1}$ at $-20\,°C$, $4\,L^{-1}$ at $-25\,°C$ and $10\,L^{-1}$ at $-30\,°C$ using a continuous flow diffusion chamber (CFDC) to study condensation and immersion mode ice nucleation during a field campaign in February-/ March 2008 in a region close to the present location of the ATTO site. However, our observed concentrations are clustered at the low end of  those presented by Prenni et al. (2009) (i.e. a factor of 5 lower on average at $-30\,°C$), which is  presumably due to the different nucleation modes addressed. During that short campaign Prenni et al. (2009) identified mineral dust and carbonaceous aerosol (mostly biological particles) to be the main contributors to atmospheric INPs in the Amazon using transmission electron microscopy and energy-dispersive X-ray spectroscopy.

Within our sampling period, Moran-Zuloaga et al. (2018) identified several long-range transport (LRT) events at the site with markedly increased concentrations of mineral dust during the wet season of 2015/2016 (Dec./Jan.). INP concentrations of these LRT samples were positively correlated with the aerosol number concentration measured with an optical particle counter (TSI OPS 3330, R = 0.80, N = 9, p < 0.01). However, mineral dust may be a relevant INP in this region even in the absence of distinct LRT events: An analysis of the average composition of INPs  of six samples (4 in April 2016, 2 in December 2016) using scanning electron microscopy (SEM, Figure 9), identified that nearly half of the particles that activated to ice crystals

in FRDIGE were mineral dust.  This finding suggests that there seems to be a well-mixed and diluted background concentration of mineral dust INPs at all times present at AZ. The diameter of most of the INPs investigated by SEM in this study  was between one and a couple of micrometers  (Figure 9b). Note however, that the contribution to the larger size bins might be potentially underrepresented due to particle losses from the inlet configuration. The second half of identified INPs had a strong carbonaceous fraction and consisted of biological particles and biomass burning products. Furthermore, it is possible that some PBAP activity was missed due to the chosen sampling strategy, given local noon is a daily minimum for PBAPs. Qualitatively, these findings agree very well to those of Prenni et al. (2009).

**3.2.2 MQ**

Of the results presented here, the average INP concentration of the Caribbean site was the highest, but only by a small margin. There is some evidence that summertime INP concentrations are higher on average than those during winter, although there is no clear seasonality. However, the possible seasonal ice nucleation effects are difficult to assess due to the large interruption of measurements between December 2015 to May 2016. Although we consider it rather speculative, a trend of higher concentrations during summer does stand to reason, as it would reflect the annual cycle of the mineral dust transport, which is driven by the movement of the ITCZ. The seasonality of mineral dust is well reflected by the $PM_{10}$ concentration, which is monitored routinely in Martinique by the local agency for air quality (MadininAir). The seasonality of dust motivates a deeper investigation with respect to INPs. In general, we observe a significant correlation between the INP concentration at OVSM and the $PM_{10}$ concentration at an air quality station close to the observatory (Schoelcher, $14\,\mathrm{km}$ distance), as well as between INPs and the OPS aerosol number concentration at the observatory. The correlations improve for colder temperatures and higher ice supersaturations. At $-30\,^{\circ}\mathrm{C}$ and $135\,\%\,\mathrm{RH_{ice}}$ the Pearson correlation coefficients between INP and aerosols are $R = 0.46$ ($N = 124$, $p \ll 0.01$) for $PM_{10}$ (Fig. S7) and $R = 0.50$ ($N = 69$, $p \ll 0.01$) for the OPS concentration, respectively. We conclude that the MQ INP concentration at the investigated temperatures is likely dominated by natural processes such as the long-range transport of Saharan mineral dust. However, there is still a large variability in the INP signal, which cannot be fully explained  by considering only the seasonal dust transport.

We observe significantly lower INP concentrations for all conditions after the large interruption in measurements. For example, the average INP concentration at $-30\,^{\circ}\mathrm{C}$ and $135\,\%\,\mathrm{RH_{ice}}$ in 2015 was $7.47 \pm 6.42\,\mathrm{L^{-1}}$ ($N = 58$) and only $1.37 \pm 1.39\,\mathrm{L^{-1}}$ ($N = 72$) in 2016. This observation does, however, correspond to measured $PM_{10}$ concentrations, which also show a significantly lower average in 2016 ($25\,\mathrm{\mu g\,m^{-3}}$, $N = 8225$) than 2015 ($35\,\mathrm{\mu g\,m^{-3}}$, $N = 8636$). Although, the observed 2015 to 2016 factor of 5 decrease in INP concentrations is large compared to the $\approx 30\%$ difference in $PM_{10}$, the cubic scaling implicit in the number to mass translation needs to be considered.

DeMott et al. (2016) presented results from offline immersion freezing experiments and characterized INP concentrations from research flights from St. Croix in the US Virgin Islands, and ground sampling from Puerto Rico, which were collected during the ICE-T campaign in July 2011. The focus was on marine INPs and determining representative marine background concentrations and only samples collected within the marine boundary layer were presented. They measured INP concentrations of $0.06\,L^{-1}$ at $-20\,^{\circ}C$ and $0.3\,L^{-1}$ at $-24\,^{\circ}C$, which  agrees within a factor of  two to our median INP concentrations at $-20\,^{\circ}C$ and $-25\,^{\circ}C$.

**3.2.3 TO**

During the time frame of the global sampling effort (about 640 days) 400 PEAC7 samples were collected and analyzed from TO (i.e. 1 sample every 1.6 days). The sampling frequency of valid INP concentrations (i.e. above the detection limit) remains as good as 1 sample every 2 days for measurements at $-20\,^{\circ}C$ and $-25\,^{\circ}C$. This is by far the best data coverage of the four stations.

We found a moderate but significant correlation between the $PM_{10}$ concentrations and INPs throughout the spectrum of $T$ and $RH$ conditions. The Pearson correlation coefficient is as high as R =  0.27 (N = 304, $p \ll 0.01$) at $-25\,^{\circ}C$, where we have the best data coverage. Although the particulate matter was significantly enhanced, when wind was coming from the heavily populated and industrialized Rhine-Main metropolitan region (Tab. 3), the average INP concentration was not found to differ significantly from other times, when air masses were arriving from other directions (Fig. 10). Therefore, a strong anthropogenic impact on INPs at TO is unlikely.

**3.2.4 SB**

 Due to its remoteness and relatively clean atmosphere, the Arctic may be particularly sensitive to small changes in aerosol particulate. Furthermore, within the Arctic climate system there are well known feedbacks that can amplify small changes in significant ways (Serreze and Francis, 2006; Boy et al., 2019). Historically, this has motivated quite a few research studies targeting ice nucleation in the Arctic environment. For example, clay was identified in the center of Greenlandic snow crystals by Kumai and Francis (1962) as early as 1960. Past studies generally agree that INP concentrations in the Arctic tend to be on the lower side of the spectrum. Yet, individual findings and conclusions vary considerably (e.g. see Tab. 2 in Thomson et al., 2018). New ice core records may illuminate long-term trends of Arctic INPs by estimating historic (pre-industrial) concentrations from droplet freezing experiments of ice core melt water (Hartmann et al., 2019; Schrod et al., 2020).

In two recent studies immersion mode ice nucleation in the Arctic was investigated by Tobo et al. (2019) and Wex et al. (2019). Tobo et al. (2019) focused on two field campaigns held in Ny-Ålesund (Zeppelin) in July 2016 (6 samples) and March 2017 (7 samples). Wex et al. (2019) report INP concentrations from four pan-Arctic locations (Canada, Alaska, Ny-Ålesund and Greenland) that cover observations ranging from 10 weeks to a full year of mostly weekly sampling. Both studies observed enhanced INP concentrations during summer months. Tobo et al. (2019) report INP concentrations at $-20\,^{\circ}C$ of about $0.01\,L^{-1}$ in March 2017 and about $0.1\,L^{-1}$ in July 2016. At $-25\,^{\circ}C$ INP concentrations were on the order of $0.1\,L^{-1}$ and $0.5\,L^{-1}$ for the March and July field campaigns, respectively. Wex et al. (2019) distinguished between samples that were collected in Ny-Ålesund from March to May 2012 (5 samples) and those from June to September 2012 (7 samples). During spring, INP concentrations at $-20\,^{\circ}C$ were consistently found to be about $0.01\,L^{-1}$. Most summertime samples were completely frozen

before reaching $-20\,°C$, and thus seem to to suggest that concentrations were up to one order of magnitude higher in summer. Very recent measurements from Greenland during March/April 2018 qualitatively agree very well to these concentration levels (Hartmann et al., 2020). However, a recent study by Rinaldi et al. (2020) did not observe a distinct seasonal signal in their INP measurements between $-15\,°C$ and $-22\,°C$ in the spring and summer of 2018 in Ny-Ålesund. Rinaldi et al. (2020) present

5  INP concentrations from two separate methods, one of which is fairly similar to FRIDGE, addressing the condensation freezing (DFPC) and immersion freezing (WT-CRAFT) modes.

Further, Wex et al. (2019) report correlation coefficients with complementary measurements that are mostly insignificant including $PM_{10}$. Exceptions include significant correlations between INPs and sulphate (R = $-0.6$) and potassium (R = $-0.57$), pointing to complex factors determining the Arctic INP population. Moreover, Tobo et al. (2019) present evidence that mineral

10  dust (possibly with organic inclusions) from Arctic glacial outwash plains influence the INP activity in Ny-Ålesund. They conclude that these glacial sediments may be a large-scale source of mineral dust in the Arctic. Rinaldi et al. (2020) present evidence that Arctic INP concentrations are influenced by sources of marine biological INPs by providing a spatio-temporal correlation analysis between Chlorophyll-a fields from satellite data and a trajectory model.

We present a significantly larger data set with respect to temporal coverage and our INP concentrations agree well with these

15  previous studies from Ny-Ålesund. At $-20\,°C$ we find concentrations of about $0.1\,L^{-1}$. At $-25\,°C$ the average INP concentration increases to about $0.3\,L^{-1}$. However, the  frequently reported finding of summertime INP enhancement, does not emerge from our analysis. Furthermore, we did not observe any seasonal changes in the INP signal with regards to the anthropogenic Arctic Haze phenomenon. Moreover, we did not observe significant correlations between INPs and available aerosol parameters. The concerning lack of meaningful correlations and/or seasonal trends may be in part related to a relatively

20  poor signal-to-noise ratio  in our SB measurements. INP concentrations were often at or close to the limit of detection or the significance level, respectively. In retrospect, we now would increase the sampling volume for SB measurements to be able to resolve lower concentrations more accurately.

**4  Conclusions**

The data from our small but unique measurement network can be considered particularly valuable, and we hope lessons can

25  be  learned from this effort that will help to guide future INP monitoring efforts. Significant infrastructural and logistical investments are represented by the INP measurements that cover an observational period of 21 months in total. Well above 1000 samples were collected, retrieved and analyzed in this project at a large array of temperature and supersaturation conditions, characterizing the INP concentrations in the deposition and condensation freezing modes. The investigated sites represent diverse climatic regions and ecosystems that experience varying degrees of anthropogenic influence.

30  In spite of the great differences in basically all characteristics that are expected to define the aerosol concentration, composition and source apportionment, we observed fairly similar INP concentrations for all four stations  for the methods

and sampling strategy applied. In our study, average concentrations differed between sites by less than a factor of 5. Short-term variability dominated most of the total variability at all locations. Trends, annual cycles and well-defined peak concentrations were prominently absent from the time series. Still, the range of observed INP concentrations do compare reasonably well with previously published literature, where available. Importantly, the relative frequencies of observed INP concentrations are generally well-represented by log-normal distributions, a finding that suggests distributed INP sources that result from INPs being well-mixed within sampled air masses. These findings emphasize the important contribution of INPs from background air masses. Moreover, no physical or chemical parameter was identified to continuously co-vary with INPs at all sites, and therefore a comprehensive causal link to INP concentrations remains lacking.

Overall, we did not detect much evidence for a strong anthropogenic impact on the concentrations of ice nucleating particles. At AZ the INP concentrations appear unrelated to human induced biomass burning, which otherwise leads to a tenfold increase in aerosol particle number concentrations during the dry season. The INP concentrations at MQ were well correlated with aerosol characteristics that are driven by natural processes, like long-range transport of Saharan mineral dust and marine aerosol production. Average TO INP concentrations showed no significant difference between wind sectors that can be separated into anthropogenically dominated areas and rural environments. Likewise, no significant changes in the INP concentration were observed at SB during the Arctic Haze period.

Considering these findings, the approach of estimating order-of-magnitude pre-industrial INP concentrations from present-day measurements in near-pristine locations does seem to both be viable and yields reasonable results, which merit further investigation. In this sense, we consider the lower concentration end of our measurements likely to be the most realistic assessment of pre-industrial atmospheric INP concentrations. However,  when using the presented data

 one should be aware of the substantial limitations of the conceptual aspect of the approach and the uncertainties that are inherent in the aerosol sampling and INP measurements themselves.

**5   Outlook**

[revised manuscript text omitted]

In addition to the goal of establishing more long-term global observations of continuous INP concentrations there are certainly other important areas for future research to address. For example, as most measurements are conducted at ground level, we believe there is a need to systematically study the vertical distribution of INPs – for example at heights where INPs are transported over long-ranges and/or where cloud formation occurs. Moreover, more extensive data sets from long-term INP monitoring might shed light on what mechanisms result in the observed log-normal INP frequency distributions (and

departures from ideality etc.) as presented here and, for example, by Welti et al. (2018). Murray et al. (2020) has recently enumerated many crucial areas into which future INP research should delve. First and foremost, the authors emphasize the need to accurately implement ice nucleation related cloud-phase interactions in climate models in order to predict future climate scenarios correctly. We gladly refer the interested reader to Murray et al. (2020) for a more extensive list of future ice nucleation related research questions, as is presented in this study.

[revised manuscript text omitted]

Lacher, L.

Lee, Y. H., Chen, K., and Adams, P. J.: Development of a global model of mineral dust aerosol microphysics, Atmos. Chem. Phys., 9, Vogel,
25    F., Nadolny, J.2441–2458, Adams, M. https://doi.org/10.5194/acp-9-2441-2009, Murray, B.2009.

Levin, E. J. T., McMeeking, G. R., DeMott, P. J., BoffoMcCluskey, C. , Pfeuffer, T. , and Möhler, O. : The Portable Ice Nucleation Experiment (PINE): A New Instrument for Semiautonomous Measurements of Atmospheric Ice Nucleating Particles, in: 99th AMS Annual Meeting, Phoenix, USA, 6–10 January, 2019. S., Carrico, C. M., Nakao, S., Stockwell, C. E., Yokelson, R. J., and Kreidenweis, S. M.: Ice-nucleating particle emissions from biomass combustion and the potential importance of soot aerosol, J. Geophys. Res. Atmos., 121 (10), 5888–5903,
30    https://doi.org/10.1002/2016JD024879, 2016.

Lohmann, U.: Aerosol-Cloud Interactions and Their Radiative Forcing, Encyclopedia of Atmospheric Sciences, 1, 17–22, https://doi.org/10.1016/B978-0-12-382225-3.00052-9, Amsterdam: Elsevier Academic Press, 2015.

MadininAir, Ministry of Ecology, Energy, Sustainable Development and the Sea for monitoring air quality in Martinique: https://www.madininair.fr/, last access: 23 June 2020.

35    Marcolli, C.: Deposition nucleation viewed as homogeneous or immersion freezing in pores and cavities, Atmospheric Chemistry and Physics, 14, 1, 2071–2104, https://doi.org/10.5194/acp-14-2071-2014, 2014.

Marinou, E., Tesche, M., Nenes, A., Ansmann, A., Schrod, J., Mamali, D., Tsekeri, A., Pikridas, M., Baars, H., Engelmann, R., Voudouri, K.-A., Solomos, S., Sciare, J., Groß, S., Ewald, F., and Amiridis, V.: Retrieval of ice-nucleating particle concentrations from lidar observations

and comparison with UAV in situ measurements, Atmos. Chem. Phys., 19, 11315—11342, https://doi.org/10.5194/acp-19-11315-2019, 2019.

McCluskey, C. S., DeMott, P. J., Prenni, A. J., Levin, E. J. T., McMeeking, G. R., Sullivan, A. P., Hill, T. C. J., Nakao, S., Carrico, C. M., and Kreidenweis, S. M.: Characteristics of atmospheric ice nucleating particles associated with biomass burning in the US: Prescribed burns and wildfires, J. Geophys. Res. Atmos., 119, 10458–10470, https://doi.org/10.1002/ 2014JD021980, 2014.

Möhler, O., Adams, M., Lacher, L., Vogel, F., Nadolny, J., Ullrich, R., Boffo, C., Pfeuffer, T., Hobl, A., Weiß, M., Vepuri, H. S. K., Hiranuma, N., and Murray, B. J.: The portable ice nucleation experiment PINE: a new online instrument for laboratory studies and automated long-term field observations of ice-nucleating particles, Atmos. Meas. Tech. Discuss., https://doi.org/10.5194/amt-2020-307, in review, 2020.

Moran-Zuloaga, D., Ditas, F., Walter, D., Saturno, J., Brito, J., Carbone, S., Chi, X., Hrabě de Angelis, I., Baars, H., Godoi, R. H. M., Heese, B., Holanda, B. A., Lavrič, J. V., Martin, S. T., Ming, J., Pöhlker, M. L., Ruckteschler, N., Su, H., Wang, Y., Wang, Q., Wang, Z., Weber, B., Wolff, S., Artaxo, P., Pöschl, U., Andreae, M. O., and Pöhlker, C.: Long-term study on coarse mode aerosols in the Amazon rain forest with the frequent intrusion of Saharan dust plumes, Atmos. Chem. Phys., 18, 10055–10088, https://doi.org/10.5194/acp-18-10055-2018, 2018.

Morris, C. E., Conen, F., Huffman, J. A., Phillips, V., Pöschl, U., and Sands, D. C.: Bioprecipitation: a feedback cycle linking Earth history, ecosystem dynamics and land use through biological ice nucleators in the atmosphere, Glob. Change Biol., 20, 341–351, https://doi.org/10.1111/gcb.12447, 2014.

Müller, W.: Über den Einfluss meteorologischer Bedingungen auf die Gefrierkerndichte der Luft, Arch. Met. Geoph. Biokl., Ser. A, 18, 55–74, https://doi.org/10.1007/BF02247864, 1969.

Mülmenstädt, J., Sourdeval, O., Delanoë, J., and Quaas, J.: Frequency of occurrence of rain from liquid-, mixed-, and ice-phase clouds derived from A-Train satellite retrievals, Geophys. Res. Lett., 42, 6502–6509, https://doi.org/10.1002/2015GL064604, 2015.

Murray, B. J.,  Carslaw, K. S., and Field, P. R.: Opinion: Cloud-phase climate feedback and the importance of ice-nucleating particles,  Atmos. Chem. Phys. Discuss., https://doi.org/10.5194/acp-2020-852, in review, 2020.

Niemand, M., Möhler, O., Vogel, B., Vogel, H., Hoose, C., Connolly, P., Klein, H., Bingemer, H., DeMott, P. and Skrotzki, J.: A particle-surface-area-based parameterization of immersion freezing on desert dust particles, J. Atmos. Sci., 69, 3077–3092, https://doi.org/10.1175/JAS-D-11-0249.1., 2012.

ÖNORM M 5852: Standard ÖNORM M 5852:2007, Austrian standards, air analysis – sampling for continuous immission monitoring, Committee 139, 2007.

O'Sullivan, D., Murray, B. J., Ross, J. F., Whale, T. F., Price, H. C., Atkinson, J. D., Umo, N. S., and Webb, M. E.: The relevance of nanoscale biological fragments for ice nucleation in clouds, Scientific Reports, 5, 8082, https://doi.org/10.1038/srep08082, 2015.

O'Sullivan, D., Adams, M. P., Tarn, M. D., Harrison, A. D., Vergara-Temprado, J., Porter, G., Holden, M. A., Sanchez-Marroquin, A., Carotenuto, F., Whale, T. F., McQuaid, J. B., Walshaw, R., Hedges, D., Burke, I. T., Cui, Z., and Murray, B. J.: Contributions of biogenic material to the atmospheric ice-nucleating particle population in North Western Europe, Scientific reports, 8(1), 13821, https://doi.org/10.1038/s41598-018-31981-7, 2018.

Ott, W.: A Physical Explanation of the Lognormality of Pollutant Concentrations, J. Air Waste Manag. Assoc., 40, 137–1383, https://doi.org/10.1080/10473289.1990.10466789, 1990.

Petters, M. D., Parsons, M. T. , Prenni, A. J., DeMott, P. J., Kreidenweis, S. M., Carrico, C. M., Sullivan, A. P., McMeeking, G. R., Levin, E., Wold, C. E., Collett, J. L. Jr., and Moosmüller, H.: Ice nuclei emissions from biomass burning, J. Geophys. Res., 114, D07209, https://doi.org/10.1029/2008JD011532, 2009.

PNRM, Parc Naturel Régional de la Martinique website: http://pnr-martinique.com/la-charte-du-pnrm/, last access: 06 March 2020.

5   Pöhlker, M. L., Pöhlker, C., Ditas, F., Klimach, T., Hrabe de Angelis, I., Araújo, A., Brito, J., Carbone, S., Cheng, Y., Chi, X., Ditz, R., Gunthe, S. S., Kesselmeier, J., Könemann, T., Lavrič, J. V., Martin, S. T., Mikhailov, E., Moran-Zuloaga, D., Rose, D., Saturno, J., Su, H., Thalman, R., Walter, D., Wang, J., Wolff, S., Barbosa, H. M. J., Artaxo, P., Andreae, M. O., and Pöschl, U.: Long-term observations of cloud condensation nuclei in the Amazon rain forest – Part 1: Aerosol size distribution, hygroscopicity, and new model parametrizations for CCN prediction, Atmos. Chem. Phys., 16, 15709–15740, https://doi.org/10.5194/acp-16-15709-2016, 2016.

10  Pöhlker, M. L., Ditas, F., Saturno, J., Klimach, T., Hrabe de Angelis, I., Araujo, A. C., Brito, J., Carbone, S., Cheng, Y., Chi, X., Ditz, R., Gunthe, S. S., Holanda, B. A., Kandler, K., Kesselmeier, J., Könemann, T., Krüger, O. O., Lavric, J. V., Martin, S. T., Mikhailov, E., Moran-Zuloaga, D., Rizzo, L. V., Rose, D., Su, H., Thalman, R., Walter, D., Wang, J., Wolff, S., Barbosa, H. M. J., Artaxo, P., Andreae, M. O., Pöschl, U., and Pöhlker, C.: Long-term observations of cloud condensation nuclei over the Amazon rain forest – Part 2: Variability and characteristics of biomass burning, long-range transport, and pristine rain forest aerosols, Atmos. Chem. Phys., 18, 10289–10331,

15  https://doi.org/10.5194/acp-18-10289-2018, 2018.

Pöhlker, C., Walter, D., Paulsen, H., Könemann, T., Rodriguez-Caballero, E., Moran-Zuloaga, D., Brito, J., Carbone, S., Degrendele, C., Després, V. R., Ditas, F., Holanda, B. A., Kaiser, J. W., Lammel, G., Lavrič, J. V., Ming, J., Pickersgill, D., Pöhlker, M. L., Praß, M., Löbs, N., Saturno, J., Sörgel, M., Wang, Q., Weber, B., Wolff, S., Artaxo, P., Pöschl, U., and Andreae, M. O.: Land cover and its transformation in the backward trajectory footprint region of the Amazon Tall Tower Observatory, Atmos. Chem. Phys., 19, 8425–8470,

20  https://doi.org/10.5194/acp-19-8425-2019, 2019.

Prenni, A. J., Petters, M. D., Kreidenweis, S. M., Heald, C. L., Martin, S. T., Artaxo, P., Garland, R. M., Wollny, A. G., and Pöschl, U.: Relative roles of biogenic emissions and Saharan dust as ice nuclei in the Amazon basin, Nature Geoscience, 2, 6, 402–405, https://doi.org/10.1038/ngeo517, 2009.

Prenni, A. J., DeMott, P. J., Sullivan, A. P., Sullivan, R. C., Kreidenweis, S. M., and Rogers, D. C.: Biomass burning as a potential source for

25  atmospheric ice nuclei: Western wildfires and prescribed burns, Geophys. Res. Lett., 39, L11805, https://doi.org/10.1029/2012GL051915, 2012.

Prospero, J. M.: Saharan Dust Transport Over the North Atlantic Ocean and Mediterranean: An Overview, In: Guerzoni S., Chester R. (eds) The Impact of Desert Dust Across the Mediterranean, Environmental Science and Technology Library, 11, Springer, Dordrecht, https://doi.org/10.1007/978-94-017-3354-0_130, 1996.

30  Prospero, J. M. and Lamb, P. J.: African Droughts and Dust Transport to the Caribbean: Climate Change Implications, Science, 302, 5647, 1024–1027, https://doi.org/10.1126/science.1089915, 2003.

Rinaldi, M., Hiranuma, N., Santachiara, G., Mazzola, M., Mansour, K., Paglione, M., Rodriguez, C. A., Traversi, R., Becagli, S., Cappelletti, D. M., and Belosi, F.: Condensation and immersion freezing Ice Nucleating Particle measurements at Ny-Ålesund (Svalbard) during 2018: evidence of multiple source contribution, Atmos. Chem. Phys. Discuss., https://doi.org/10.5194/acp-2020-605, in review, 2020.

35  Saturno, J., Holanda, B. A., Pöhlker, C., Ditas, F., Wang, Q., Moran-Zuloaga, D., Brito, J., Carbone, S., Cheng, Y., Chi, X., Ditas, J., Hoffmann, T., Hrabe de Angelis, I., Könemann, T., Lavric, J. V., Ma, N., Ming, J., Paulsen, H., Pöhlker, M. L., Rizzo, L. V., Schlag, P., Su, H., Walter, D., Wolff, S., Zhang, Y., Artaxo, P., Pöschl, U., and Andreae, M. O.: Black and brown carbon over central Amazonia:

long-term aerosol measurements at the ATTO site, Atmos. Chem. Phys., 18, 12817–12843, https://doi.org/10.5194/acp-18-12817-2018, 2018.

Schill, G. P., DeMott, P. J., Emerson, E. W., Rauker, A. M. C., Kodros, J. K., Suski, K. J., Hill, T. C. J., Levin, E. J. T., Pierce, J. R., Farmer, D. K., and Kreidenweis, S. M.: The contribution of black carbon to global ice nucleating particle concentrations relevant to mixed-phase clouds, Proceedings of the National Academy of Sciences, 2020; 202001674, https://doi.org/10.1073/pnas.2001674117, 2020.

Schmale, J., Henning, S., Decesari, S., Henzing, B., Keskinen, H., Sellegri, K., Ovadnevaite, J., Pöhlker, M. L., Brito, J., Bougiatioti, A., Kristensson, A., Kalivitis, N., Stavroulas, I., Carbone, S., Jefferson, A., Park, M., Schlag, P., Iwamoto, Y., Aalto, P., Äijälä, M., Bukowiecki, N., Ehn, M., Frank, G., Fröhlich, R., Frumau, A., Herrmann, E., Herrmann, H., Holzinger, R., Kos, G., Kulmala, M., Mihalopoulos, N., Nenes, A., O'Dowd, C., Petäjä, T., Picard, D., Pöhlker, C., Pöschl, U., Poulain, L., Prévôt, A. S. H., Swietlicki, E., Andreae, M. O., Artaxo, P., Wiedensohler, A., Ogren, J., Matsuki, A., Yum, S. S., Stratmann, F., Baltensperger, U., and Gysel, M.: Long-term cloud condensation nuclei number concentration, particle number size distribution and chemical composition measurements at regionally representative observatories, Atmos. Chem. Phys., 18, 2853–2881, https://doi.org/10.5194/acp-18-2853-2018, 2018.

Schneider, J., Höhler, K., Heikkilä, P., Keskinen, J., Bertozzi, B., Bogert, P., Schorr, T., Umo, N. S., Vogel, F., Brasseur, Z., Wu, Y., Hakala, S., Duplissy, J., Moisseev, D., Kulmala, M., Adams, M. P., Murray, B. J., Korhonen, K., Hao, L., Thomson, E. S., Castarède, D., Leisner, T., Petäjä, T., and Möhler, O.: The seasonal cycle of ice-nucleating particles linked to the abundance of biogenic aerosol in boreal forests, Atmos. Chem. Phys. Discuss., https://doi.org/10.5194/acp-2020-683, in review, 2020.

[revised manuscript text omitted]